# Egg multivesicular bodies elicit an LC3-associated phagocytosis-like pathway to degrade paternal mitochondria after fertilization

Sharon Ben-Hur [1], Shoshana Sernik[1], Sara Afar[1], Alina Kolpakova [1], Yoav Politi[1], Liron Gal[1], Anat Florentin [1,5], Ofra Golani [2], Ehud Sivan[2], Nili Dezorella [3], David Morgenstern[4], Shmuel Pietrokovski [1], Eyal Schejter [1], Keren Yacobi-Sharon [1] & Eli Arama [1] ✉

Mitochondria are maternally inherited, but the mechanisms underlying paternal mitochondrial elimination after fertilization are far less clear. Using *Drosophila*, we show that special egg-derived multivesicular body vesicles promote paternal mitochondrial elimination by activating an LC3-associated phagocytosis-like pathway, a cellular defense pathway commonly employed against invading microbes. Upon fertilization, these egg-derived vesicles form extended vesicular sheaths around the sperm flagellum, promoting degradation of the sperm mitochondrial derivative and plasma membrane. LC3-associated phagocytosis cascade of events, including recruitment of a Rubicon-based class III PI(3)K complex to the flagellum vesicular sheaths, its activation, and consequent recruitment of Atg8/LC3, are all required for paternal mitochondrial elimination. Finally, lysosomes fuse with strings of large vesicles derived from the flagellum vesicular sheaths and contain degrading fragments of the paternal mitochondrial derivative. Given reports showing that in some mammals, the paternal mitochondria are also decorated with Atg8/LC3 and surrounded by multivesicular bodies upon fertilization, our findings suggest that a similar pathway also mediates paternal mitochondrial elimination in other flagellated sperm-producing organisms.

Organisms as diverse as plants, fungi, protists, and animals differ greatly in external morphology, internal structure, and physiology. However, when it comes to sexual mode of reproduction, they share multiple similar cellular and molecular patterns, such as meiosis, merger of the gametes, and a uniparental mode of mitochondrial inheritance[1–9]. In many organisms, including mammals and *Drosophila*, the union of the gametes during fertilization occurs between a pair of highly distinct cells, the elongated male sperm, and the spherical female egg, each contributing half of the parental genomic material to the zygote[10–12]. In contrast, only one parent, generally the mother, passes on the mitochondria to the offspring[8,13,14]. Although this uniparental mode of inheritance is ubiquitous among eukaryotes, there

[1]Department of Molecular Genetics, Weizmann Institute of Science, Rehovot, Israel. [2]Department of Life Sciences Core Facilities, Weizmann Institute of Science, Rehovot, Israel. [3]Department of Chemical Research Support, Weizmann Institute of Science, Rehovot, Israel. [4]de Botton Institute for Protein Profiling, The Nancy and Stephen Grand Israel National Center for Personalised Medicine, Weizmann Institute of Science, Rehovot, Israel. [5]Present address: Department of Microbiology and Molecular Genetics, Faculty of Medicine, The Hebrew University of Jerusalem, Jerusalem, Israel. ✉e-mail: eli.arama@weizmann.ac.il

are still abundant gaps in our knowledge concerning the molecular mechanisms underlying maternal mitochondrial inheritance, the fate of the paternal mitochondria after fertilization, and the rationale behind this highly conserved phenomenon[8,9,13–15]. These ambiguities may be attributed to various factors, including the widely held misconception that upon fertilization, sperm cells discard both the flagellum and the mitochondria, and that only the sperm head enters the egg[16]. In addition, it is difficult to exclude technical artefacts and false positive or negative findings due to assay sensitivity limits, such as assays for detection of scarce paternal mitochondrial (mt)DNA copies among the vast copy number of maternal mtDNA molecules[17]. Moreover, such assays may be highly misleading, as in some organisms, including *Drosophila* and perhaps also in humans, elimination of the paternal mtDNA has been reported to be uncoupled from, and already occurring prior to, degradation of the paternal vacuolar mitochondria (i.e., mtDNA-free mitochondria)[18–24]. The anatomical divergence in male gametes and their mitochondria among different organisms is another impeding factor, as this often hinders cross-study comparisons.

Two general theories have emerged to explain the mechanisms underlying paternal mitochondrial elimination (PME) after fertilization: A passive model of simple dilution of the paternal mitochondria by the excess copy number of egg mitochondria; and an active model, in which egg-derived mechanisms specifically target the paternal mitochondria for degradation. Both models have their drawbacks. The passive model is primarily based on two studies using mainly, albeit not exclusively, interspecific mouse crosses[21,25]. This experimental setup could lead to paternal mitochondrial leakage in the progeny, as has been previously documented with murine and bovine hybrids, presumably due to failure of the egg to recognize and target the paternal mitochondria[26–28]. Another reason to be cautious about this model is that proteins which may imply involvement of active mechanisms in mediating PME, were shown to decorate the paternal mitochondria after fertilization in mice and other mammals[21,29–31]. Most relevant are proteins involved in recycling of damaged mitochondria by selective autophagy (mitophagy), including ubiquitin, p62, and Atg8/LC3. Genetic studies in the roundworm *Caenorhabditis elegans*, which constitute the basis for the active model, suggest that PME is mediated by mechanisms reminiscent of mitophagy[18,30,32–37]. However, as opposed to flagellated sperm in mammals, in which the mitochondria undergo extensive structural organization and become part of the flagellum midpiece[6,38], the *C. elegans* amoeboid sperm lacks a flagellum and contains mitochondria of simple morphology, which can be accommodated by mitophagy[39].

Unlike *C. elegans*, *Drosophila melanogaster* flies produce unusually long sperm, consisting of an approximately 10 μm long needle-shaped nucleus and a ~1.8 mm long flagellum (~35 times the length of human sperm). The flagellum is mainly occupied by two major organelles: a single cylindrical mitochondrion, termed the mitochondrial derivative (MD), which extends from the posterior edge of the nucleus and along the entire length of the flagellum, and an intimately associated axoneme, a microtubule cytoskeleton-based structure which generates the propulsive force for sperm cell movement. The extraordinary size of the MD in *Drosophila*, which completely penetrates the egg, facilitates high spatiotemporal resolution of the PME process, with possible implications for PME in mammals[5,40,41]. Indeed, we previously demonstrated that in *Drosophila*, PME is an active process mediated by two types of egg-derived vesicles: multivesicular bodies (MVBs) of the endocytic pathway, which are the first to engage and target large segments of the MD for degradation; and autophagosomes of the autophagy pathway, that recycle small remnants of the degraded MD[42]. However, the pathway by which the MVBs recognize the MD and mediate PME remains unknown.

Although originating in the endocytic pathway and expressing late endosomal markers, such as Rab7, the MVBs that associate with the MD also display Atg8, a ubiquitin-like protein of the autophagy system, on their limiting membrane[42]. In recent years, several related noncanonical autophagy pathways, which mediate conjugation of Atg8 to single membranes (CASM), have emerged. As opposed to the double membrane autophagosomes that mediate classical autophagy, CASM pathways commonly utilize a subset of the autophagy machinery for Atg8 lipidation onto single-membrane vesicles[43–47]. The most extensively investigated CASM pathway is LC3-associated phagocytosis (LAP), which is involved in the clearance of different extracellular cargos by phagocytosis, including a variety of pathogenic microbes, but also dying cells, axon debris, soluble ligands, and protein aggregates, in different tissues and cell types[48–60]. LAP is activated upon engagement of extracellular cargos with specific surface receptors, mainly but not exclusively Toll-like receptors (TLRs), triggering inward invagination of the plasma membrane and engulfment of the cargo into a single-membrane phagosome (also termed LAPosome)[49,59]. A LAP-specific class III PI(3)K complex (PI3KC3; also known as complex III), composed of at least five proteins (Rubicon, Uvrag, Pi3k/Vps34, Atg6/Beclin1, and Vps15)[61,62], is recruited to the LAPosome membrane, where it generates the phospholipid phosphatidylinositol-3-phosphate (PtdIns(3)P). Consequently, via a yet unknown mechanism, PtdIns(3)P recruits the Atg8/LC3 conjugating system, which in turn promotes recruitment of Atg8/LC3 to the LAPosomal membrane in a manner that requires the NADPH oxidase-2 (Nox2)-derived reactive oxygen species (ROS). Atg8/LC3 lipidation then facilitates fusion of the LAPosomes with lysosomes, promoting cargo digestion[46,51,52,61,63,64].

Here, to uncover the sperm MD cellular degradative pathway mediated by the MVBs, we isolated these vesicles from early fertilized *Drosophila* eggs, using a subcellular fractionation procedure. Mass spectrometry (MS) analysis of the MVB-enriched fractions, followed by pathway enrichment analysis of the protein composition data, suggested the involvement of an innate immunity pathway with prevailing elements of phagocytosis. This, and the observation that Atg8 is expressed on the limiting membrane of the MVBs that associate with the MD[42], prompted us to explore the idea that the MVBs might mediate PME by activating a LAP-like CASM pathway. We demonstrate that mutants and RNAi knockdown of the LAP-specific PI3KC3 component, Rubicon, as well as knockdown of three other main components of the complex, all significantly attenuate degradation of the MD. Maternal expression of a *rubicon-eGFP* rescue transgene, and of tdTomato-tagged endogenous and transgenic Rubicon in the early fertilized egg, revealed the presence of multiple MVBs expressing Rubicon mainly on their limiting membrane. We then show that immediately after fertilization, many of the Rubicon-positive MVBs engage the sperm flagellum, creating extended vesicular sheaths around large flagellar segments. The flagellum vesicular sheaths (FVS) mediate rapid degradation of the sperm plasma membrane in a Rubicon-dependent manner, exposing the MD and axoneme to the luminal contents of these vesicular coats. Other LAP pathway events, such as Rubicon-dependent production of PtdIns(3)P and association of ROS with the sperm flagellum, also unfold during this process. Subsequently, Atg8 is recruited to the FVS in a Rubicon- and Atg7-dependent manner, ultimately leading to degradation of the MD, but not the axoneme, within strings of large Atg8 positive vesicles that originate in the FVS and contain lysosomal contents. These findings indicate that the egg activates a LAP-like pathway to target the MD for degradation. Given previous observations of MVBs in the vicinity of the paternal mitochondria in both bovine and hamster early embryos[27,65], as well as scattered indications of the presence of LC3 decorated paternal mitochondria in early embryos of some mammalian organisms[21,29–31], these findings suggest that a similar pathway might also mediate PME in mammals.

## Results

### MVBs contain elements of innate immunity and phagocytosis

In a previous study, we reported that degradation of the paternal mitochondria after fertilization in *Drosophila* is an efficient and rapid process, such that within 1.5-h after egg laying (AEL), the entire -1.8 mm long mitochondrial derivative (MD) is eliminated[42]. To further optimize this system for live imaging and minimize photobleaching, we generated a transgenic line that produces sperm with red fluorescently labeled MD (red-MD). This transgene consists of a fusion of a mitochondrial targeting sequence (MTS) with the red fluorescent protein tdTomato, which is both one of the brightest and most resistant to photobleaching among the DsRed derivatives[66]. Monitoring PME in eggs laid by wild-type (WT) females and fertilized by males producing red-MD sperm, revealed that this process is even faster, and that the MD is eliminated within only 1-hour AEL (Supplementary Movie 1). Note that time zero AEL is not equivalent to time zero after fertilization, as the latter occurs prior to egg laying, inside the female reproductive tract.

A key step in PME is the association of egg-derived MVBs with the sperm flagellum, an event that gradually and asynchronously culminates in the degradation of large MD segments (illustrated in Fig. 1a; Supplementary Fig. 1a–c)[42]. Accordingly, inactivation of genes required for proper biogenesis of the intraluminal vesicles (ILVs), the functional units of the MVBs, strongly attenuate PME[42]. Reasoning that the MVBs trigger a cellular degradation pathway to eliminate the MD, we set out to isolate the MVBs from early fertilized eggs and determine their proteome by MS. To label the MVBs, we generated a transgenic fly line containing an eGFP-tagged human CD63 (hCD63) gene, a member of the tetraspanin superfamily and a general marker of MVBs in human and *Drosophila*, which mainly localizes to the ILV compartment of these vesicles[67]. This transgene is expressed under the control of the *UASp* regulatory region, which allows for efficient expression of transgenes in the female germline by the maternal driver *mata-GAL4*. Note that some of the maternal transgenes were placed under *UASz*, an equivalent regulatory region[68,69]. Eggs laid by females maternally expressing the *eGFP-hCD63* transgene and fertilized by males producing red-MD sperm displayed multiple eGFP-hCD63 labeled vesicles, including vesicles associated with large segments of the MD (Fig. 1b).

To isolate the MVBs, extracts prepared from eGFP-hCD63-expressing early fertilized eggs were subjected to a modified OptiPrep ™ density gradient procedure of subcellular fractionation, and the MVB-enriched fractions were detected using Western blotting with anti-GFP antibodies (Supplementary Fig. 1d). The enrichment of largely intact MVBs and their ILVs in the peak fractions was confirmed by transmission electron microscopy (TEM; Supplementary Fig. 1e), and the composition of the proteins in these fractions was revealed by MS (Supplementary Data 1). Pathway enrichment analysis of the data revealed six major pathways, five of which were somewhat trivial or unrelated directly to cellular degradation processes (i.e., mitochondria, metabolism, endomembrane system, glycosylation, and transport), but one pathway, innate immunity with elements of phagocytosis, seemed particularly intriguing, as it implied possible involvement of a targeted cellular destruction pathway in PME (Supplementary Fig. 1f).

### Components of the LAP-specific PI3KC3 are required for PME

In the past decade, an unconventional phagocytosis pathway termed LC3-associated phagocytosis (LAP) has emerged as an important defense pathway against invading microbes. LAP combines the pathways of endocytosis and autophagy, mediating the delivery of pathogens and other extracellular elements to lysosomes by phagocytosis[51,52,57–59,63,64]. The finding that the MVBs, which are known to originate in the endocytic pathway, contain elements of phagocytosis, as well as the observed association of Atg8/LC3 with the limiting membrane of the MVBs that engage the flagellum[42], raised the

hypothesis that the MVBs mediate PME by activating a LAP-like pathway (Fig. 1c). Conventional autophagy and LAP are functionally and mechanistically distinct, as the autophagosome is a double-membrane structure, while the LAPosome, the characteristic vesicle mediating LAP phagocytosis, is a single membrane structure[49]. Furthermore, although LAP involves a subset of the autophagic machinery, it does not require the very first autophagy-specific complex. Also known as the pre-initiation complex, the first autophagy complex translocates to autophagy initiation sites and regulates the recruitment of the next autophagy complex[57,60,70,71]. Moreover, as opposed to the PI3KC3s complexes I and II that promote autophagy, the PI3KC3 complex III that is essential for LAP, contains a specific protein, Rubicon, which functions to inhibit autophagy (Fig. 1d)[61,62,72–75]. Significantly, MVBs are also vesicles with a single limiting membrane, and as in LAP, inactivation of several components of the autophagy pre-initiation complex, including Atg1/Ulk1, Fip200, and Atg13, had either weak or no effect on PME[42,76]. In contrast, downregulation of *uvrag*, which is a component of the PI3KC3 complexes II and III, significantly attenuates PME[42].

Given the anatomical and molecular similarities between the LAP and PME events, we set out to explore the possibility that a LAP-like pathway underlies PME in fertilized *Drosophila* eggs. For this, we first asked whether Rubicon, the main protein that distinguishes between the LAP-specific and the autophagy-specific PI3KC3s, is involved in this process[57,61,62,77]. Using CRISPR/Cas9, we generated four *rubicon* mutant alleles, all containing small frameshifting deletions at the beginning of the second exon, leading to premature stop codons and putative truncated proteins that lack the highly conserved RUN domain and the Rubicon homology (RH) domain (Supplementary Fig. 2a). At the RNA level, two *rubicon* mutant alleles, *rubicon^{A10}* and *rubicon^{A13}*, display a significant reduction in the expression levels of maternal *rubicon* mRNA, whereas a third mutant allele, *rubicon^{A196}*, containing a deletion encompassing the first splice-acceptor site, generates a truncated mRNA lacking part of the translated 5′ region (Supplementary Fig. 2b).

To test for possible effects of loss of Rubicon on PME, females homozygous for each of the four *rubicon* mutant alleles were crossed with males producing red-MD sperm. To ensure that effects on PME are not a consequence of failed embryonic development, we monitored normal progression in development of the early fertilized eggs through the process of blastoderm cellularization and omitted from our analysis the early fertilized eggs that failed to develop. Critically, all *rubicon* mutant fertilized eggs displayed significant attenuation in the degradation of the MD, as revealed by the persistence of the red-MD in the developing mutant fertilized eggs, even at 3 h AEL (Fig. 1e, f and Supplementary Fig. 2c–i). Furthermore, we generated a *rubicon-eGFP* transgene, which upon maternal expression restores degradation of the MD in *rubicon* mutant eggs (Fig. 1e, f and Supplementary Fig. 2d, e).

Flies homozygous for the *rubicon* mutant alleles are viable, but display varying levels of female and male fertility. Whereas homozygous females for the four *rubicon* mutant alleles show a significant reduction in fertility, with *rubicon^{A13}* and *rubicon^{A196}* causing the most dramatic effect, a milder but still significant reduction in male fertility is only detected in flies homozygous for these two alleles (Supplementary Fig. 3a, b). Accordingly, maternal but not male germline knockdown of *rubicon*, leads to a significant reduction in female fertility (albeit the effect is less pronounced than in the mutants) with no change in male fertility, respectively (Supplementary Fig. 3c, d). However, whereas these findings indicate that maternal *rubicon* is required for normal development, this effect likely occurs at post-embryonic cellularization stages, as both maternal *rubicon* mutants and knockdown progressed to the embryonic cellularization stage with similar kinetics as their respective control early fertilized eggs (Supplementary Fig. 3e, f). In conclusion, these findings suggest that PME and maybe other maternal Rubicon-dependent processes, are highly important for later organismal development.

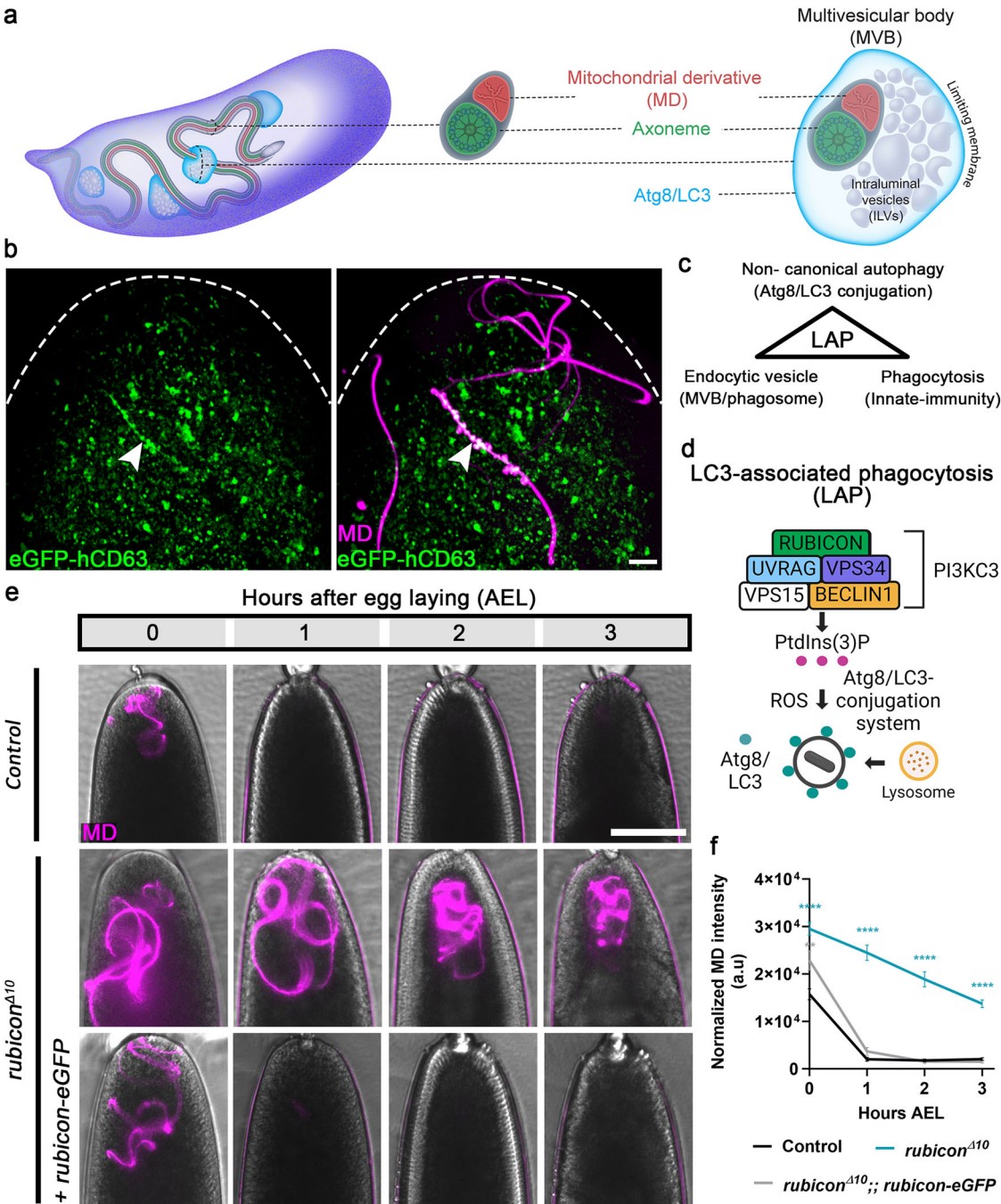

**Fig. 1 | Rubicon is required for PME after fertilization. a** Illustration of the onset of PME in *Drosophila* based on[42]. Left, an early fertilized egg. Egg-derived MVBs associate with the 1.8 mm long sperm flagellum. Middle, a cross-section through the sperm flagellum illustrating a single cylindrical MD (red) and an intimately associated axoneme (green), both coated by a common plasma membrane (gray) and extend along the entire length of the flagellum. Right, enlargement of an MVB (shown in a cross-section displaying the ILVs and the encapsulating limiting membrane) that encapsulates the flagellum. Atg8/LC3 (cyan) is loaded on the MVB limiting membrane. **b** A representative confocal image of the anterior (top) region of an egg maternally expressing the MVB transgenic construct *UASp-hCD63-eGFP* (green), and fertilized by a coiled red-MD sperm cell (magenta; *dj-(MTS)tdTomato*). MVBs that coat large flagellar segments are in white (arrowhead). Scale bar, 10 μm. **c** Schematic view of three main features shared by the egg MVBs and the LAP pathway. **d** Schematic view of the LAP pathway as defined in mammalian cells

(created with BioRender.com). **e** Live imaging of the anterior regions of early fertilized eggs laid by females of the indicated genotypes and fertilized by red-MD sperm. Early fertilized eggs carrying a single copy of the *rubicon* gRNA served as control. Representative time-lapse images of each fertilized egg were taken at 0, 1, 2, and 3 h AEL. Brightfield and fluorescence channel image overlay reveals the egg (gray) and the red-MD sperm flagellum (magenta), respectively. Scale bar, 100 μm. **f** Quantifications of normalized red-MD fluorescence intensities [arbitrary units (a.u.)] in fertilized eggs represented in **e**. Error bars indicate standard error of the mean (SEM). The respective numbers of scored early fertilized eggs (n) laid by females of the control, *rubicon*$^{Δ10}$, and *rubicon*$^{Δ10}$; *rubicon-eGFP* genotypes are 61, 32, and 34. **$P < 0.01$ and **$P < 0.0001$. Two-way repeated measures ANOVA, followed by Dunnett's multiple comparisons test. The *P* values are as follows: control vs. *rubicon*$^{Δ10}$, $P < 0.0001$ (all tested hours AEL); control vs. *rubicon*$^{Δ10}$; *rubicon-eGFP*, $P = 0.0070$ (0 h AEL). Source data are provided as a Source Data file.

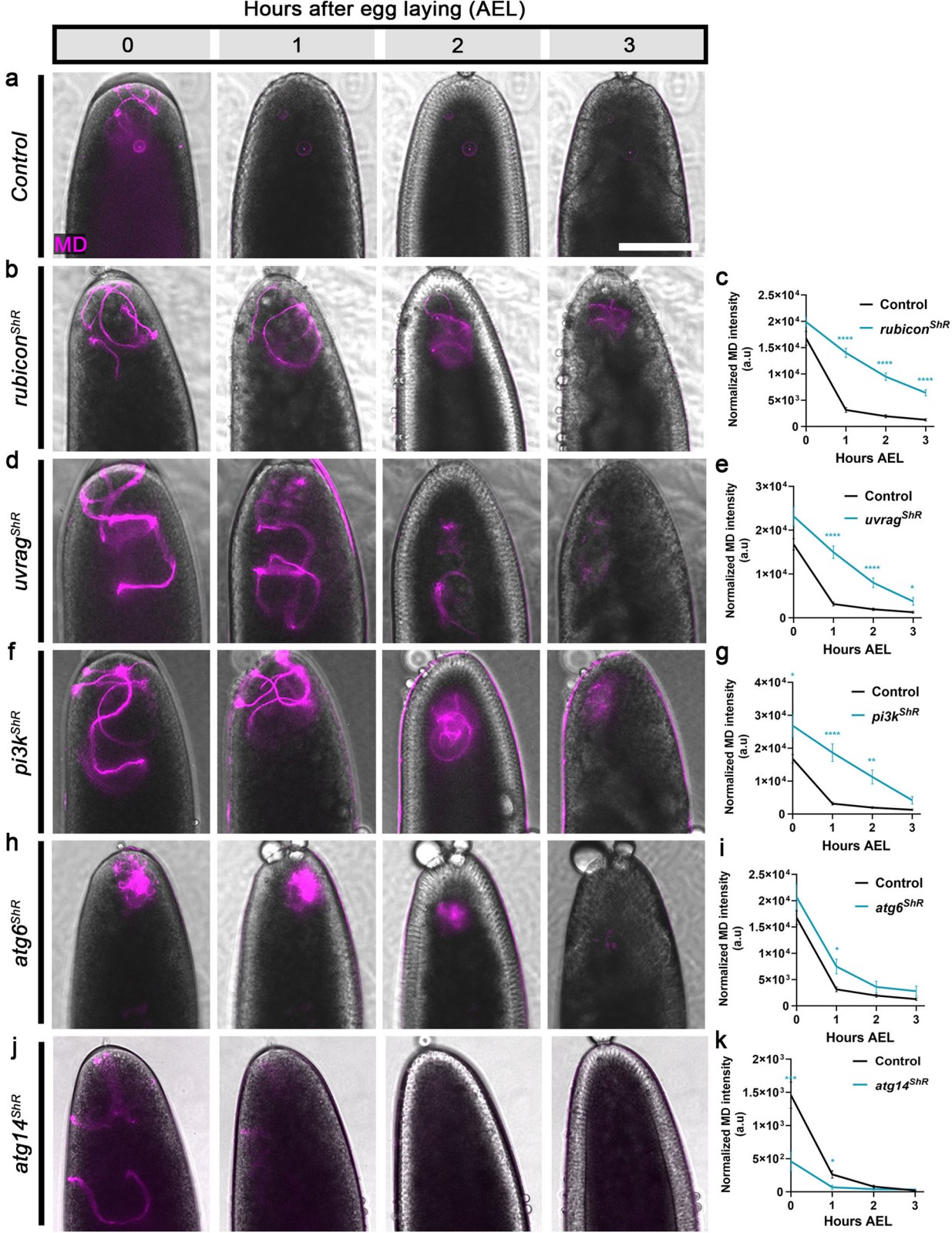

In addition to Rubicon, the LAP-specific PI3KC3 protein complex also contains Uvrag, Pi3k/Vps34, and Atg6/Beclin1 (Fig. 1d). Consistent with our previous report[42], knocking down *uvrag* by maternal expression of specific shRNA transgenes (ShR), significantly attenuated PME also in the current setup (Fig. 2d, e). Likewise, knockdowns of *pi3k* and *atg6* attenuated PME, although the effect of *atg6* knockdown was less pronounced than that of the

*uvrag*, *pi3k* and *rubicon* knockdowns (Fig. 2a–i and Supplementary Fig. 4a–d). In contrast, no effect on PME was detected upon knockdown of *atg14*, a key component of the autophagy-specific PI3KC3 (complex I)[78,79] (Fig. 2j, k and Supplementary Fig. 4e). Taken together, these findings suggest that a LAP-specific PI3KC3 is assembled in fertilized eggs and mediates degradation of the MD.

**Fig. 2 | Main members of the LAP-specific, but not the autophagy-specific PI3KC3 complex are required for PME. a, b, d, f, h, j** PME kinetic assays in control (expressing the maternal driver only) (**a**); and shRNA (ShR) mediated maternal knockdown in early fertilized eggs of *rubicon* (**b**), *uvrag* (**d**), *pi3k* (**f**), *atg6* (**h**), and *atg14* (**j**). A live imaging assay performed and presented as in Fig. 1e. Scale bar, 100 µm. **c, e, g, i, k** Quantifications of normalized red-MD fluorescence intensities in the fertilized eggs [magenta; arbitrary units (a.u.)] of the genotypes indicated on the left. Error bars indicate SEM. The respective numbers of scored fertilized eggs (n) laid by females of the control, *rubicon^ShR^* (**c**), *uvrag^ShR^* (**e**), *pi3k^ShR^* (**g**), and *atg6^ShR^*

(**h**), are 68, 60, 24, 22, and 24. For *atg14* knockdown and its control, 30 and 51 fertilized eggs were scored, respectively (**k**). Note that *atg14* has its own control group, as this experiment was conducted at a later time. *$P < 0.05$, **$P < 0.01$, ***$P < 0.001$, and ****$P < 0.0001$. Two-way repeated measures ANOVA, followed by Šidák's multiple comparisons test. *P* values in **c**, $P < 0.0001$ (1–3 h AEL); **e**, $P < 0.0001$ (1–2 h AEL), $P = 0.0446$ (3 h AEL); **g**, $P = 0.0436$ (0 h AEL), $P < 0.0001$ (1 h AEL), $P = 0.0011$ (2 h AEL); **i**, $P = 0.0244$ (1 h AEL); **k**, $P = 0.0005$, (0 h AEL), $P = 0.0122$ (1 h AEL). MD mitochondrial derivative. Source data are provided as a Source Data file.

### Rubicon vesicles associate with the sperm and degrade the MD

To examine whether the LAP-specific PI3KC3 assembles on egg vesicles, live fertilized eggs, laid by females maternally expressing the *rubicon-eGFP* rescuing transgene, and fertilized by males producing red-MD sperm, were collected 0–15 min (min) AEL and imaged for 45 min. Remarkably, Rubicon-eGFP specifically labeled three elements in the egg; numerous vesicles scattered throughout the ooplasm, large sperm flagellar segments, and large vesicles, 1–1.5 µm in diameter on average, that bud from, but remaining in the vicinity of, the flagellum, and contain MD pieces and debris (Fig. 3a–c and Supplementary Movie 2). Interestingly, within the large vesicles that bud from the flagellum, Rubicon-eGFP primarily localizes to the limiting membrane, reminiscent of the assembly of the LAP-specific PI3KC3 on the LAPosome limiting membrane (Fig. 3c). It is important to note that, consistent with the requirement of PI3KC3 for PME, the appearance of Rubicon on the flagellum always precedes the elimination of the MD fluorescent signal (Supplementary Movie 2).

To confirm that the expression dynamics and localization pattern of the *UASz*-based *rubicon-eGFP* transgene in early fertilized eggs reliably reflect those of the endogenous *rubicon* locus and encoded protein, we used CRISPR/Cas9 to tag the endogenous *rubicon* gene with tdTomato. Live imaging of eggs laid by Rubicon-tdTomato females and fertilized by males producing green-MD sperm (flies expressing the DJ-GFP transgenic protein under the regulation the *dj* regulatory regions[42,80]), revealed that endogenous Rubicon is expressed in essentially identical spatiotemporal patterns as those of the *rubicon-eGFP* transgene (Fig. 3d).

To visualize the anatomical details of the PME process in high resolution, we adopted an expansion microscopy (ExM) protocol to visualize Rubicon and the sperm MD in super resolution[81,82]. This protocol, which expands the samples by a factor of ~4 in all three dimensions, was applied to fertilized eggs laid by females maternally expressing a *rubicon-tdTomato* transgene, and fertilized by males producing green-MD sperm. Importantly, as shown in Fig. 3e, the MD resides within Rubicon positive extended vesicular sheaths. Moreover, segmentation of the surfaces of the expanded fertilized egg at the area of the sperm flagellum and subsequent 3D surface rendering, clearly revealed numerous Rubicon-positive vesicles densely arranged one next to the other, generating flagellum vesicular sheaths (FVS) that completely enwrap the degrading MD (Fig. 3f and Supplementary Movie 3).

### Breakdown of the sperm membrane precedes MD degradation

In many insects, the fertilizing sperm cell enters the egg through an opening in the chorion-containing eggshell termed the micropyle[83]. In *Drosophila*, the sperm cell enters through the micropyle while coated by its own plasma membrane[40,84,85]. This means that to degrade the MD, the sperm plasma membrane is likely to break down soon after fertilization, before engagement of the egg vesicles with the MD, or that the egg vesicles engage and form the FVS over the intact sperm plasma membrane. To determine which of these possibilities prevails, we first generated a transgenic marker for the sperm plasma membrane, composed of the mouse CD8 transmembrane domain fused to a Venus fluorescent protein, and expressed under the control of the late

spermatid promoter and regulatory sequences of the *dj* gene. We then applied the ExM protocol to eggs maternally expressing the *rubicon-tdTomato* transgene and fertilized by sperm expressing the CD8-Venus marker of the plasma membrane. As a control for a non-degradable sperm organelle, the early fertilized eggs were also stained with the anti-polyglycylated Tubulin antibody to label the axoneme[42,86,87]. Significantly, optical sections through the sperm flagellum revealed the sperm plasma membrane breaking down over an intact axoneme, and both are entrapped within Rubicon-positive FVS (upper panels in Fig. 4a). Furthermore, surface segmentation and subsequent 3D rendering of these fertilized eggs, readily indicated the dense arrangement of the Rubicon vesicles that form FVS around the sperm plasma membrane (lower panels in Fig. 4a).

We then examined whether, similar to MD degradation, sperm plasma membrane breakdown is also mediated by Rubicon. Monitoring maternal *rubicon* knockdown eggs fertilized by sperm expressing the CD8-Venus plasma membrane marker, and stained to visualize the axoneme, detected a highly significant attenuation in sperm plasma membrane breakdown upon maternal *rubicon* knockdown, as compared with counterpart control eggs (Fig. 4b–d). Note that the examined early fertilized eggs were all at a similar stage (syncytial early fertilized eggs up to nuclear division cycle two).

Given that the MD is still coated by the sperm plasma membrane after fertilization, the findings that the breakdown of the sperm plasma membrane and degradation of the MD both occur within the FVS, raise the hypothesis that in order to degrade the MD, the FVS first break down the sperm plasma membrane. To test this possibility, we used live imaging to monitor sperm plasma membrane breakdown kinetics after fertilization in eggs maternally expressing the *rubicon-tdTomato* transgene and fertilized by sperm expressing the plasma membrane CD8-Venus marker. As shown in Supplementary Movie 4 and representative still images in Fig. 4e, complete breakdown of the sperm plasma membrane already occurs within 8–10 min AEL, which is much faster than the time it takes to degrade the MD (40–60 min AEL), consistent with a timeline where sperm plasma membrane breakdown precedes MD degradation.

Since the axoneme and the MD are intimately associated, an intriguing question is how the axoneme, which is also initially embedded within the FVS, is spared from degradation. Unexpectedly, examining ExM images of the axoneme in eggs maternally expressing the *rubicon-tdTomato* transgene and fertilized by the green-MD sperm, revealed that whereas both the MD and the axoneme are enwrapped by the FVS, they occupy different compartments, such that the axoneme resides in a narrow vesicular tube generated by the FVS (Fig. 4f). This finding suggests that the FVS actively separate between the MD and the axoneme, likely protecting the axoneme from sharing the same fate as the MD and the sperm plasma membrane.

### Rubicon localizes on MVBs

Given the unique FVS formation generated by the Rubicon vesicles, we asked whether these vesicles are the MVBs detected by TEM (Supplementary Fig. 1a–c)[42]. For this, we generated flies maternally co-expressing the Rubicon-eGFP and the MVB marker protein hCD63 fused to tdTomato (i.e., *UASz-hCD63-tdTomato*). Eggs laid by these

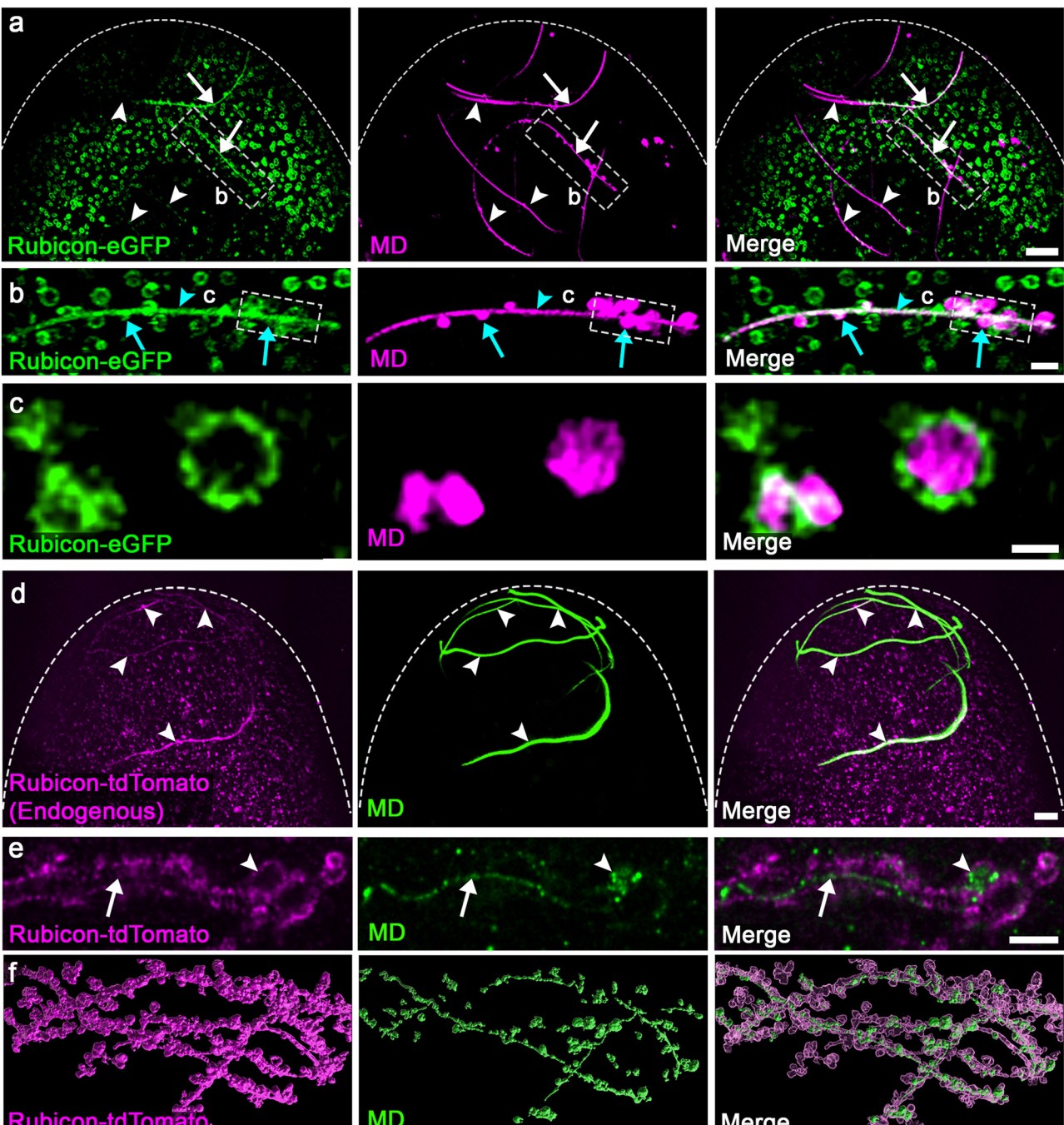

**Fig. 3 | Rubicon is localized to the limiting membrane of multiple egg vesicles which generate extended vesicular sheaths that enwrap the sperm flagellum. a, d** Representative confocal images of the anterior (top) region of early fertilized eggs. **a** An egg maternally expressing the *UASz-rubicon-eGFP* transgene (green) and fertilized by a red-MD sperm cell (magenta). Numerous egg vesicles display Rubicon-eGFP, many of which associate with large segments of the sperm flagellum (arrows), whereas other flagellar segments show only a few vesicles or are still free of vesicles (arrowheads), due to the asynchronous manner of the PME process along the sperm flagellum. Scale bar, 10 µm. **b** Magnification of the area outlined by a dashed rectangle in (**a**). Two types of Rubicon positive vesicles are detected: Numerous small vesicles that densely associate with flagellar segments in a rather unified order (arrowhead); Large Rubicon positive vesicles (green) that appear to bud from the flagellum, encapsulating MD pieces and debris (magenta; arrows). Scale bar, 2 µm. **c** Magnification of a projection of a few Z-sections in the area

outlined by a dashed rectangle in (**b**). Rubicon (green) remains localized on the limiting membrane of the large vesicles that contain MD fragments and bud off from the flagellum. Scale bar, 1 µm. **d** An egg expressing tdTomato-tagged endogenous Rubicon (magenta) and fertilized by green-MD sperm (green). Rubicon is localized on multiple egg vesicles, many of which associate with large flagellar segments (arrowheads). Scale bar, 10 µm. **e, f** Shown are super-resolution images of flagellar regions in eggs maternally expressing the *UASz-rubicon-tdTomato* transgene (magenta), fertilized by green-MD sperm (green; *dj-DJ-GFP*), and prepared for ExM. The images feature large MD segments that are readily enwrapped within Rubicon positive FVS (arrow), as well as large Rubicon positive vesicles that contain MD fragments and bud off from the flagellum (arrowhead). Note that the image in (**e**) presents a single non-computed optical section, while the image in (**f**) presents a 3D computer rendering of segmented surfaces of this early fertilized egg. MD, mitochondrial derivative.

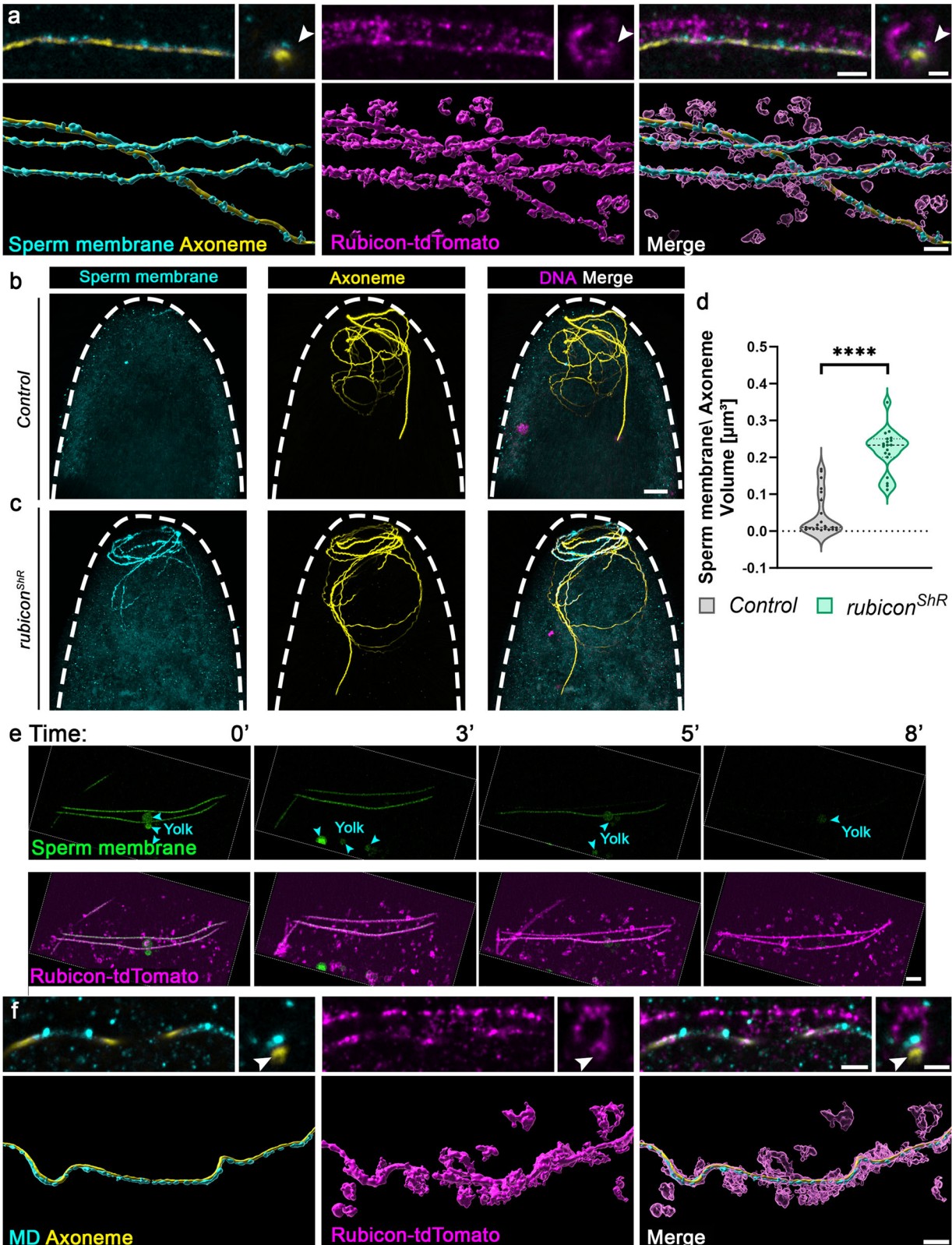

females and fertilized by WT males displayed numerous vesicles labeled by either Rubicon-eGFP, hCD63-tdTomato, or both (Fig. 5a). Whereas multiple vesicles labeled only by hCD63-tdTomato and located in the egg periphery were not associated with the MD, 85% of the Rubicon-eGFP positive vesicles were also positive for hCD63-tdTomato, including large vesicles associated with the flagellum (Fig. 5a–c; note that the large flagellar segments can be detected

merely by virtue of being enwrapped by Rubicon positive FVS). Closer examination of the dually labeled vesicles revealed that the two tagged proteins occupy different compartments in these vesicles: Rubicon-eGFP is primarily located at the limiting membrane of the MVBs, while hCD63-tdTomato resides in the ILV compartment (Fig. 5b). Interestingly, traces of hCD63-tdTomato are detected along the flagellar segments within the Rubicon-eGFP FVS (Fig. 5a), which is consistent with

**Fig. 4 | Sperm plasma membrane breakdown after fertilization occurs within the FVS at a much faster rate than that of MD degradation. a, f** Super-resolution images of flagellar regions in an egg maternally expressing the *UASz-rubicon-tdTomato* transgenic construct (magenta; immunostained with anti-RFP antibody), fertilized by sperm cells expressing the *dj-CD8-Venus* transgene that labels the plasma membrane (**a**) or the *dj-DJ-GFP* transgene that labels the MD (**f**, a green-MD sperm cell), both immunostained with an anti-GFP antibody (cyan), immunostained to visualize the axoneme (yellow), and prepared for ExM. The images at the top depict single optical sections before computing [scale bar (left), 1 μm; scale bar (right), 0.5 μm], while the images at the bottom depict 3D computer rendering of segmented surfaces in the early fertilized eggs (scale bar, 2 μm). Note the narrow vesicular tube within the FVS that confines the axoneme (arrowheads). **b, c** Representative confocal images of the anterior regions in early fertilized eggs laid by control (**b**) and maternal *rubicon* knockdown (**c**) mothers. The eggs were fertilized by sperm cells expressing the *dj-CD8-Venus* transgene (cyan; immunostained with anti-GFP antibody) and immunostained to visualize the axoneme

(yellow). DAPI stained the DNA (magenta). Scale bar, 20 μm. **d** The graphs depict staining volumes of the sperm plasma membrane relative to the axoneme volume in each fertilized egg, corresponding to the fertilized eggs in (**b,c**). The respective numbers of examined fertilized eggs (*n*) laid by control and *rubicon*^ShR^ females are 22 and 18. Two-tailed unpaired Student's *t* test. \*\*\*\**P* < 0.0001. Summary of the statistics reported in the violin plots: the center line represents the median of the frequency distribution of the data. Quartiles are represented by a dashed black line. Each dot corresponds to the relative sperm plasma membrane/Axoneme volume in a single fertilized egg. **e**, Live confocal imaging of the anterior region of an early fertilized egg laid by a female maternally expressing the *rubicon-tdTomato* transgene (magenta) and fertilized by a sperm cell expressing the CD8-Venus transgenic marker of the plasma membrane (green). Shown are representative images from Supplementary Movie 10 at the indicated times (min AEL). Scale bar, 3 μm. Yolk autofluorescence is indicated by arrowheads. MD mitochondrial derivative. Source data are provided as a Source Data file.

our ultrastructural studies suggesting that upon engagement with the flagellum, the MVBs release their ILVs in the vicinity of the MD (Supplementary Fig. 1c and[42]).

To test whether Rubicon might function in biogenesis and/or flagellar recruitment of the MVBs, we took advantage of the eGFP-hCD63 transgenic marker of the ILV compartment in MVBs. We monitored both the percentage of maternal *rubicon* knockdown early fertilized eggs that exhibit flagellar positive eGFP-hCD63, as well as the intensity of this fluorescent marker. Whereas ~95% of control early fertilized eggs display flagellar eGFP-hCD63, about 30% of the maternal *rubicon*-deficient eggs were completely devoid of flagellar eGFP-hCD63 (Supplementary Fig. 5a, b). Moreover, the fluorescence intensity of flagellar eGFP-hCD63 that is associated with the sperm MD was significantly reduced in the maternal *rubicon*-deficient eggs, as compared with control eggs (Supplementary Fig. 5a, c). Whereas these findings indicate that Rubicon is required for the proper function of the egg MVBs in the context of PME, we cannot differentiate between an effect on MVB biogenesis or on the recruitment of the MVBs to the FVS. Given that maternal inactivation of Uvrag and Atg7 resulted in deformed ILV formation[42], it is likely that *rubicon* knockdown might also affect MVB biogenesis in a similar manner. Collectively, these observations indicate that the egg contains numerous MVBs pre-loaded with Rubicon, which densely enwrap the sperm flagellum immediately after fertilization, generating extended FVS that mediate PME.

### Rubicon positive MVBs and the FVS exhibit PI3KC3 activity

Active Rubicon PI3KC3 facilitates sustained PtdIns(3)P production by Pi3k/Vps34, which is required for recruitment of Atg8/LC3 to LAPosomes (Fig. 1d). We therefore tested whether the Rubicon-positive MVBs are also associated with PtdIns(3)P. For this, we first maternally co-expressed the *rubicon-eGFP* transgene together with TagRFPt-2xFYVE, a genetically encoded reporter for PtdIns(3)P based on the PtdIns(3)P-binding peptide motif FYVE, fused to a red fluorescent protein[88,89]. Multiple PtdIns(3)P positive vesicles were detected by live imaging of eggs laid by these females and fertilized with WT sperm, while a subset of these vesicles were also positive for Rubicon-eGFP (Fig. 5d). Note that at least some of the vesicles labeled by PtdIns(3)P alone may correspond to other endosomes, such as late endosomes that enwrap yolk granules during the cellularization stage[90]. Significantly, the presence of flagellar PtdIns(3)P is detected in eggs maternally expressing the PtdIns(3)P reporter and fertilized by green-MD sperm, indicating that the FVS PI3KC3 is active (Fig. 5e and Supplementary Movie 5).

To examine whether the flagellar PtdIns(3)P is produced by the Rubicon PI3KC3, we first wanted to track the flagellum association kinetics of Rubicon and PtdIns(3)P by time-lapse confocal microscopy. However, already at 0 h AEL, which- as previously mentioned- is a later

timepoint than actual fertilization, both Rubicon and PtdIns(3)P were associated with the sperm flagellum, precluding the ability to determine which of these events comes first (Supplementary Movie 6). Nevertheless, monitoring the presence of flagellar PtdIns(3)P in maternal *rubicon* knockdown early fertilized eggs, revealed no flagellar PtdIns(3)P in 100% of the fertilized eggs, as compared with equivalent controls where flagellar PtdIns(3)P was present in ~50% of the early fertilized eggs (Fig. 5f, g). We therefore conclude that the Rubicon PI3KC3 on the FVS is indeed the complex that produces the observed flagellar PtdIns(3)P.

### ROS are involved in PME

In addition to PtdIns(3)P, recruitment of the Atg8/LC3 to the maturing LAPosome was reported to also require ROS production, which in phagocytes is generated by the NOX2 multiprotein complex[51,64,75,77,91,92]. WT eggs fertilized by red-MD sperm and labeled with the cell-permeant, fluorescent ROS indicator H2DCFDA[93,94], revealed high levels of ROS on the MD (Fig. 5h). Two NADPH oxidase (NOX) homologs, Nox and Duox, are present in *Drosophila*[95]. Quantitative reverse transcription PCR (RT-qPCR) analyses of early fertilized egg transcripts indicated that of the two, *duox* is essentially the sole maternally expressed gene (Supplementary Fig. 5d), in agreement with omics data in two public databases[96,97]. Duox is also enriched in the egg MVB proteome (Supplementary Data 1). Accordingly, knockdown of *duox*, but not *nox*, in eggs fertilized by red-MD sperm, moderately attenuated PME, suggesting that Duox might be responsible for a portion of the ROS required for PME (Supplementary Fig. 5e–i and Supplementary Fig. 4f, g). In addition, since mitochondria are known as a main source of ROS, we also tested if ROS is already present in the mature sperm flagellum prior to fertilization. Indeed, mature sperm in the seminal vesicle, the male sperm storage organ, stained with H2DCFDA, revealed pronounced ROS expression in the sperm flagellum (Supplementary Fig. 5j). Taken together, we conclude that the two mechanistic requirements for recruitment of Atg8/LC3 to the LAPosome, production of PtdIns(3)P and ROS, are also fulfilled during the PME process.

### Atg8/LC3 is recruited to the FVS and is required for PME

In a previous study, we showed that transgenes expressing fusions of the key autophagic protein Atg8 with fluorescent proteins are recruited to the vicinity of the MD after fertilization[42]. However, several questions remain unanswered regarding the significance of this association. First, we asked whether endogenous Atg8a is also recruited to the vicinity of the MD, and if so, where it is localized and at which stage. To address these questions, we immunostained ~50 WT eggs fertilized by the green-MD sperm and collected at 0–1-hour AEL, to visualize the MD (anti-GFP antibody), endogenous Atg8a (anti-Atg8a antibody), and the axoneme (anti-polyglycylated Tubulin antibody). Three

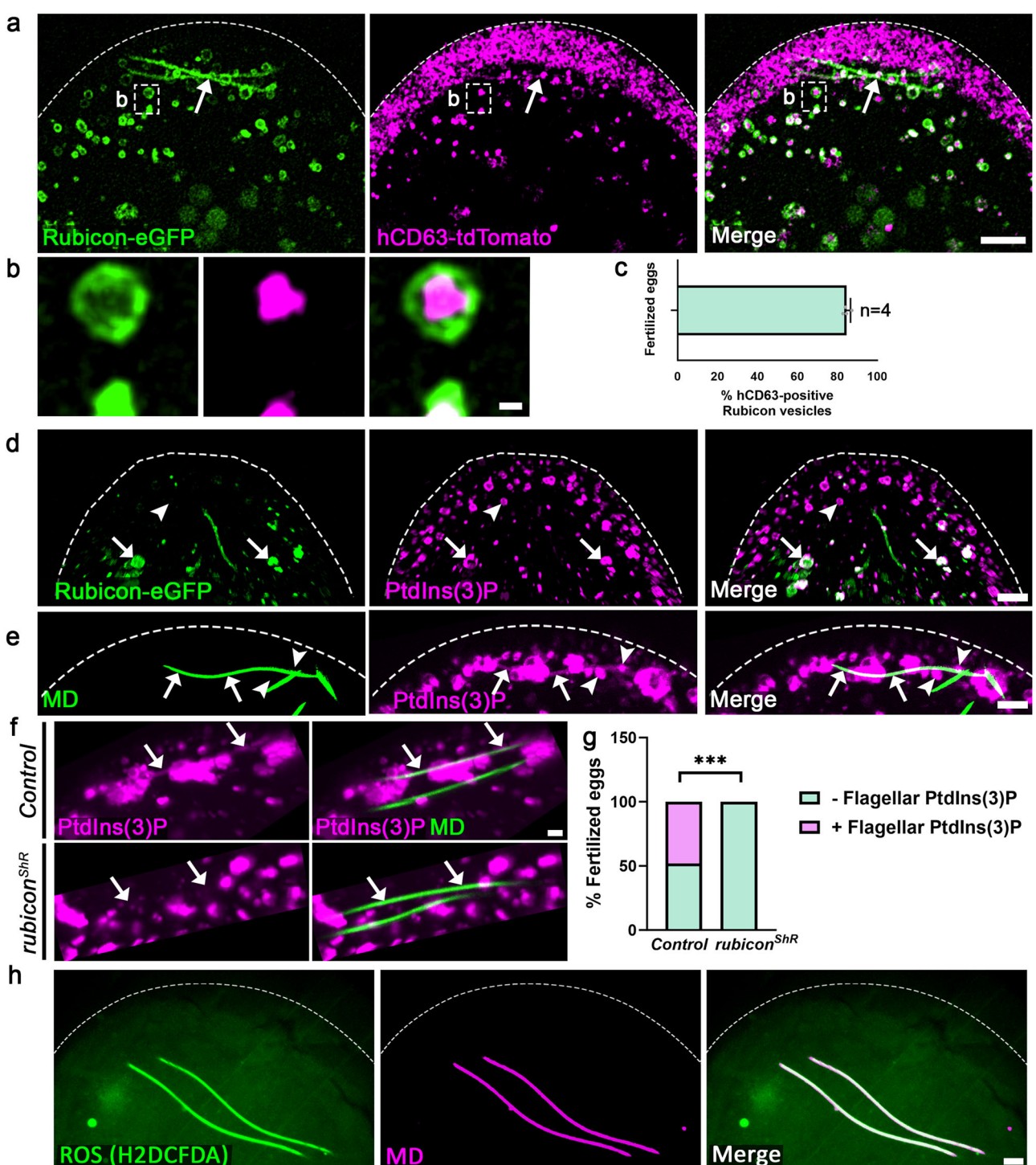

consecutive and distinct PME phases can be clearly detected, often within a single fertilized egg: (1) Intact MD segments that are Atg8a negative and usually still reside in the vicinity of parallel axonemal segments (Fig. 6a, I); (2) FVS that become positive for Atg8a, which often display strings of large Atg8a positive vesicles that encapsulate degrading MD fragments (Fig. 6a, IIa), resulting in MD-free axonemal segments (Fig. 6a, IIb); (3) MD-free axonemal segments displaying neither MD traces nor Atg8a positive FVS, indicative of completion of the PME process (Fig. 6a, III). Taken together, these observations indicate that endogenous Atg8a is recruited to the FVS at a relatively late PME stage, during which the MD is readily degraded.

Given the pronounced association of Atg8a with the strings of large vesicles that contains degrading MD fragments, we next asked whether Atg8a is required for efficient PME. However, maternal knockdown of *atg8a*, the major fly *atg8* gene, had no significant effect on PME kinetics (Supplementary Fig. 6b, c and Supplementary Fig. 4h). *Drosophila* has another closely related but distinct *atg8* gene, *atg8b*, although its expression is mainly restricted to the testis and is barely detected in other tissues[98–100]. Accordingly, maternal knockdown of *atg8b* also had no effect on PME kinetics (Supplementary Fig. 6d, e).

It was previously reported that *atg8b* was elevated in *Drosophila* eye imaginal discs homozygous for a hypomorphic *atg8a* mutant allele[101]. Likewise, RT-qPCR analysis of early fertilized eggs also revealed a significant increase in the level of *atg8b* transcripts upon maternal knockdown of *atg8a*, which was abrogated in maternal double knockdown of both *atg8a* and *atg8b* (Supplementary Fig. 4i, j).

**Fig. 5 | Rubicon-coated MVBs express active PI3KC3. a, b** A live imaging projection of the anterior region of an egg maternally expressing both a *UASz-rubicon-eGFP* (green) and the *UASz-hCD63-tdTomato* (magenta; MVBs) transgenic constructs, and fertilized by a WT (non-fluorescent) sperm cell. Magnification of the area outlined by a dashed rectangle in (**a**) is shown in (**b**). Rubicon positive vesicles and FVS also display hCD63-tdTomato (arrow in **a**). Scale bars in (**a**) 5 μm; (**b**) 0.5 μm. **c** Most (~85%) of the Rubicon-eGFP positive vesicles also display the MVB marker hCD63-tdTomato. Measure of center is the mean percentage. Error bar indicates standard deviation (SD). The pulled Pearson's Correlation Coefficient (PCC) is 0.3402292 (*P*-value < 2.2e−16). The pulled Manders' Colocalization Coefficient is 0.9984238. *n* = 4. **d, e, h** Representative confocal images of the anterior region of early fertilized eggs. **d** An egg maternally expressing both the *rubicon-eGFP* transgene (green) and the PtdIns(3)P reporter TagRFPt-2xFYVE (magenta), and fertilized by a WT (non-fluorescent) sperm cell. The egg contains multiple PtdIns(3)P positive vesicles (arrowhead), including large Rubicon-eGFP positive vesicles (arrows). Scale bar, 6 μm. **e** An egg maternally expressing the PtdIns(3)P reporter (magenta) and fertilized by green-MD sperm (green). In addition to the large PtdIns(3)P positive vesicles derived from the flagellum (arrowheads), flagellar PtdIns(3)P is also readily detected (arrows). Scale bar, 10 μm. **f** Representative confocal images of control (top) and maternal *rubicon* knockdown (bottom) early fertilized eggs at 0–15 min AEL, maternally expressing the PtdIns(3)P reporter (magenta) and fertilized by green-MD sperm (green). Arrows indicate flagellar areas. Scale bar, 2 μm. **g** Quantification of the number of early fertilized eggs (corresponding to the eggs in **f**) that display flagellar PtdIns(3)P. Two-sided Fisher's exact test. ***\*P* = 0.0002. The number of examined fertilized eggs (*n*) laid by females of both genotypes is 23. **h** A WT egg fertilized by red-MD sperm (magenta) and stained with the fluorescent ROS indicator H2DCFDA (green). Scale bar, 10 μm. MD mitochondrial derivative. Source data are provided as a Source Data file.

Importantly, monitoring early fertilized eggs with the double maternal knockdowns revealed a significant attenuation in PME, demonstrating that Atg8 is required for an efficient PME process (Fig. 6b, c).

To evaluate the time elapsed between coating of the flagellum by the Rubicon MVBs and recruitment of Atg8a to the FVS, we maternally expressed *rubicon-eGFP* and *mCherry-atg8a* transgenes, and used live imaging to monitor their relative spatiotemporal patterns of expression. Large Rubicon-eGFP positive FVS that are mCherry-Atg8a negative were readily detected at time 0 AEL, while recruitment of mCherry-Atg8a to the FVS occurred 6–12 min later (Fig. 6d–k and Supplementary Movie 7). Altogether, these findings are consistent with the LAP pathway sequence of events, in which loading of the Rubicon PI3KC3 on the FVS precedes recruitment of Atg8/LC3.

### Atg8/LC3 recruitment to the FVS requires Rubicon

To examine whether early loading of Rubicon PI3KC3 is also required for Atg8a association with the FVS, we next monitored the staining volume of Atg8a in 0–1-hour AEL fertilized eggs maternally expressing an shRNA transgene against *rubicon*. Maternal expression of an *atg7* shRNA transgene served as positive control, whereas down regulation of *atg8a* was used to validate the specificity of the anti-Atg8a antibody (Supplementary Fig. 4h, k). To avoid possible unrelated Atg8a positive vesicles, such as autophagosomes, we restricted the measurements to an area surrounding the axoneme. In addition, to ensure that the developmental stages of the early fertilized eggs were highly synchronized, we monitored syncytial early fertilized eggs up to nuclear division cycle six. The Atg8a staining volume in a 0–10 μm radius around the axoneme, corresponding to the FVS and its immediate surrounding, was significantly reduced upon down regulation of *rubicon*, *atg7*, and *atg8a*, as compared with counterpart WT eggs, indicating that Atg8a recruitment to the FVS requires both Rubicon and Atg7 (Fig. 7).

The observed reduced level of flagellar Atg8a in *rubicon* knockdown early fertilized eggs might be the consequence of inefficient Atg8a recruitment to the FVS and/or delayed recruitment. To determine which of the possibilities is correct, we used live imaging to monitor WT and *rubicon* knockdown eggs maternally expressing the *mCherry-atg8a* transgene and fertilized by the green-MD sperm. In an otherwise WT background, most of the flagellar segments recruit Atg8a and contain degraded MD within 10–20 min AEL, such that only a very few short MD segments are still detected at 50 min AEL, by virtue of their mCherry-Atg8a positive FVS (Supplementary Fig. 7a and Supplementary Movie 8). In contrast, eggs maternally expressing the *rubicon* shRNA transgene display a delayed appearance of mCherry-Atg8a on FVS (40–50 min AEL), which coated only a few short flagellar segments (Supplementary Fig. 7b and Supplementary Movie 9). These findings indicate that Rubicon is required for both fast and efficient accumulation of Atg8a on FVS.

### FVS-derived vesicles with MD fragments fuse with lysosomes

In the final stage of LAP, Atg8/LC3 is suggested to promote fusion of the LAPosome with lysosomes to accelerate degradation of the phagolysosomal cargo (Fig. 1d)[49,52,102]. To examine whether during PME lysosomes fuse with the FVS to promote efficient degradation of the MD, we monitored live eggs ubiquitously expressing the transgenic lysosomal marker GFP-LAMP1[103,104], and fertilized by the red-MD sperm. As shown in Supplementary Movie 10 and in Fig. 8a, multiple lysosomes are detected at the vicinity of the degrading MD, some of which appear to specifically co-localize with MD fragments. To examine whether the MD fragments correspond to the observed degrading MD fragments within the string of large Atg8a positive vesicles derived from the FVS (Fig. 6a), we imaged GFP-LAMP1-expressing eggs fertilized by WT sperm and stained to visualize LAMP1 (anti-GFP), Atg8a, and the axoneme. The lysosomal marker, GFP-LAMP1, was readily detected within the Atg8a positive large vesicles, indicating that degradation of the MD fragments within these vesicles is mediated by lysosomal hydrolases (Fig. 8b, c).

### The Atg5-Atg12-Atg16 complex is not required for PME

In conventional autophagy, Atg7 serves as an E1 ligase-like enzyme that activates two different ubiquitin-like proteins, Atg12 and Atg8/LC3[105,106]. Atg12 then undergoes conjugation to Atg5, and the Atg12-Atg5 complex interacts with Atg16L1 to form a large multimeric complex that facilitates efficient Atg8/LC3 lipidation[107]. Likewise, it has been shown that at least in some LAP paradigms, the Atg12-Atg5-Atg16L1 complex is required for Atg8/LC3 conjugation to the LAPosome membrane[49,108–110]. However, whereas maternal mutant and knockdown of *atg7* significantly attenuates PME (Supplementary Fig. 6f, g)[42], maternal knockdowns of *atg12*, *atg5*, and *atg16* had almost no effect on PME kinetics, indicating that this complex is dispensable for the LAP-like pathway underlying PME (Supplementary Fig. 4l–n and Supplementary Fig. 6h–m). It is noteworthy that similar results were also obtained with *Atg5* knockout early mouse embryos[21]. Given that Atg8/LC3 also decorates the mouse sperm mitochondria immediately after fertilization[21], it is plausible that a similar non-conventional autophagy pathway might also mediate PME in mammals.

## Discussion

In this work, we uncovered the pathway by which the sperm MD is eliminated after fertilization in *Drosophila*. We show that special MVBs, which are abundant in the early fertilized egg, engage the sperm flagellum immediately after fertilization, while triggering a LAP-like pathway to specifically target the MD for degradation. This highly efficient process takes about 20–40 min to complete, as revealed by the disappearance of transgenic fluorescent signals that label the MD, with an average MD degradation rate of 100 nm/min. According to the

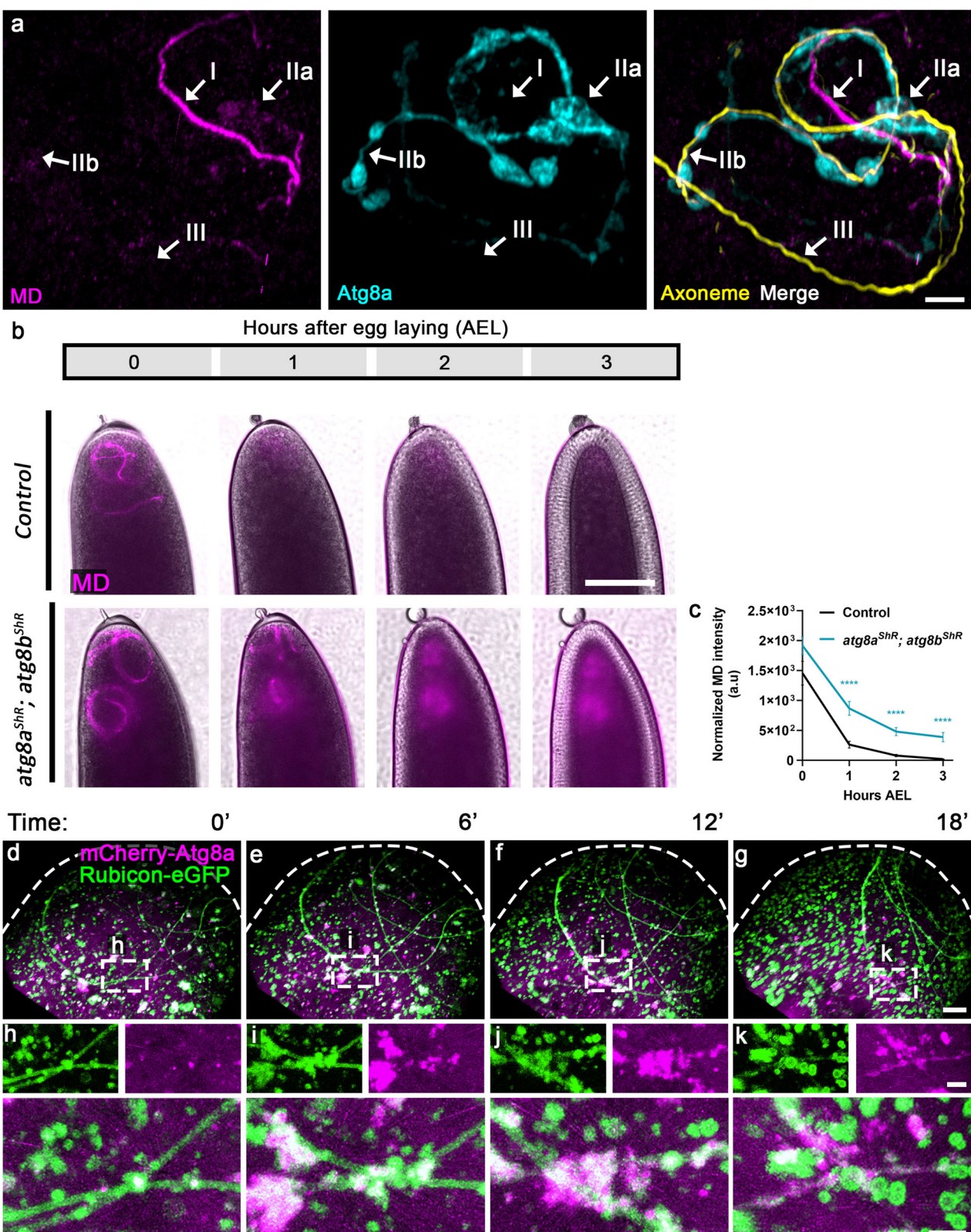

sequence of events we uncovered, numerous egg MVBs that are loaded with the Rubicon-containing PI3K complex (PI3KC3) on their limiting membrane approach and densely coat large segments of the penetrating sperm flagellum in an asynchronous manner (Fig. 8d, I). These MVBs wrap around the sperm flagellum plasma membrane, forming FVS within which the sperm plasma membrane breaks down and the MD is degraded (Fig. 8d, II–VI). At the molecular level, activated LAP-specific PI3KC3 generates PtdIns(3)P and recruits Atg8 to the FVS, presumably with the assistance of ROS generated by Duox in the egg and/or the MD itself (Fig. 8d, IV, V). Consequently, lysosomes fuse with strings of large Atg8 positive vesicles derived from the FVS and contain degraded MD pieces and debris, leaving behind the axoneme (Fig. 8d, VI, VII). Ultimately, this process assures inheritance of the mitochondria from the maternal lineage only.

**Fig. 6 | Atg8a recruitment to the FVS coincides with advanced MD degradation stages. a** A representative confocal image of a WT egg fertilized by green-MD sperm (magenta; stained with an anti-GFP antibody) and immunostained to visualize the axoneme (yellow) and Atg8a (cyan). Shown is a sperm flagellar region exhibiting at least three defined consecutive PME stages: Intact MD segments that are Atg8a negative and usually still reside in the vicinity of parallel axonemal segments (I); Atg8a positive FVS and derived strings of large vesicles that encapsulate degrading MD fragments (IIa) and hence result in MD-free axonemal segments (IIb); MD-free axonemal segments displaying neither MD traces nor Atg8a positive FVS, indicative of completion of the PME process (III). Scale bar, 4 µm. **b, c** Atg8 is required for efficient MD degradation. A live imaging assay performed and presented as in Fig. 1e. Early fertilized eggs expressing the maternal driver alone served as control (**b**, upper panel). Scale bar, 100 µm. **c** Quantifications of normalized red-MD fluorescence intensities [arbitrary units (a.u.)] in early fertilized eggs represented in (**b**), indicating that although single maternal knockdowns of *atg8a* and

*atg8b* have no effect on PME kinetics (Supplementary Fig. 6b–e), double maternal knockdown fertilized eggs exhibit significant PME attenuation. Error bars indicate SEM. The respective numbers of examined fertilized eggs (*n*) laid by control and *atg8a^ShR^; atg8b^ShR^* females are 51 and 76. ****$P < 0.0001$. Two-way repeated measures ANOVA, followed by Šídák's multiple comparisons test. MD, mitochondrial derivative. **d–g** Live confocal imaging of the anterior region of an early fertilized egg laid by a female maternally expressing both the *UASz-rubicon-eGFP* (green) and *UASp-mCherry-atg8a* (magenta) transgenic constructs. Shown are representative images from Supplementary Movie 7 at the indicated times (min AEL). Note the 6–12 min lag between the formation of Rubicon-eGFP positive FVS and the subsequent recruitment of mCherry-Atg8a. Scale bar, 10 µm. **h–k** Enlargement of the areas confined by dashed rectangles in (**d–g**), presenting the same flagellar region at different timepoints AEL. Scale bars, 4 µm. Source data are provided as a Source Data file.

An interesting question is how the MVBs specifically recognize and mediate degradation of the sperm MD, but not the axoneme or the maternal mitochondria. Since we show that the egg MVBs first engage the sperm flagellum plasma membrane, it is likely that they recognize specific lipid/s or protein/s placed on the exterior side of the sperm plasma membrane. Breakdown of the sperm plasma membrane exposes the two main flagellar organelles, the MD and the axoneme, to the luminal contents of the FVS. This leads to the degradation of the MD, but the axoneme is spared from a similar fate. In this study we also observed that the two organelles reside in separate compartments within the FVS, suggesting the existence of active mechanisms that direct the MD, but not the axoneme, to a degradative compartment within the FVS. Interestingly, cargo ubiquitination is a major MVB sorting mechanism[111–113], and we previously demonstrated that the MD, but not the axoneme, is specifically decorated with lysine 63-linked ubiquitin chains after fertilization[42], which may explain the different fates of these two organelles.

Whereas LAP is known to target extracellular pathogenic microbes by phagocytosis, targeting of the MD occurs within the egg ooplasm, following sperm entry. Nevertheless, these two processes are not only molecularly reminiscent of each other but also anatomically similar. In LAP, a single membrane LAPosomal vesicle encapsulates the cargo by inward invagination of the plasma membrane, while during PME, single membrane MVBs form extended FVS that encapsulate large flagellar segments. Interestingly, in addition to LAP, other pathways, such as LC3-associated endocytosis (LANDO) and LC3-dependent extracellular vesicle loading and secretion (LDELS), were reported to involve conjugation of Atg8/LC3 to the limiting membrane of endosomes and MVBs, respectively, indicating that functional convergence of the endocytosis and autophagy pathways may be more common than has been previously recognized. Furthermore, similar to *Drosophila* PME, induction of LDELS does not require surface receptor activation, and LANDO requires Rubicon[63,110,114–116].

A generally held belief is that elimination of the paternal mitochondria is important to prevent mtDNA heteroplasmy (i.e., a state of having more than one type of mtDNA in a cell), partially because sperm mtDNA was reported to have a higher mutation rate than egg mtDNA[117]. Furthermore, deliberate major mtDNA heteroplasmy in mice was reported to lead to reduced animal activity, increased anxiety, reduced feeding, and impaired spatial learning[118]. However, prevention of mtDNA heteroplasmy may not be the entire story, as the heteroplasmic state has been shown to be very unstable, and documented negative effects of paternal mtDNA leakage on the well-being of the organism are rare[8,17,118–129]. Furthermore, several reports suggest that in some cases, including in human and *Drosophila*, the sperm mtDNA is degraded much before the observed degradation of the vacuolar paternal mitochondria[18–22,24], implying that the latter degradation is a second line of defense against paternal mtDNA

transmission, or that other paternal mitochondrial factors are deleterious to the organism. An alternative theory is that, similar to the role of macroautophagy in providing nutrients during starvation, recycling of the paternal mitochondria may support proper embryonic development, by providing developmental cues or important substances to the fertilized egg, such as iron, copper, folates, etc[130–132]. Our findings that the egg triggers a conserved cellular defense pathway against invading microbes to target the MD for degradation, may support the idea that the paternal mitochondria might deliver deleterious factors to the embryos, as they imply that the egg regards the paternal mitochondria as a potentially dangerous trespasser, reminiscent of the manner by which other cells react against intrusive bacteria.

How Atg8 is attached to the FVS is unclear. Although the Atg5-Atg12-Atg16 complex is required for conventional autophagy[107], and is also reported to participate in some LAP paradigms[49,108–110], maternal knockdown of these genes had no effect on PME. In contrast, Atg7 and Rubicon are both involved in recruitment of Atg8 to the FVS. Interestingly, double knockdown of *atg8a* and *atg8b* significantly attenuates PME. These findings suggest that the Atg5-Atg12-Atg16 complex is likely dispensable for Atg8 recruitment to the FVS. It is noteworthy that other mechanisms of Atg8 recruitment and roles, in a manner independent of the Atg5-Atg12-Atg16 complex, have also been documented[100,133–138].

Paternal mitochondrial elimination and maternal mitochondrial inheritance are highly ubiquitous across the animal kingdom. Scattered evidence from different research groups suggests that at least some elements of the autophagy pathway are expressed on the flagellum of mammalian sperm, either before and/or immediately after fertilization[21,29–31]. Similar to the finding in the current study that the Atg12-Atg5-Atg16 complex is dispensable for PME in *Drosophila*, *Atg5*-deficient early mouse embryos also displayed normal PME[21]. Evidence from electron microscopy studies showed the presence of MVBs in the vicinity of the paternal mitochondria in some mammalian early embryos[27,65]. Collectively, it is attractive to propose that a LAP-like pathway similar to the pathway underlying PME in *Drosophila* might also mediate the process in mammals and perhaps also in other organisms that produce flagellated sperm.

The importance of understanding the mechanisms underlying PME is not only of academic interest, as the use of medically assisted reproduction (MAR) treatments is constantly increasing, leading to diversion of research efforts from molecular studies to improvement of MAR technologies. Consequently, much less attention has been given to possible negative impacts of some of these treatments, which may involve paternal mitochondrial leakage, on embryo development, implantation, and clinical and obstetric outcomes[139–141]. Furthermore, the novel reproductive technologies known as mitochondrial donation or mitochondrial replacement therapy (MRT), raise challenging ethical

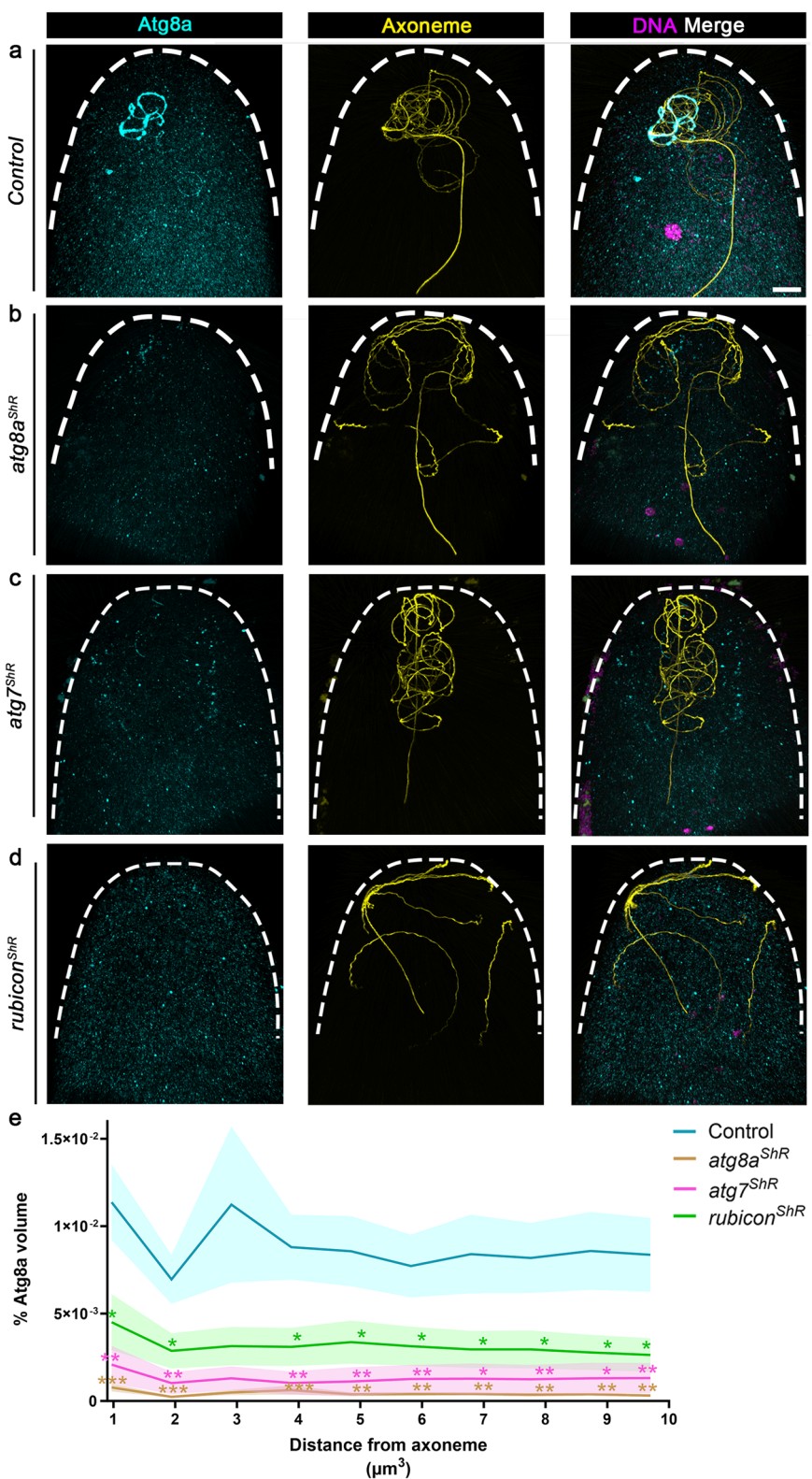

and safety issues, as only a few successful MRT case reports are available. In addition, since we still lack long-term studies of MRT outcomes, the potential risks of both major mtDNA heteroplasmy and the mixing of genetic materials (nuclear and mitochondrial genomes) from three parents are under much debate[142–146]. Therefore, delineating the molecular pathway underlying PME and the (possible negative) effects of paternal mitochondrial persistence on the developing embryo, will allow for a deeper understanding of this highly conserved phenomenon for embryo development, and may ultimately lead to the development and/or modification of MAR technologies that guard against these effects.

## Methods

### Fly strains used in this study

Fly stocks were maintained on a standard yeast/molasses medium at 25 °C. Unless indicated otherwise, *yellow white (yw)* flies were used as

**Fig. 7 | Atg8 recruitment to FVS requires Rubicon. a–d** Representative confocal images of the anterior regions in early fertilized eggs laid by control (**a**) and maternal *atg8a* (**b**), *atg7* (**c**), and *rubicon* (**d**) knockdown mothers. The eggs were fertilized by WT sperm cells, and immunostained to visualize Atg8a (cyan) and the axoneme (yellow). The 0–1-hour AEL fertilized eggs were further staged by the number of nuclei, as revealed by DAPI staining of the DNA (magenta). Scale bar, 20 μm. **e** The graphs depict staining volumes of Atg8a in early fertilized eggs corresponding to the fertilized eggs in (**a–d**). Calculations were performed in intervals of 1 μm in 10 μm radius area around the axoneme. The respective numbers of scored early fertilized eggs (*n*) laid by females of the control, *atg8a^ShR^*, and *atg7^ShR^*, and

*rubicon^ShR^*, are 20, 12, 11, and 15. *$P < 0.05$, **$P < 0.01$, and ***$P < 0.001$. *P* values are as follows: control vs. *atg8a^ShR^*, $P = 0.0003$, 0.0004, 0.0009, 0.0019, 0.0019, 0.0063, 0.0029, 0.0047, and 0.0035, for the distances 1, 2, 4, 5, 6, 7, 8, 9, and 10 μm³ from axoneme, respectively; control vs. *atg7^ShR^*: $P = 0.0013$, 0.0013, 0.0013, 0.0042, 0.0059, 0.0138, 0.0076, 0.0111, and 0.0098, for the distances 1, 2, 4, 5, 6, 7, 8, 9, and 10 μm³ from axoneme, respectively; control vs. *rubicon^ShR^*: $P = 0.0154$, 0.0236, 0.0128, 0.0348, 0.0379, 0.0371, 0.0294, 0.025, and 0.0203, for the distances 1, 2, 4, 5, 6, 7, 8, 9, and 10 μm³ from axoneme, respectively. Two-way repeated measures ANOVA, followed by Holm-Šídák's multiple comparisons test. Error bars indicate SEM. Source data are provided as a Source Data file.

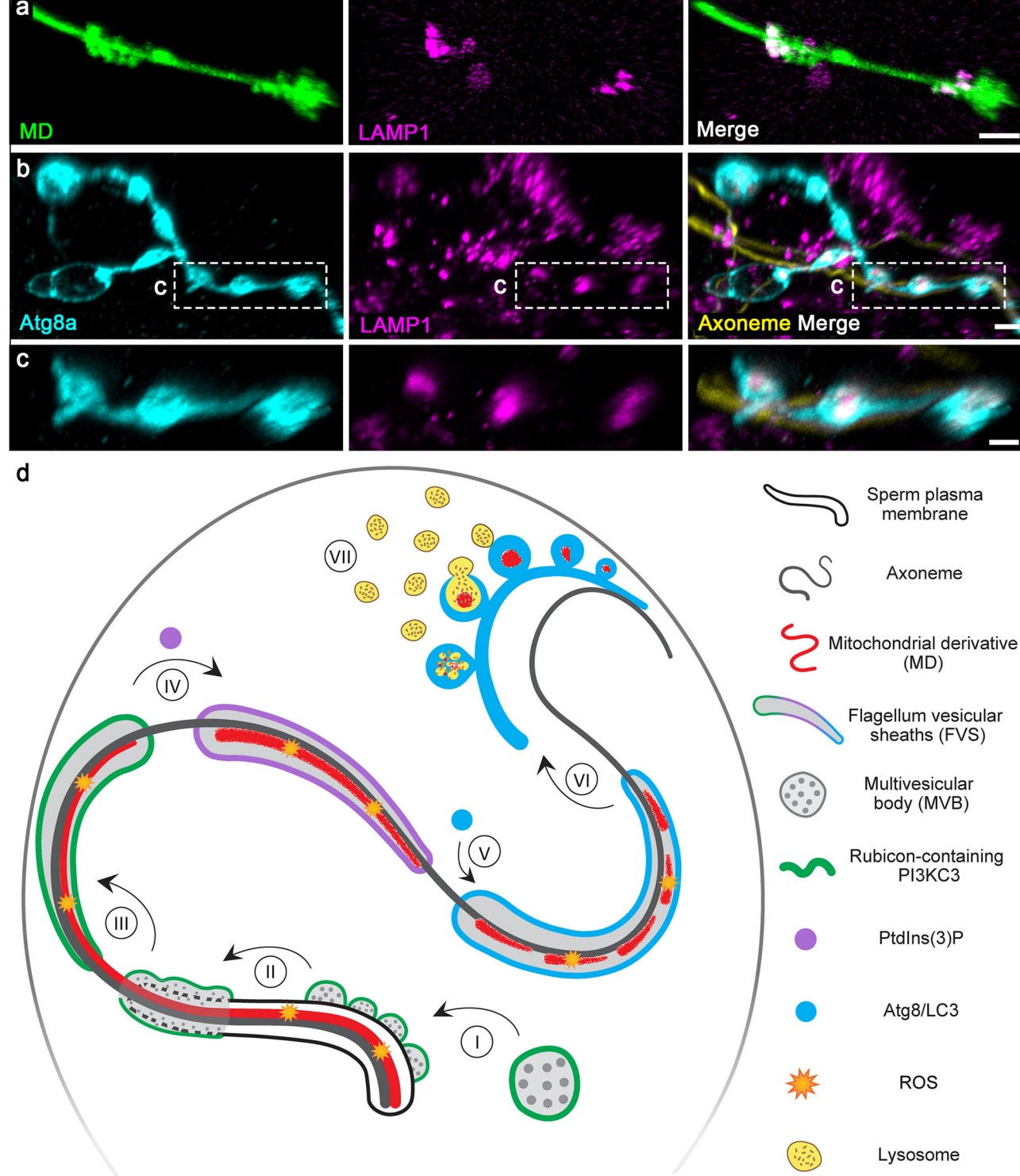

**Fig. 8 | Atg8 positive strings of vesicles derived from the FVS and contain MD fragments fuse with lysosomes. a** Live confocal imaging of a sperm flagellar region in an egg ubiquitously expressing a transgenic construct that marks the lysosomes (magenta; *tub-GFP-LAMP1*), and fertilized by a red-MD sperm cell (green). Scale bar, 2 μm. **b** A sperm flagellar region in an egg ubiquitously expressing the lysosomal transgenic marker, GFP-LAMP1 (magenta), fertilized by a WT (non-fluorescent) sperm cell, and immunostained to visualize Atg8a (cyan) and the axoneme (yellow). Note the budding of the string of Atg8 positive large vesicles from the FVS (i.e., Atg8a positive regions that are not associated with the axoneme), and the association of GFP-LAMP1 with these large vesicles, which correspond to the vesicles that contain degrading MD fragments (**a** and Fig. 6a). Scale bar, 3 μm. **c** Enlargement of the area confined by a dashed rectangle in (**b**). Scale bar, 2 μm. **d** An integrated model of PME by a LAP-like pathway in *Drosophila*. An illustration of a sperm flagellum inside the anterior part of an early fertilized egg. Relevant flagellar and LAP pathway components are indicated by different signs and colors as defined on the right-hand side. Seven PME pathway steps are indicated by Roman numerals. Egg-derived Rubicon positive MVBs engage the sperm flagellum plasma membrane in an asynchronous manner immediately after fertilization (I). The MVBs densely coat the sperm flagellum, forming extended vesicular sheaths, within which the sperm plasma membrane rapidly breaks down (II). The MD is exposed to the degradative luminal area of the FVS that express the LAP-specific Rubicon PI3KC3 (III). Active Rubicon PI3KC3 generates PtdIns(3)P (IV), which presumably with the help of ROS (produced by the egg and the MD), promotes the recruitment of Atg8/LC3 to the FVS (V). The Atg8/LC3 positive FVS bud off strings of large vesicles that contain degrading MD fragments (VI). Lysosomes fuse with the strings of large vesicles, promoting efficient degradation of the MD fragments (VII).

WT control. Fly strains, including mutant and transgenic flies used in this study are listed in Supplementary Table 1.

## Generation of the *rubicon* mutant alleles
The *rubicon^Δ8^*, *rubicon^Δ10^*, *rubicon^Δ13^* and *rubicon^Δ196^* mutant alleles were generated using CRISPR/Cas9-mediated mutagenesis. Briefly, *nos-Cas9* expressing flies were crossed to flies ubiquitously expressing a single guide RNA (gRNA) against *rubicon* (BDSC #81781). F1 females carrying both the *nos-Cas9* and the *rubicon* gRNA, and hence subjected to potential germline mutagenesis in the *rubicon* gene, were crossed with males carrying an X chromosome balancer (FM7), as the *rubicon* gene is located on the X chromosome. F2 females were then individually crossed again with FM7 males to establish stable fly lines with potential mutations in *rubicon*. Genomic DNA was extracted from each line, the *rubicon* gene was PCR amplified using forward (F) primer: CCGAGACAGTGCGTCAATG and reverse (R) primer: CTTCCTCGCTGGGATAGTACA, and sequenced to identify *rubicon* mutant lines.

## Generation of DNA constructs and transgenic lines
***dj-(MTS)tdTomato* (red-MD).** The tdTomato coding sequence (CDS) was PCR amplified from the *pRSETB-tdTomato* vector[66] using F primer: CATTCGTTGGGGGATCCACCGGTCGCCACCATGGTGAGCAAGGGCGAGGAGG and R primer: ATCTTGTTGTTTCGCAGGCGATTGCGGCCGTTACTTGTACAGCTCGTCCATGC. Applying the restriction free technique (RF)[147], the DsRed CDS in the *dj-(MTS)DsRed* plasmid[42] was replaced with the tdTomato CDS. Note that for expression during late-stage spermatogenesis, the *(MTS)tdTomato* is placed under the regulatory elements of the *dj* gene (i.e., promoter and 5′ UTR)[80,148], as well as the 3′ UTR of the male germ cell-specific gene *cyt-c-d*[149] [this vector is termed *dj*(P + 5′ UTR)//*cyt-c-d*(3′ UTR)]. Embryo injection to introduce the transgene into the attP2 site was carried out by BestGene Inc.

***dj-CD8-PARP^(DEVG)^-Venus* (sperm plasma membrane reporter).** This line was generated using restriction-ligation cloning. The *CD8::parp^DEVG^::Venus (UAS-CP^G^V)* transgene[150] was PCR amplified from the *pUASt-CD8:PARP:Venus* plasmid[151] using F primer: CGTAGATCTATGGCCTCACCGTTGACCCGCTTT and R primer: ATCCGCGGCCGCCTATTTGTACAATTCATCCATAC, and subcloned into the *dj*(P + 5′ UTR)//*cyt-c-d*(3′ UTR) vector. Embryo injection to introduce the transgene into the 3^nd^-chromosome through *P* element-mediated transformation was carried out by Genetic Services, Inc.

***UASp-eGFP-hCD63*.** The eGFP-hCD63 CDS was PCR amplified from a *pUASt-eGFP-hCD63* plasmid (kindly provided by Suzanne Eaton, the Max Plank Institute of Molecular Cell Biology and Genetics, Germany) using F primer: CCGCGCGGCCGCCATGGTGAGCAAGGGCGAGGAGCTG and R primer: GCGGACTAGTCTACATCACCTCGTAGCCAC. The eGFP-hCD63 CDS was subcloned into the *pUASp2* vector[68] using the restriction enzymes NotI and SpeI for the insert, and NotI and XbaI for the vector. Embryo injection to introduce the transgene into the 2^nd^-

chromosome through P element-mediated transformation was carried out by BestGene Inc.

***UASz-rubicon-eGFP*.** This line was generated using the *In-fusion* cloning technique (Takara) in two steps as follows: First, the *eGFP* DNA sequence was PCR amplified using F primer: GTACCGCCTCTCTAGGTGAGCAAGGGCGAGGAG and R primer: GAATTCACACTCTAGTTACTTGTACAGCTCGTCCATGCC. This amplicon was inserted into the *pUASz1.1* plasmid[69], obtained from the *Drosophila* Genomics Resource Center (DGRC; stock #1433), following digestion of the plasmid with XbaI to obtain a *pUASz-eGFP* vector. Next, the full-length *rubicon* CDS was PCR amplified from the cDNA clone *RH61467* (DGRC, Stock #11152) using F primer: CAAAGGATCCCTCGAATGACCACGCCCCCG and R primer: AGGCGGTACCCTCGAGCTGGCACGGCTTTGA. The amplicon was inserted into the *pUASz-eGFP* vector following digestion with XhoI to generate the *pUASz-Rubicon-eGFP* vector. Embryo injections to introduce the transgene into the attP2 and attP40 sites were carried out by FlyORF.

***UASz-hCD63-tdTomato*.** The *tdTomato* gene was first isolated from a plasmid by digestion with BamHI and ligated into BamHI linearized *pUASz1.1* plasmid to obtain the *pUASz-tdTomato* vector. *hCD63* CDS was PCR amplified from the *UASp-eGFP-hCD63* vector using F primer: CAAAGGATCCACCGGATGGCGGTGGAAGGAGGAAT and R primer: CATGGTGGCGACCGGCATCACCTCGTAGCCACTTCTGA. The amplicon was then inserted into the *pUASz-tdTomato* vector using the *In-fusion* cloning technique (Takara) following digestion with AgeI. Embryo injection to introduce the transgene into the attP2 site was carried out by FlyORF.

**Tagging the endogenous *rubicon* gene with tdTomato.** A single gRNA was designed to target Cas9-mediated excision around the stop codon of the *rubicon* gene (TTCTTTGATCGGATGTTAGC in the *pCFD5* plasmid). For homology-directed repair, we generated a donor plasmid in several steps as follows: We first inserted the *tdTomato* gene into the *pHD-DsRed-attP* vector, which contains a loxP flanked 3XP3-DsRed cassette that produces eye-specific expression of the DsRed[152]. Then, using *In-fusion* cloning (Takara), we subcloned into the *pHD-tdTomato-DsRed-attP* vector the two homology arms (HAs) which contain sequences flanking the Cas9 cleavage site. The HAs were PCR amplified from *yw Drosophila* genomic DNA using F primer: CTGGGCCTTTCGCCCCCAGTCTGACCAGAGGCACC and R primer: CCCCATAATTGGCCCTTGCTGGCACGGCTTTGAAGG, for the 5′ HA, and F primer: ATAGAAGAGCACTAGCATCCGATCAAAGAAAATCGAAGGG and R primer: GGAGATCTTTACTAGACGCGTTCGGCAAAATACC, for the 3′ HA. The *pHD-tdTomato-DsRed-attP* vector was digested with SmaI to first insert the 5′ HA and subsequently with SpeI to insert the 3′ HA, obtaining the *pHD-5′HA-tdTomato-3′HA-DsRed-attP* donor plasmid. Embryo injections to obtain transgenic lines were carried out by FlyORF. F1 progeny was screened for insertion events

using the eye DsRed expression. Positive lines were validated and then crossed to *nos*-Cre flies to remove the 3XP3-DsRed cassette.

## MVB purification from *Drosophila* eggs

To isolate the egg-derived MVBs, we devised a procedure that is based on two protocols, one for mitochondria isolation from *Drosophila* early fertilized eggs and the second for isolation of exosomes from cultured cells[153,154]. To adjust the procedure, MVB isolation trials were first performed on small-scale eGFP-hCD63 expressing early fertilized eggs (i.e., early fertilized eggs laid by females carrying the maternal driver and the *UASp-eGFP-hCD63* transgene and fertilized by WT males). Once adjusted, the procedure was then scaled up to produce a large amount of protein that is needed for MS analysis. To avoid slow fertilized egg collection due to repeated crossings, the large-scale MVB isolation was performed with WT (*yw*) early fertilized eggs. Collection, preparation, and MVB extraction from the eGFP-hCD63 expressing early fertilized eggs was performed as follows: Flies were placed in a population cage, allowing to lay eggs on juice agar plates with a dollop of thick yeast paste for 1 h. Early fertilized eggs were then collected in mesh baskets, dechorionated in 6% bleach, washed with water, shock frozen in liquid nitrogen, and stored at −80 °C until the day of procedure. All further steps were performed on ice and using a cooled centrifuge. 60 mg of early fertilized eggs were thawed on ice and transferred to a Dounce homogenizer containing 6-times volume of homogenization buffer (HB; 0.25 M sucrose, 1 mM EDTA, 0.03 M Tris pH 7.4, supplemented with protease inhibitor cocktail (Sigma-Aldrich, P8340)). The homogenate was centrifuged three times at 500 x g for 15 min to pellet cell debris. A fraction of the supernatant was removed and served as cell lysate control, while the remaining supernatant was centrifuged at 12,000 x g for 20 min. The pellet was then resuspended in HB and layered onto a 5-50% OptiPrep™ Density Gradient Medium (Sigma-Aldrich, D1556), and ultracentrifuged at 50,000 x g for 3 h in a SW41Ti rotor (Beckman). 12 fractions of 1 ml each were manually separated, diluted 1:4 in OptiPrep™ buffer (OB, 0.07 M EDTA, 0.03 M Tris pH 7.4) and centrifuged at 30,000 x g for 15 min. Fractions pellet was resuspended in SDS sample buffer and analyzed by Western blotting (WB) using anti-GFP antibody (Abcam, ab 290, 1:1,000). GFP-positive fractions (5 and 6) were subjected to TEM analysis to assess the level of MVB purification. The scaled-up procedure was applied to 2 gm of *yw* early fertilized eggs, and the homogenization buffer volume was accordingly adjusted (all other steps were not modified). Fractions 5 and 6 were combined and subjected to proteomic analysis.

## Transmission electron microscopy

Electron micrographs of early fertilized eggs shown in Supplementary Fig. 1a–c, were generated essentially as described in ref. 42.

Electron micrographs of isolated MVBs were prepared as follows. Samples were fixed in a solution containing 4% paraformaldehyde (Electron Microscopy Sciences, EMS) and 2% glutaraldehyde (EMS) in 0.1 M cacodylate buffer containing 5 mM CaCl2 (pH 7.4). Fixed samples were drawn into cellulose capillary tubes with an inner diameter of 200 μm[155]. Samples were postfixed in 1% osmium tetroxide (EMS) supplemented with 0.5% potassium hexacyanoferrate tryhidrate (BDH chemicals) and potassium dichromate (BDH chemicals) in 0.1 M cacodylate for 1 h, stained with 2% uranyl acetate (EMS) in double distilled water for one hour, dehydrated in graded ethanol solutions and embedded in epoxy resin (Agar scientific Ltd.). Ultrathin sections (70 nm) were obtained with a Leica EMUC7 ultramicrotome. Sections were transferred to 200 mesh copper transmission electron microscopy grids (SPI). Grids were stained with lead citrate (Merk) and examined with a Tecnai T12 transmission electron microscope (Thermo Fisher Scientific). Digital electron micrographs were acquired with a bottom-mounted TVIPS TemCam-XF416 4k × 4k CMOS camera.

## Mass spectrometry

MVB-containing fractions and cell lysate control were resuspended in lysis buffer (5% SDS, 50 mM Tris pH 7.4) and centrifuged at 14,000 x g for 5 min. Pellets were discarded, and the supernatant was collected. Samples were reduced in 6 mM dithiothreitol and alkylated with 12 mM iodoacetamide in the dark. Each sample was loaded onto S-Trap minicolumns (Protifi, USA) according to the manufacturer's instructions. In brief, after loading, samples were washed with a solution of 90% methanol and 10% 50 mM Triethylammonium bicarbonate. Samples were then digested with trypsin (Promega) at 1:50 trypsin/protein ratio for 90 min at 47 °C. The digested peptides were eluted using 50 mM ammonium bicarbonate; trypsin was added to this fraction and incubated overnight at 37 °C. Two more elutions were made using 0.2% formic acid and 0.2% formic acid in 50% acetonitrile. The three elutions were pooled together and vacuum-centrifuged to dry.

UPLC/MS grade solvents were used for all chromatographic steps. Each sample was loaded using splitless nanoUltra Performance Liquid Chromatography (10k psi nanoAcquity; Waters, Milford, MA, USA). The mobile phase was: A) H2O + 0.1% formic acid and B) acetonitrile + 0.1% formic acid. The samples were desalted online using a reversed-phase Symmetry C18 trapping column (180 μm internal diameter, 20 mm length, 5 μm particle size; Waters). The peptides were then separated using a T3 HSS nano-column (75 μm internal diameter, 250 mm length, 1.8 μm particle size; Waters) at 0.35 μL/min. Peptides were eluted from the column into the mass spectrometer using the following buffer B (99.9% Acetonitrile, 0.1% formic acid) gradient in three steps: (1) 4% to 25% of buffer B for 150 min, (2) 25% to 90% of buffer B for 5 min, (III) 5 min incubation in 90% buffer B, and then back to initial conditions. Samples were run in random order.

A nanoESI emitter (20 μm tip, Fossilion Tech, Madrid, Spain) was used on a FlexIon source (Thermo Fisher), mounted on a Tribrid orbitrap mass spectrometer (Fusion Lumos, Thermo Fisher). Data were acquired in data-dependent acquisition (DDA) mode, using a Top Speed method for 3 sec. MS1 resolution was set to 120,000@200 m/z, mass range of 380–1650 m/z, AGC at Standard and maximum injection time was set to 50 msec. Precursors selected for MS2 were limited to charge states of 2–8 with a minimum intensity of 50000. MS2 resolution was set to 15,000@200 m/z, quadrupole isolation 1 m/z, first mass at 130 m/z, AGC set at Standard. Dynamic exclusion of 30 sec with +/−10 ppm window and maximum injection time set to 100 ms. Ions were fragmented using HCD at 30 NCE.

The data analysis was performed using Byonic (v3.3.11) search engine (Protein Metrics)[156] against the *Drosophila melanogaster* proteome database (UniProt Nov 2018). Data were searched against specific C-terminal cleavage at K and R, allowing one missed cleavage at a tolerance of 10 ppm in MS1 and 20 ppm in MS2, using HCD settings for precursors up to 5500 Da. Modifications allowed were fixed Carbamidomethylation on C and variable modifications on the following amino acids: oxidation on M, deamidation on NQ and protein N-terminal acetylation. Data was filtered for 1% FDR. Quantification was performed by FlashLFQ[157] without match between runs or normalization since the samples were very different from one another. Protein annotations obtained from MS analysis were scored by the number of peptide sequences that are unique to a protein group (unique peptides value), and the number of peptide spectrum matches (PSM value), which is the total number of identified peptide spectra matched for the protein. Low abundance proteins (PSM < 10) were disregarded, and $Log_2$ fold change ($log_2FC$) ratios were calculated to obtain a list of MVB-enriched proteins ($log_2FC > 1$). Pathway enrichment analysis was performed using Flymine[158].

## Western blotting

Samples were run in SDS-PAGE and transferred to nitroglycerin membrane. After blocking in 5% milk solution (dry milk in PBTw), the membrane was incubated with anti-GFP (Abcam, ab290 1:1000),

followed by incubation with anti-rabbit HRP secondary antibody (1:10,000) for 1 h at room temperature (Jackson Immuno-Research). After washes in PBTw, images were obtained using the ImageQuant LAS 4000 mini.

## Live imaging

**MD elimination kinetic assay.** Female flies of the appropriate genotype were crossed with males producing red-MD sperm in population cages, and were allowed to lay eggs on juice agar plates with a dollop of thick yeast paste for 15 min. Early fertilized eggs were collected in mesh baskets, dechorionated in bleach and washed with water. Early fertilized eggs were then aligned on an agar pad, picked up on an embryo glue (Permanent double-sided tape, Scotch®, 665)-coated coverslip and covered with halocarbon oil 27 and 700 (1:1 mixture, Sigma-Aldrich H8773 and H8898, respectively). Early fertilized eggs were then imaged under a Nikon Eclipse fluorescence microscope. 2D images were acquired for each early fertilized egg at 1 h intervals for the duration of 3 h AEL. Live imaging assays of *atg14, atg8b* and double knockdown of *atg8a* and *atg8b* were imaged under a Zeiss Axio Observer Z1 inverted brightfield/fluorescence microscope and a Hamamatsu ORCA-Flash4.0 digital CMOS camera.

**High-resolution 3D live imaging.** Early fertilized eggs at 0–15 min AEL of the appropriate genotype were collected and mounted for imaging as described above. Imaging was performed using a Dragonfly Spinning disc confocal system (Andor Technology PLC) mounted on an inverted Leica Dmi8 microscope (Leica GMBH). Images shown in Fig. 3a–d and Fig. 5a, b, d, e were deconvolved with the internal Andor Fusion deconvolution application. MIP (Max-Intensity-Projection) processing was performed using Imaris 9.5.0 (Bitplane, http://www.bitplane.com/, RRID:SCR_007370).

## ROS staining

**Early fertilized eggs.** WT early fertilized eggs at 0–15 min AEL were collected and dechorionated as described above. The early fertilized eggs were then permeabilized by 1 min incubation in EPS solution (90% D[+] Limonene [Thermo Fisher Scientific, FL/1860/07], 5% ethoxylated alcohol [Bio-Soft N1-7], 5% cocamide DEA [kind gift from Sano International; https://www.sano-international.com/], diluted 1:10 in pre-warmed [37 °C] PBS)[159]. Early fertilized eggs were washed four times in PBS and incubated in 10 µM 2,7-Dichlorodihydrofluorescein diacetate solution (H2DCFDA; Cayman chemical, 85155) for 3 min. Early fertilized eggs were then washed twice in PBS and mounted and imaged as described above.

**Sperm.** Seminal vesicles (SVs) were dissected from males producing the red-MD sperm. The SVs were transferred to a drop of Schneider's *Drosophila* Medium with L-Glutamine (Biological industries, Beit Haemek, Israel, 01-150-1 A) on a Poly-L-Lysine coated coverslip. SVs were punctured using forceps and incubated in 1 µM H2DCFDA for 10 min. H2DCFDA solution was then replaced with fresh Schneider's medium and imaged as described above.

## Early fertilized egg fixation and immunostaining

Early fertilized eggs at 0–1-hour AEL of the appropriate genotype were collected and dechorionated as described above. Early fertilized eggs were placed in boiling Triton/Salt solution (0.7% NaCl, 0.04% TritonX-100) for 5 sec, and immediately moved to cool Triton/Salt solution on ice for 15 min. Vitelline membranes were removed by manual shaking in 50% methanol: 50% heptane for 1 min. Early fertilized eggs were rehydrated by passing through a series of solutions with decreasing methanol concentrations, washed with 1× PBS + 0.1% TritonX-100 (PBTx), and blocked in 5% normal goat serum in PBTx for 1 h. Then, the early fertilized eggs were incubated with the relevant primary antibody overnight at 4 °C, washed in PBTx, and incubated with the secondary antibody at room temperature for 2 h. Early fertilized eggs were washed in PBTx and then mounted onto slides in Fluoromount medium (SouthernBiotech, Birmingham, AL, USA).

For expansion microscopy, early fertilized eggs were fixed as described in ref. 160. Briefly, dechorionated early fertilized eggs were placed in a mix of 4% formaldehyde (diluted in PBS) and heptane (1:1 ratio) and vigorously shaken for 20 min. After removal of the aqueous layer, the early fertilized eggs were placed in ice-cold 50% methanol: 50% heptane and vigorously shaken for 1 min. All further steps were conducted as described above.

Primary antibodies used in this study are biotin Anti-GFP antibody (1:100, Abcam, ab6658), anti-*Drosophila melanogaster* Atg8a polyclonal antibody (1:100, Creative-diagnostics,CABT-L1690), anti-pan polyglycylated Tubulin antibody, clone AXO 49 (1:5000, Sigma-Aldrich, MABS276). Anti-RFP (1:100, ROCKLAND, 600-401-379). All secondary antibodies (Jackson Immuno-Research) were used at a dilution of 1:250. For expansion microscopy, antibody concentrations were doubled. Anti-Mouse-IgG-Atto-647N (Sigma-Aldrich, 50185) was used at a dilution of 1:10. All images were acquired using the Andor Dragonfly Spinning disc confocal imaging system (Andor Technology PLC) mounted on an inverted Leica DMi8 microscope (Leica GMBH).

## Expansion microscopy

Early fertilized eggs expansion protocol was adapted from[82]. Early fertilized eggs were incubated with 1 mM MA-NHS (methacrylic acid N-hydroxy succinimidyl ester, Sigma-Aldrich, 730300) in PBS for 1 h at room temperature, followed by 3 × 20 min washes in PBS. Samples were then incubated in monomer solution [2 M NaCl, 8.625% sodium acrylate (Sigma-Aldrich, 408220), 2.5% acrylamide (Bio-Rad, 1610140), 0.15% bisacrylamide (Bio-Rad, 1610142) in PBS] for 1 h at 4 °C. The solution was then replaced with gelation solution {monomer solution + 0.01% 4-hydroxy-2,2,6,6-tetramenthylpiperidin-1-oxyl [4-hydroxy-TEMPO (Sigma-Aldrich, 176141)], 0.2% tetramethylethylenediamine (Bio-Rad, 1610800) and 0.2% ammonium persulfate (Bio-Rad, 1610700)}. The samples were mounted in bridged glass slides (Superfrost microscope slide with two glued 24 mm coverslips with a gap in between) and incubated in a humid chamber at 37 °C for 2 h. At this stage, the gelated samples were taken to the microscope for pre-expansion imaging. Next, the gels were cut around each early fertilized egg and digested in 8 U/ml proteinase K solution in digestion buffer (40 mM Tris, pH 8, 1 mM EDTA, 0.5% Triton, 0.8 M guanidine HCl) for 1 h at 37 °C. The samples were washed 3 X in PBS and placed in double distilled water for expansion. Water was exchanged three time before a final overnight incubation. The samples were transferred into Poly-L-Lysine coated 8 well chamber slide (µ-Slide 8 Well high, Ibidi, 80806), covered with water and imaged as described above.

Expansion factors were calculated by dividing post-expansion average early fertilized egg width by its pre-expansion measurement. Scale bars have been divided by their respective measured expansion factors of approximately four times and therefore correspond to pre-expansion dimensions.

## Image analysis

**MD elimination.** To quantify MD intensity, two-channel 2D images were acquired for each early fertilized egg: A bright field (BF) channel and a fluorescent channel of the (MTS)tdTomato (red-MD). Early fertilized eggs that failed to complete cellularization were omitted from the analysis. Each image includes one or two early fertilized eggs. We used the BF channel to segment the whole early fertilized egg using Ilastik AutoContext pixel classifier[161]. We trained the Ilastik classifier on multiple images from multiple different conditions. To avoid artifacts on the edge of the early fertilized egg, we eroded the early fertilized egg segments by 15 pixels. Small non-early fertilized egg segments were discarded from further analysis. We then identified positive MD

regions by applying background subtraction with Rolling ball (sigma = 35 pixels) to the fluorescent channel and selecting all the pixels above a fixed selected value (1200). For each early fertilized egg, we measured the total and average signal in the whole early fertilized egg, MD positive regions and MD-negative regions, and calculated the difference between the average MD positive signal to the average MD-negative signal. We saved segmented regions on top of the BF channel and the fluorescent channel for quality control. The regions-of-interest of all early fertilized eggs were saved into a file, to enable manual correction in case of incorrect early fertilized egg segmentation. Normalized MD intensity values were calculated for each early fertilized egg, by subtracting the mean intensity value of the background (MD-negative) from the mean MD intensity value (MD positive), and multiplying by the fraction area of intensity above threshold, to take into account the MD size. The above workflow was implemented as a Fiji 125 macro[162], which also enables the above measurement from manually modified regions instead of automatic regions, to overcome segmentation problems. Experiments imaged on the Zeiss Axio Observer Z1 inverted brightfield/fluorescence microscope were analyzed by the same code with new Ilastik classifier to segment the early fertilized eggs. The intensity threshold value was adjusted to 250. All other parameters were the same.

**Atg8a recruitment.** To determine Atg8a recruitment levels to the sperm flagellum, we used an Arivis Vision4D pipeline to analyze the relative volume of Atg8a as a function of the distance from the axoneme, in 0–1-hour AEL early fertilized eggs immunostained to reveal the axoneme and Atg8a. First, the axoneme was identified using a random forest pixel classifier. False positive axoneme objects were manually removed. Then, the Atg8a signal was identified using a random forest pixel classifier. To save computation time and avoid background signal, the smallest 40% Atg8a objects were filtered out from each file, such that only 60% of the total Atg8a volume was further analyzed. We verified on a subset of the data that we get the same spatial distribution of the Atg8a objects when analyzing all the objects or just the bigger ones as described herein. Next, total Atg8a object volume was measured in 10 envelopes of $1\,\mu m$ thick around the identified axoneme (the first envelope captures Atg8a objects with less than $1\,\mu m$ distance from the axoneme, the second envelope captures Atg8a objects residing between 1 and $2\,\mu m$ distance and so on). Then, the normalized Atg8a volume was obtained by dividing the total Atg8a volume within an envelope by the corresponding envelope volume.

**Sperm plasma membrane analysis.** To determine sperm plasma membrane volume, we used an Arivis Vision4D pipeline to analyze the sperm plasma membrane signal in proximity to (touching) the axoneme. 0–1-hour AEL early fertilized eggs (further staged to 0–4 nuclei) were immunostained to reveal the axoneme and the sperm plasma membrane. First, the axoneme was identified using a random forest pixel classifier. False-positive axoneme objects were manually removed. Then, the sperm plasma membrane signal was identified using a random forest pixel classifier. The normalized sperm plasma membrane volume was obtained by dividing the total sperm plasma membrane volume with the corresponding axoneme volume.

**Rubicon-eGFP vesicles and hCD63-tdTomato analysis.** To determine the correlation between Rubicon and hCD63 signal in Rubicon positive vesicles, we used an Arivis Vision4D pipeline, applied on live images of 0–15 min AEL early fertilized eggs, dually expressing Rubicon-eGFP and hCD63-tdTomato. First, Rubicon vesicles were identified using a random forest pixel classifier. Objects with a volume smaller than $15\,\mu m^3$ were filtered out. hCD63-tdTomato mean intensity

value was determined as follows: First, hCD63-tdTomato mean intensity signal was measured within each Rubicon-eGFP vesicle. To evaluate hCD63-tdTomato background intensity values, two surrounding envelopes were generated around each vesicle. A first 200 nm thick hCD63-tdTomato signal envelope was used to separate between each vesicle and its background. A second $300\,\mu m$ thick hCD63-tdTomato signal envelope was used to determine hCD63-tdTomato background signal, by averaging the total mean intensity of all second envelopes in each early fertilized egg. To obtain the normalized hCD63-tdTomato mean intensity value of each vesicle, the total mean intensity value of the background was extracted from each vesicle. Vesicles in which the fluorescent signal was lower compared to the engulfing envelopes signal, were omitted from further analysis. Rubicon vesicles that exhibit the hCD63-tdTomato signal above background were considered positive.

**eGFP-hCD63 signal analysis.** To determine the eGFP-hCD63 signal intensity on the MD, we used an Arivis Vision4D pipeline, applied on live images of 0–15 min AEL early fertilized eggs, expressing eGFP-hCD63 and fertilized with red-MD sperm. First, the red-MD was identified using a random forest pixel classifier. Then, eGFP-hCD63 mean intensity signal within the red-MD was measured. To separate between the red-MD and its background, a first $0.5\,\mu m$ thick eGFP-hCD63 envelope signal was not measured. Then, an additional $1\,\mu m$ thick eGFP-hCD63 envelope was generated, and its mean intensity was considered as background. The eGFP-hCD63 mean intensity of the second envelope was calculated (background) and extracted from the corresponding red-MD, to obtain the normalized eGFP-hCD63 mean intensity value.

**3D reconstruction and movie of ExM images.** Images generated by expansion microscopy were computed using the Imaris surfaces model. To generate the movie presented in Supplementary Movie 3, we used 3D maximum intensity projection of confocal optical section, followed by surface reconstruction computationally rendered with the Imaris software. The sequence of z-slices was generated using the Ortho slicer.

**Determination of relative RNA expression levels by RT-qPCR**
Maternal gene knockdowns were validated by comparing the levels of the examined genes in control early fertilized eggs (expressing the *matα-GAL4* driver alone) and early fertilized eggs expressing corresponding *UAS-ShR* transgenes. Total RNA was isolated from 0–30 min AEL early fertilized eggs, using the Quick-RNA Microprep Kit (Zymo Research, R1051). cDNA was then synthesized using the High-Capacity cDNA Reverse Transcription Kit (Applied Biosystems™, 4368814). Gene expression levels were determined by RT-qPCR using the StepOnePlus™ Real-Time PCR System (Applied Biosystems™, 4376600) with the KAPA SYBR® FAST qPCR Master Mix Kit (KAPA Biosystems, KR0389_S-v2.17). Expression values were normalized to the *αTub84B* gene (CG1913). Fold change in the expression levels in the knockdown validation experiments were analyzed using the ΔΔCt method[163], while relative mRNA levels in Supplementary Fig. 5d are represented by ΔCt values. Primer sets were designed using the Fly-PrimerBank online database[164] and are listed in Supplementary Table 2.

***rubicon* mRNA levels determined by RT-PCR**
mRNA levels of *rubicon* were determined as follows. cDNA was synthesized from 0–30 min AEL early fertilized eggs as described above. Fragments from three different areas across the *rubicon* gene were PCR amplified using Phusion® High-Fidelity DNA Polymerase (New England Biolabs, M0530S). *aTub84b* was used as cDNA template

loading control (primers are indicated above). The primers used to amplify *rubicon* are listed herein.

| Fragment 1 | ATGACCACGCCCCCG |
|---|---|
| | CAGTCCATGCTGCAAAATTCTAGTGC |
| Fragment 2 | GGCCCAAAGCTGCACTAGAAT |
| | AACAAGAACTGGTAGATGGGGCA |
| Fragment 3 | TTCCTGCGACGATCCGG |
| | AACAAGAACTGGTAGATGGGGCA |

### Fertility tests

Fertility was evaluated using either 3-day-old adult males or females that are *rubicon* mutant or knockdown [germline knockdown (males) and maternal knockdown (females)]. Each fly was crossed for 3 (males) or 5 (females) days with 3 WT virgin females or males, respectively. Subsequently, the parents were removed, and adult progeny numbers were scored for a week after the eclosion of first progeny.

To determine embryonic cellularization stage, we used brightfield microscopy combined with MD fluorescence, as described in the "Image analysis" paragraph of the Methods section. Fertilization of the eggs was assessed by the presence of the red-MD sperm. Of these fertilized eggs, we counted the number of eggs that developed to the cellularization stage by the third hour AEL.

### Data visualization, statistics, and reproducibility

Illustrations in Fig. 1d and Supplementary Fig. 1d were generated using BioRender.com. The pathway enrichment analysis graph was generated using R (R Core Team, 2020 https://www.r-project.org/, RRID:SCR_001905). Illustrations in Fig. 1a and Fig. 8d were generated using Adobe Illustrator (Adobe Inc. [2019]. Retrieved from https://adobe.com/products/illustrator).

All graphs and statistical analyses in this manuscript were generated using the GraphPad Prism software version 9.5.1 for Windows (GraphPad Software, San Diego, California USA, www.graphpad.com), except for the evaluation of the co-localization of Rubicon vesicles with hCD63, which was done using the colocr R package[165]. The specific statistical tests that were used for determining the significance between experimental groups, as well as the experimental reproducibility, are all indicated in the relevant figure legends. Significance is indicated by asterisks as follows: $*p < 0.05$, $**p < 0.01$, $***p < 0.001$ and $****p < 0.0001$.

For all MD elimination kinetic assay experiments, 2–3 biological replicates were performed. Specific n numbers for each genotype are indicated in the figure legends. Note that for *rubicon*, several different mutant alleles and an shRNA transgene displayed similar phenotypes.

At least three biological replicates were performed in Figs. 1b, 3a, 5d, e, and 6a. For Figs. 3d, 5h, 6d–k, and 8a, two biological replicates were performed. Note that early fertilized eggs stained with H2DCFDA display variations in intensity levels of this indicator signal, likely due to penetration efficiency. A single biological replicate was performed in Figs. 4a, f, and 8b, each containing at least eight early fertilized eggs with identical results. The MS analysis in this paper is a result of a single biological replicate.

No statistical method was used to predetermine the sample size. Sample size was chosen according to standard practices in the laboratory and general area of study. For the PME kinetics assay, more than 20 early fertilized eggs were analyzed per genotype.

To reduce background noise in the proteomic analysis (Supplementary Data 1 and Supplementary Fig. 1), proteins with the lowest abundance (PSM < 10) were discarded from the analysis (both from the

MVB fraction and control lysate). To save computation time and avoid background signal in the Atg8a recruitment experiment (Fig. 7e), 40% of the smallest Atg8a objects were filtered out from each image, and only 60% of the total Atg8a volume was further analyzed. Using a subset of the data, we verified that the same spatial distribution of the Atg8a objects is obtained when analyzing all the objects or only the larger (60%) objects as described herein.

The experiments were not randomized.

The investigators were not blinded to allocation during experiments and outcome assessment.

### Reagents and software

Reagents and software used in this study are listed in Supplementary Tables 3 and 4, respectively.

### Reporting summary

Further information on research design is available in the Nature Portfolio Reporting Summary linked to this article.

## Data availability

The authors declare that the data supporting the findings of this study are available within the paper and its supplementary information files, and that all additional data are publicly available. The MVB proteomics data generated in this study have been deposited in the MassIVE database under accession code MSV000093306 [https://massive.ucsd.edu/ProteoSAFe/dataset.jsp?task=a6ef3212194849cdb44b3b50f5d8488e]. Source data are provided with this paper.

## Code availability

The custom code used to quantify the kinetics of PME is deposited into Github repository (github.com/WIS-MICC-CellObservatory/PME_Kinetics). Arivis pipelines generated in this study are deposited into Github repository (github.com/WIS-MICC-CellObservatory/Atg8a-and-Axoneme-analysis/tree/main).

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

## Acknowledgements

We are grateful to Satomi Takeo, Graydon B. Gonsalvez, Suzanne Eaton, the *Drosophila* Genomics Resource Center (DGRC), the *Drosophila* RNAi Screening Center (DRSC) and Transgenic RNAi Project (TRiP), the Bloomington *Drosophila* Stock Center, and the Forchheimer Plasmid Bank of the Weizmann Institute of Science for providing additional stocks and reagents. We thank Johannes Bischof from FlyORF for help with generating endogenously tagged *rubicon*, Jordana Lindner-Ovadia for creating the graph in Supplementary Fig. 1f, Yoseph Addadi and the de Picciotto Cancer Cell Observatory in Memory of Wolfgang and Ruth Lesser at the Weizmann Institute of Science for help with microscopy imaging and analysis, Moshe Peretz for help with subcellular fractionation assay, Tsviya Olender for help with MS data analysis, Ron Rotkopf for help with statistical analysis, and Tslil Braun for providing additional reagents. We note Naama Afgin, Bar Lavi Lib, Doreen Padan Ben Yashar, and Rebeca Gonzalez-Rolfe, for helping to carry out experiments. We acknowledge the BioRender website which was used to create the illustrations presented in Fig. 1d and Supplementary Fig. 1d. We warmly thank Genia Brodsky and Iryna Savych from the WIS Graphic Design Department for help with the graphic illustration in Fig. 1a and movie editing and annotation, respectively. Electron microscopy studies were conducted at the Irving and Cherna Moskowitz Center for Nano and Bio-Nano Imaging at the Weizmann Institute of Science. Mass spectrometry was conducted at the de Botton Institute for Protein Profiling, The Nancy and Stephen Grand Israel National Center for Personalized Medicine, at the Weizmann Institute of Science. We thank the Arama laboratory members for encouragement and advice. This research was supported by grants from the European Research Council under the European Union's Seventh Framework Programme (FP/2007-2013)/ERC grant agreement (616088), the ISRAEL SCIENCE FOUNDATION (grant No. 1279/19), the Minerva Foundation with funding from the Federal German Ministry for Education and Research, and the Kekst Family Institute for Medical Genetics at The Weizmann Institute of Science. E.A. is supported by research grants from the Estates of Emile Mimran, Zvia Zeroni, Manfred and Margaret Tannen, and Betty Weneser. E.A. is the Incumbent of the Harry Kay Professional Chair of Cancer Research. A.K. is supported in part by the Crown Human Genome Center at The Weizmann Institute of Science.

## Author contributions

S.B.-H. designed, performed, and analyzed the experiments, and performed statistical analyses. S.S. provided invaluable help with several experiments and analyses during the revision period. S.A. provided technical assistance with specific experiments. A.K. performed the expansion microscopy experiments, imaging, and computing. Y.P., L.G., and A.F. generated the Red-MD sperm-producing transgenic flies (*dj-(MTS)tdTomato*), the *UASp-eGFP-hCD63*, and the *dj-CD8-Venus* transgenic fly lines, respectively. Y.P. also performed the TEM experiment shown in Supplementary Fig. 1a–c. O.G. wrote the data analysis code for the MD elimination kinetic assay. E.Si. wrote the data analysis code for Atg8a recruitment assay and generated the pipeline for analyzing the sperm plasma membrane volume, the vesicle segmentation of Rubicon-eGFP and hCD63-tdTomato signal analysis, and the flagellar eGFP-hCD63 signal intensity analysis. N.D. performed the TEM experiment shown in Supplementary Fig. 1e. D.M. performed the MS experiment and related initial analysis. S.P. performed the pathway enrichment analysis of the proteomic data. E.Sc. provided critical advice throughout the project and helped with the manuscript for publication. K.Y.-S. performed the fertility tests and staining of the mature sperm with the ROS indicator, and was responsible for the supervision of some

aspects of the study, E.A. led the project, designed experiments, interpreted results, was responsible for the general supervision of the study, wrote the manuscript, and procured funding.

## Competing interests

The authors declare no competing interests.
