## [Peer Review file · Nature Communications]

REVIEWER COMMENTS

Reviewer #1 (Remarks to the Author):

In this manuscript, the authors reported that paternal mitochondria are eliminated by a LC3-associated phagocytosis (LAP)-related mechanism in *Drosophila* embryos.

The authors have demonstrated that paternal mitochondria are surrounded by egg multivesicular bodies (MVBs) in embryos and degraded within 1 hour after egg laying. In this process, egg-derived MVBs associate with the sperm flagellum and drive the segmentation of the sperm flagellum. The authors have also revealed that LAP-specific class III PI(3)K complex (PI3KC3) components including Rubicon localize to the limiting membrane of egg MVBs, which surround the sperm flagellum and are required for the recruitment of Atg8/LC3 to extended flagellum vesicular sheaths (FVSs) and the degradation of paternal mitochondria during embryogenesis. Furthermore, they have shown that the production of PI(3)P by PI3KC3 and ROS by Duox, respectively, is required for the paternal mitochondria elimination (PME). Finally, the authors have shown that an anti-human RUBICON antibody labeled mouse and bull sperm midpiece including flagellum and mitochondria, suggesting that a LAP-like pathway mediates PME in mouse and bull embryos. These results raise a possibility that the LAP-like pathway is required for PME not only in fly but also in mammalian embryos. This paper contains some interesting results, but several points are not sufficiently supported and additional data or further explanation is necessary.

Major concerns:

1) The authors showed that paternal mitochondria are eliminated by a LAP-like mechanism in *Drosophila* eggs. LAP was originally identified as a system for immune regulation against pathogens. In contrast to mammals, the *Drosophila* sperm cell penetrates the egg with its plasma membrane through the micropyle. The sperm plasma membrane is then broken down and paternal mitochondria are actively destroyed in the egg cytoplasm. It would be good to analyze the process of sperm plasma membrane disruption as well as paternal mitochondrial elimination according to the time series by live-imaging analysis. The authors showed that egg MVBs cover the sperm flagellum and paternal mitochondria and form FVSs. However, the location of egg MVBs, sperm flagellum, and paternal mitochondria is unclear in Fig. 1b. If possible, ultrastructural analysis by immunoelectron microscopy should be performed to analyze how egg MVBs are positioned relative to the sperm-derived organelle.

2) It is difficult to discern the subcellular localization of each protein due to the low resolution of the images shown in this paper. Although the authors claimed that small vesicles around paternal mitochondrial form extended flagellum vesicular sheaths (FVSs), the presented data is not enough to support this conclusion. For example, it seems that Rubicon directly localizes to the whole region of the paternal mitochondrial membrane as well as large MVBs containing PM pieces. Does Rubicon directly bind to paternal mitochondria membrane as well as MVBs or indirectly bind to PM via MVBs? Super-resolution microscopy or EM would be necessary to distinguish these possibilities. At least, the authors should provide images with higher magnification. In addition, PI(3)P also seems to distribute throughout the paternal mitochondrial membrane. If so, how does PI3KC3 localize to the paternal mitochondrial membrane and produce PI(3)P. When Rubicon is knocked down, is the signal of PI(3)P abolished on MVBs and the paternal mitochondrial membrane?

3) LAP was originally identified as system for immune regulation against pathogens. Does the knockdown of PI3KC3 component genes affect the sperm plasma membrane breakdown? Do Rubicon and other factors localize to the sperm plasma membrane in the egg cytoplasm? It was reported that Sneaky is required for the breakdown of sperm plasma membrane. When sneaky is knocked down, is the paternal mitochondria elimination affected?

4) The authors previously showed that PME is attenuated and the MVBs are deformed due to improper biogenesis of the ILVs in atg7 mutant eggs. In this manuscript, the authors showed that efficient maternal knockdowns of *Drosophila* atg5, atg12, and atg16 homologs caused only weak PME attenuation. From these results, it is possible that ATG7 may have a distinct function in the PME independently of the ATG8-conjugation system. The authors need to explain why the phenotypes caused by the knockdown of these genes are different in PME.

5) The authors showed that a LAP-specific PI3K complex is required for the initial step of PME. In canonical autophagy, the PI3KC3 complex contains the subunits ATG14L and AMBRA1 that are dispensable for LAP. It would be better to examine whether PME is not affected by shRNA of ATG14L or AMBRA1 to confirm the involvement of the LAP-specific PI3KC3 complex in PME.

6) The authors showed that egg MVBs are targeted around the sperm flagellum and paternal mitochondria and form FVSs. What is the targeting mechanism of egg MVBs to sperm flagellum and paternal mitochondria? Is egg MVBs targeting to sperm organelles affected by the knockdown of Rubicon and Duox?

7) The authors showed that maternal knockdown of Atg8a did not significantly affect PME. It should be examined whether a simultaneous knockdown or knockout of ATG8a and ATG8b blocks PME. If not, what is the physiological function of Atg8 proteins in PME? Do Atg8a-positive structures

contain paternal mitochondria fragments? Do Atg8a-positive structures directly fuse with lysosomes?

8) The authors showed that immunostaining of mouse and bull sperm cells with an anti-human RUBICON antibody labeled the entire flagellum and even head region (not specific for mitochondria). However, these results do not reflect the localization of RUBICON in fertilized eggs. Since sperm contain minimal cytosol, this data may just show that RUBICON exists in cytosol of sperm cell. The authors should conduct immunostaining of mouse and bull-fertilized eggs with this antibody. In addition, absorption control, which is the incubation of the primary antibody with the antigen used to generate the antibody, is necessary for this experiment. Since Rubicon KO mice is fertile (Yamamuro et al. 2021 PLOS Genetics), sperms derived these mice would be a good negative control for immunostaining of Rubicon in mouse sperm and fertilized eggs.

Minor comments

1) Line 66, Line 344

P62 should be changed to p62.

2) line 99 and Fig. 1d

Vps15 should be included in a LAP-specific PI3 kinase complex.

3) Lines 143-145

“all transgenes and RNAi lines used in this study are under the control of the maternal promoters UASp or UASz.” should be changed to “all transgenes and RNAi lines used in this study are under the control of the maternal promoters UASp or UASz except for transgenes labeling PM.”

4) Fig 5B: It is difficult to see what the arrowheads indicate, so an enlarged figure should be produced and shown with the arrowheads.

5) Fig 5c: “atg8ShR” should be changed to “atg8aShR”.

6) Line 343

“Although the involvement of classical autophagy was proposed to be dispensable for PME in mice” would be changed to “Although the involvement of classical autophagy was controversial for PME in mice”.

Reviewer #2 (Remarks to the Author):

Mitochondria are generally considered to be maternally-inherited, but how paternal mitochondria (PM) are eliminated in the embryo has not yet been resolved. This paper extends a prior study from this group that showed that extracellular multivesicular bodies (MVB) in the egg target PM for degradation and recycle the remains. Here the authors used imaging and proteomics to show that some of the egg's MVB contain components of the LAP phagocytosis pathway, and that mutation or knockdown of some LAP components decreased elimination of PMs. The extent of interference in PM elimination differed for different components. The authors also showed that MVBs that contain LAP components fuse with the sperm, and that some of the ROS that is involved in PM elimination comes from the egg's Duox. They use imaging in wildtype and in knockdowns to order some of the steps in MVB's association with sperm and PM elimination. Finally, they show that mouse and bovine sperm immunostain for Rubicon, and suggest that the model they generate for *Drosophila* may also apply to mammals.

This was a difficult paper to assess, because it had great strengths, but also significant weaknesses.

Strengths were that the question is important and pressing, the experiments were designed well, the data were clear and convincing, and the authors' comprehensive cell biological and genetic interrogation of the LAP pathway showed that many of its members have roles in degrading paternally-derived sperm mitochondria. The encapsulation of distinct sections of the sperm tail in sheath-like groups of MVBs is also intriguing. The authors' case for doing this study with *Drosophila* was compelling, and they used marked and functional proteins and genetic tools excellently.

Weaknesses were that multiple previous studies (cited by the authors) already indicate that certain autophagy/phagocytosis pathways are implicated in PME, and the authors' previous paper (ref 53) already showed a role for MVBs in degradation of PM, association of Atg8a, and suggested involvement of the endocytic/autophagy pathways. Also, the results in the present manuscript are largely correlative, being consistent with the authors' model but not conclusively proving it. In multiple cases the effects were partial, making the interpretations suggestive but not firm. The authors were careful and honest about this in their writing, but one is left not fully convinced that their conclusions are unambiguous.

Specific concerns:

The *Drosophila*, sperm with membrane and acrosome intact, enters the egg. The sperm membrane must be dissolved before the PM is accessible to MVBs. Are the immunofluorescence signal locations precise enough to be sure that what the authors see is fusion of MVBs with PMs, followed by PM degradation? Or could MVBs be fusing with the sperm membrane and breaking it, then exposing the PMs to maternal cytoplasm, which degrades them? Is there something special about PM mitochondria that makes them susceptible to this degradation, or could some maternal mitochondria also be degraded by this system? There are many Rubicon-positive vesicles in the egg cytoplasm. If mitochondrial degradation is a general paradigm in eggs, the interpretations related to

immune defense are less supported. Can the authors confirm degradation or loss of the PMs by methods other immunofluorescence methods, such as loss of PM DNA?

Line 183 notes that rubicon mutants are viable and fertile, though no data were provided. Is this true for females as well as males? If so, it indicates that either Rubicon-mediated processes are not essential for PM degradation, or PM degradation is not necessary for normal embryogenesis. Either of these is important to establish and affects interpretation of the authors' model. Related to this concern, in lines 197-197 the authors note that early embryo development is normal in such embryos. But then they note that they did not examine "the embryos that failed to develop", leading one to wonder whether they missed an important embryo population. Related, interpretation of the significance the rubicon's mutants' partial effects on PME depends on whether they are null alleles or retain partial functionality. Can the authors clarify?

While consistent with the authors' data, the model needs stronger support regarding its timeline. For example, while the section about Rubicon²Atg8a shows this order convincingly, the involvement and timing of ROS and PtdIns(3)P is less clear.

Atg8a doesn't affect PM degradation, but is still used in the process. Perhaps Atg8b provides redundant function. The authors note that Atg8b exists, but didn't test it.

It would be useful to stain sperm in males with for H2DCFDA to rule out that ROS was not already present on sperm before fertilization.

Fig. 3, and the text around line 204, is convincing that Rubicon-containing vesicles associate with sperm. Do the larger vesicles described in line 208 also contain Rubicon? Do all Rubicon vesicles in the vicinity of the sperm associate with the sperm?

While interesting, the mammalian data belong in a different paper. In the Extended Data of this paper, the authors show Rubicon staining on mammalian sperm only. This is different from their model in which maternally-supplied Rubicon plays a role in PM elimination. And, staining does not on its own indicate function. More work is needed to convince readers that the model that the authors posit for *Drosophila* applies to mammals.

Minor:

Lines 55-58, it was confusing to read details about degradation of mtDNA, since that process is not addressed in this paper.

Line 231 should read: dually labeled vesicles revealed that the two tagged proteins occupy different compartments

How many embryos were needed for MVB purification?

Please note which images were subject to deconvolution. Can raw images be provided as Extended Data?

Reviewer #3 (Remarks to the Author):

OVERVIEW

Although mitochondria are maternally inherited, the mechanisms that destroy paternal mitochondria are poorly understood. The authors previously showed (Politi et al., 2014) that in *Drosophila* embryos paternal mitochondria are rapidly eliminated following fertilization in a process that involves association of multivesicular bodies (MVBs) with the sperm tail and ultimate breakdown of the mitochondrial derivative. In addition, they demonstrated that mitochondrial destruction (PMD) involves ubiquitination and relies on functional autophagic and endocytic pathways, as well as the autophagy receptor p62.

Here, the authors provide evidence that the pathway for paternal mitochondrial degradation (which they have renamed paternal mitochondrial elimination, PME) resembles a more recently described pathway, LC3-associated phagocytosis (LAP), which is typically employed to destroy invading microbes. They show that MVBs form extended flagellum vesicular sheaths (FVSs) that coat the sperm tail and promote PME. They find that both Rubicon, a subunit of the LAP-specific class III PI3K complex, and the LC3/Atg8 conjugation machinery are required for recruitment of Atg8 to FVSs. They show that PtdIns(3)P and ROS production also occur during this process and provide evidence that they argue suggests a similar pathway might occur in mammals.

Overall, the *Drosophila* experiments are nicely done, the videos are convincing, and the topic will be of interest to readers who care about fertilization, organelle inheritance and autophagy-related processes. Nonetheless, the experiments using mammalian sperm are insufficient to support their claim that the phenomenon they are studying is likely conserved from *Drosophila* to mammals. Moreover, there are a number of issues with the writing that will need to be corrected prior to publication.

MAJOR COMMENTS

1. By far the weakest part of the manuscript is the section that the authors claim supports the idea of a LAP-like pathway mediating PME in mammals. The authors “immunostained spermatozoa from mouse cauda epididymis and from bull semen with two anti-Rubicon antibodies, raised against different human RUBICON epitopes” and pre-incubated samples with mitotracker to detect paternal mitochondria. Although the antibodies clearly label the sperm tails of both mouse and bull sperm, and the secondary antibodies have no signal on their own, there are several strong caveats to the experiment as performed.

First, the authors did not use any methods to deplete Rubicon from the sperm to show that the staining was specific. It could be that the primary antibodies are just sticky and will bind to sperm tails without specifically binding to Rubicon. Ways to demonstrate specificity would preferably include using a mouse Rubicon null mutant (if feasible) to demonstrate that the staining goes away or, alternatively, pre-incubating the antibodies with the Rubicon peptides that were used as immunogens to confirm that each specific immunogen (but not the other immunogen) blocks binding of the corresponding antibodies to the sperm tails. This would provide support for the idea that Rubicon is present on the sperm tails prior to fertilization.

Second, and more important, these experiments, which were performed on mammalian sperm in the absence of oocytes/fertilization, are in no way comparable to the experiments performed in *Drosophila* embryos, where maternally provided Rubicon (Rubicon-eGFP) was observed to coat flagellar segments and participate in PME. In the absence of significantly more compelling data, for example, a genetic experiment showing a requirement for maternal (or potentially paternal) Rubicon in PME in mice, I strongly recommend dialing back on the authors’ claim, as stated in the Abstract, that their data “provide evidence that a similarly pathway might also mediate PME in mammals”.

2. It would help the reader if the authors would spell out more clearly what precisely they showed in their previous manuscript (Politi et al., *Dev Cell* 2014). For example, they previously uncovered a role for *Drosophila* Uvrag (part of the same LAP-associated PI3K complex as Rubicon) in PME, but this was not evident from the description of their previous results. It would better set up the context to more clearly indicate what was known/ not known prior to the experiments described in the current manuscript.

3. The authors use a large number of abbreviations, including some that are non-intuitive. For example, in a manuscript describing membrane trafficking through different cellular compartments, it seems a bit odd to use PM to refer to paternal mitochondria rather than the plasma membrane. It is unclear why the authors didn’t just use MD (mitochondrial derivative), as they did in their previous paper. Another abbreviation that I recommend avoiding is FVS (flagellum vesicular sheath). Like PM for paternal mitochondria, this is not a standard abbreviation and made it difficult, at least for this reader, to follow the meaning of the text. Other abbreviations (PME for paternal mitochondrial elimination, MVBs for multivesicular bodies, LAP for LC3-associated phagocytosis) were much easier to remember, in part because they are in wider use.

4. The authors state (lines 182-183) that “Flies homozygous for the rubicon mutant alleles are viable and fertile.” If so, does this mean that PME is not important for embryogenesis? Or is there an alternative pathway that can destroy the mitochondrial derivative in the absence of rubicon? This is an important point that should be made clear in the text.

MINOR CORRECTIONS TO THE TEXT

Line 63: rather than “mice cross”, this should read “mouse crosses”

Line 64: ref. 37 does not appear to refer to an interspecific mouse cross

Line 114: for clarity, add a comma after “rescue transgene”

Line 131: the marker shown in magenta in Supplementary Video 1 should be described here or labeled in an informative way in the video (the label “PM” is confusing and suggests that the sperm tail plasma membrane is labeled)

Line 137: this could be the start of a new paragraph, as it provides the rationale for the first set of experiments

Line 149: delete comma after “sperm”

Lines 159-160: it would help to provide (either in the Results or in the Extended Data Figure) examples of the proteins associated with innate immunity and phagocytosis that were identified in the proteomics experiments

Lines 175-176: it would help to define and remind the reader what is in “the autophagy pre-initiation complex”

Line 177: suggest replacing “In order to” with “To”

Lines 194-195: suggest deleting “It is important to note that” and starting the sentence with “To”

Line 194: delete comma after “cellularization”

Line 219: delete comma after parenthesis

Lines 228-229: the statement, “essentially all the Rubicon-eGFP positive vesicles were also positive for hCD63-tdTomato, including large vesicles associated with the flagellum” requires quantitation of the degree of correlation and overlap (Pearson’s correlation and Manders’ overlap coefficients)

Lines 245-247: similarly, the claim that “a subset of [PtdIns(3)P positive] vesicles were also positive for Rubicon-eGFP” also requires quantitation; this would clarify the degree of correlation and overlap for the reader

Lines 250-252: ref. 75 describes PtdIns(3)P positive endosomes that accumulate yolk proteins during oogenesis; do these persist into embryonic stages?

Line 257: has the ROS indicator H2DCFDA been used previously in Drosophila? If so, please provide a reference; if not, the indicator needs to be validated in some way

Line 263: replace “Extended Data Fig. 3e,f” with “Extended Data Fig. 3f,g” (unless change order of panels)

Lines 261-264: does knockdown of duox lead to reduced levels of ROS?

Line 276: suggest replacing “Consistently” with “Consistent with this”

Line 277: replace “Extended Data Fig. 3g” with “Extended Data Fig. 3e” (unless change order of panels)

Line 303: delete comma after “MVBs and insert before “therefore”

Line 322: the effect of rubicon knockdown on Atg8a accumulation is not very significant; is the effect stronger in rubicon mutants?

Line 324: replace “which” with “that”

Line 325: for clarity, replace “that” with “and” and change “flagellum, both require” to “ flagellum require both”

Line 363: delete comma after “membrane”

Line 378: are the “extended FVs that encapsulate large flagellar segments” single or double membrane structures? Is this evident from TEM images? it would help to clarify this for the reader, as the topology relative to the plasma membrane of the sperm tail is not obvious

Line 396: replace “deliver” with “delivers”

Line 408: suggest deleting “in terms of unusual size, anatomy and structure,” as this is awkward and uninformative

Line 1117: replace “transgenes” with “proteins”

Lines 1255 and 1263: spell out the paternal mitochondrial marker (red-PM) that was used in Videos 1 and 2

Lines 1272, 1289, 1297-1298: similarly, spell out the paternal mitochondrial marker (green-PM) that was used in Videos 3, 5 and 6

COMMENTS ON THE FIGURES

Fig. 4c: this experiment lacks a negative control; is there a genetic condition or chemical treatment that could prevent or deplete egg-generated ROS?

Fig. 4e: does knockdown of duox or nox affect ROS levels as measured by the indicator used in panel c?

Fig. 2b: why does the magenta signal appear on the outside of the embryos starting at 1 hour AEL (this appears to happen to at least some extent in all samples)?

Fig. 3e-g: recommend changing the order of the panels to correspond to the order described in the text (move current Fig. 3f,g to the left of current Fig. 3e)

REVIEWER COMMENTS

Reviewer #1 (Remarks to the Author):

In this manuscript, the authors reported that paternal mitochondria are eliminated by a LC3-associated phagocytosis (LAP)-related mechanism in *Drosophila* embryos.

The authors have demonstrated that paternal mitochondria are surrounded by egg multivesicular bodies (MVBs) in embryos and degraded within 1 hour after egg laying. In this process, egg-derived MVBs associate with the sperm flagellum and drive the segmentation of the sperm flagellum. The authors have also revealed that LAP-specific class III PI(3)K complex (PI3KC3) components including Rubicon localize to the limiting membrane of egg MVBs, which surround the sperm flagellum and are required for the recruitment of Atg8/LC3 to extended flagellum vesicular sheaths (FVSs) and the degradation of paternal mitochondria during embryogenesis. Furthermore, they have shown that the production of PI(3)P by PI3KC3 and ROS by Duox, respectively, is required for the paternal mitochondria elimination (PME). Finally, the authors have shown that an anti-human RUBICON antibody labeled mouse and bull sperm midpiece including flagellum and mitochondria, suggesting that a LAP-like pathway mediates PME in mouse and bull embryos. These results raise a possibility that the LAP-like pathway is required for PME not only in fly but also in mammalian embryos. This paper contains some interesting results, but several points are not sufficiently supported and additional data or further explanation is necessary.

Major concerns:

1) The authors showed that paternal mitochondria are eliminated by a LAP-like mechanism in *Drosophila* eggs. LAP was originally identified as a system for immune regulation against pathogens. In contrast to mammals, the *Drosophila* sperm cell penetrates the egg with its plasma membrane through the micropyle. The sperm plasma membrane is then broken down and paternal mitochondria are actively destroyed in the egg cytoplasm. It would be good to analyze the process of sperm plasma membrane disruption as well as paternal mitochondrial elimination according to the time series by live-imaging analysis.

Indeed, in many insects, including *Drosophila*, the sperm penetrates the egg through a special structure in the eggshell called the micropyle. Reportedly, when it enters the egg, the sperm is

still covered by its plasma membrane¹⁻³. However, how the sperm plasma membrane breaks down after fertilization is unclear.

Relevant to this study, since the plasma membrane coats the MD (and the axoneme), the anatomical details of the MVBs engagement with the MD remained largely elusive. To address this issue, we first generated a new transgenic fly line, *dj-CD8-Venus*, which labels the sperm plasma membrane. This transgene is composed of the mouse CD8 transmembrane domain fused to a Venus fluorescent protein [and expressed under the control of the promoter and regulatory sequences of the late spermatid gene *don-juan (dj)*].

Using this transgene, we monitored the fate of the sperm plasma membrane after fertilization both in super-resolution (using an expansion microscopy protocol) and by live imaging. This data appears in **new Fig. 9a-e** and **new Supplementary Video 10**. Concisely, we show that the Rubicon positive MVBs engage the sperm plasma membrane soon after fertilization, forming FVS that enwrap the entire sperm flagellum (with its plasma membrane and the two main organelles, the MD and the axoneme). In addition, we also show that sperm plasma membrane breakdown requires Rubicon, and that this process is faster (within 8-10 min AEL) than degradation of the MD (within 40-60 min AEL).

Also related, we discovered that the FVS contain at least two separate compartments: A degradative compartment, in which the sperm plasma membrane breaks down and the MD is degraded, and a “non-degradative” compartment, in a form of a narrow vesicular tube, that specifically enwraps around the axoneme.

The authors showed that egg MVBs cover the sperm flagellum and paternal mitochondria and form FVSs. However, the location of egg MVBs, sperm flagellum, and paternal mitochondria is unclear in Fig.1b. If possible, ultrastructural analysis by immunoelectron microscopy should be performed to analyze how egg MVBs are positioned relative to the sperm-derived organelle.

Using an expansion microscopy protocol to achieve super-resolution imaging, we now clearly demonstrate that the MVBs densely coat the flagellum, forming vesicular sheaths that fully enwrap the flagellum, including the MD (**new Fig. 3e,f** and **new Supplementary Video 3**).

Furthermore, we added three electron micrographs which demonstrate the engagement of the MVBs with the sperm flagellum (**new Extended Data Fig. 1a-c**).

Our reply to the previous comment of this Reviewer is also highly relevant for the detailed anatomy of the PME process.

2) It is difficult to discern the subcellular localization of each protein due to the low resolution of the images shown in this paper. Although the authors claimed that small vesicles around paternal mitochondrial form extended flagellum vesicular sheaths (FVSs), the presented data is not enough to support this conclusion. For example, it seems that Rubicon directly localizes to the whole region of the paternal mitochondrial membrane as well as large MVBs containing PM pieces. Does Rubicon directly bind to paternal mitochondria membrane as well as MVBs or indirectly bind to PM via MVBs? Super-resolution microscopy or EM would be necessary to distinguish these possibilities. At least, the authors should provide images with higher magnification.

Please refer to our replies to the two previous comments.

2.1) In addition, PI(3)P also seems to distribute throughout the paternal mitochondrial membrane. If so, how does PI3KC3 localize to the paternal mitochondrial membrane and produce PI(3)P.

When Rubicon is knocked down, is the signal of PI(3)P abolished on MVBs and the paternal mitochondrial membrane?

Our data show that Rubicon, a main component of the LAP-specific PI3KC3, resides on the FVS, implying that the activity of the Rubicon PI3KC3 originates on the FVS. Using live imaging to monitor the generation of PtdIns(3)P and Rubicon, we show close correlation between the appearance of Rubicon on the FVS and the generation of flagellar PtdIns(3)P.

However, we could not uncouple between the two events by live imaging, suggesting that they occur almost simultaneously (**new Supplementary Video 5**). We therefore, as suggested by the reviewer, quantified the number of early fertilized eggs displaying flagellar PtdIns(3)P in WT versus maternal *rubicon* knockdown eggs. **New Fig. 5c,d** now indicates that Rubicon is absolutely required for production of flagellar PtdIns(3)P, as essentially all *rubicon* knockdown eggs exhibited no flagellar PtdIns(3)P, as compared to ~50% of WT eggs that exhibited flagellar PtdIns(3)P.

3) LAP was originally identified as system for immune regulation against pathogens. Does the knockdown of PI3KC3 component genes affect the sperm plasma membrane breakdown?

Please see our reply to the first point of this reviewer. Concisely, we now show that knockdown of *rubicon* significantly attenuates sperm plasma membrane breakdown after fertilization.

Do Rubicon and other factors localize to the sperm plasma membrane in the egg cytoplasm?

Please see our replies to the first two points of this reviewer. Concisely, we now show that Rubicon resides on the FVS that entirely enwrap the sperm flagellum with its sperm plasma membrane.

It was reported that Sneaky is required for the breakdown of sperm plasma membrane. When sneaky is knocked down, is the paternal mitochondria elimination affected?

As suggested by the reviewer, we monitored the fate of the MD in eggs fertilized by *sneaky* knockdown sperm. As previously reported, these fertilized eggs failed to form viable zygotes. We observed that Rubicon-positive vesicular sheaths are formed around and degrade the *sneaky* knockdown sperm MD. Furthermore, the MD of *sneaky* knockdown sperm is degraded after fertilization at a normal rate.

However, to avoid diverging from the focus of the paper, we decided not to include this data in the manuscript, but to present it here for the reviewers' eyes (**Figure R1**).

4) The authors previously showed that PME is attenuated and the MVBs are deformed due to improper biogenesis of the ILVs in *atg7* mutant eggs. In this manuscript, the authors showed that efficient maternal knockdowns of *Drosophila atg5*, *atg12*, and *atg16* homologs caused only weak PME attenuation. From these results, it is possible that ATG7 may have a distinct function in the PME independently of the ATG8-conjugation system. The authors need to explain why the phenotypes caused by the knockdown of these genes are different in PME.

Indeed, although the Atg5-Atg12-Atg16 complex is required for conventional autophagy⁴, and is reported to also participate in some LAP paradigms⁵⁻⁸, maternal knockdown of these genes had no effect on PME (Extended Data Fig. 5f-k; and Extended Data Fig. 4i-k). In contrast, *Atg7* and

Rubicon are both involved in MVB biogenesis and/or recruitment to the flagellum (previous study⁹ and **new Fig. 4d-f**, respectively), as well as in recruitment of Atg8 (Fig. 7). Interestingly, we now also show that double knockdown of *atg8a* and *atg8b*, which as single knockdowns do not affect PME, significantly attenuates PME (**new Fig. 8d-j and Extended Data Fig. 4l-n**). Taken together, we conclude that the Atg5-Atg12-Atg16 complex is likely dispensable for MVB biogenesis/recruitment to the flagellum and for the LAP-like pathway of Atg8 recruitment to the FVS. It is noteworthy that other mechanisms of Atg8 recruitment, in a manner independent of the Atg5-Atg12-Atg16 complex, have also been documented¹⁰⁻¹³. These points are now discussed in the *Discussion* section of the paper.

5) The authors showed that a LAP-specific PI3K complex is required for the initial step of PME. In canonical autophagy, the PI3KC3 complex contains the subunits ATG14L and AMBRA1 that are dispensable for LAP. It would be better to examine whether PME is not affected by shRNA of ATG14L or AMBRA1 to confirm the involvement of the LAP-specific PI3KC3 complex in PME.

Drosophila contains a single *Atg14* gene but no AMBRA1 ortholog¹⁴. We now show that knockdown of *atg14* does not affect PME, further fortifying the findings that PME is mediated by the LAP-specific, but not the autophagy PI3KC3 complex (**new Fig. 2j,k and new Extended Data Fig. 4e**).

6) The authors showed that egg MVBs are targeted around the sperm flagellum and paternal mitochondria and form FVSs. What is the targeting mechanism of egg MVBs to sperm flagellum and paternal mitochondria?

The *Discussion* section now contains the following paragraph that summarizes the known data and the consequent hypotheses:

“Since the egg MVBs first engage the sperm flagellum plasma membrane, it is likely that they recognize specific lipid/s or protein/s placed on the exterior side of the sperm plasma membrane. Breakdown of the sperm flagellum plasma membrane exposes the two main flagellar organelles, the MD and the axoneme, to the luminal contents of the FVS. However, although this leads to degradation of the MD, the axoneme is spared from a similar fate. In this study we also observed that the two organelles reside in separate compartments within the FVS, suggesting the existence of active mechanisms that direct the MD, but not the axoneme, to a degradative compartment. Interestingly, cargo ubiquitination is a major MVB sorting mechanism¹⁵⁻¹⁷, and we previously demonstrated that the MD, but not the axoneme, is specifically decorated with lysine 63-linked ubiquitin chains after fertilization⁹, which may explain the different fates of these two organelles.”

Is egg MVBs targeting to sperm organelles affected by the knockdown of Rubicon and Duox?

We tested whether engagement of the egg MVBs with the sperm flagellum is affected by maternal *rubicon* knockdown, using the *hCD63-eGFP* transgene that localizes to the ILV compartment of the MVBs. In **new Fig. 4d-f**, we show that about 20% of the early fertilized eggs with *rubicon* knockdown display no targeting of the MVBs to the sperm flagellum at all. Furthermore, the flagellar hCD63-eGFP fluorescent signal was dramatically reduced in the eggs that display flagellar MVBs, indicating severe impairment in either the biogenesis of the ILV compartment of the MVBs or in the recruitment of the MVBs to the flagellum. Note that this is

also consistent with our previous report, showing that maternal inactivation of Atg7 and Uvrag results in MVBs that largely lack ILVs⁹. See also our reply to point 4 of this reviewer.

7) The authors showed that maternal knockdown of Atg8a did not significantly affect PME. It should be examined whether a simultaneous knockdown or knockout of ATG8a and ATG8b blocks PME. If not, what is the physiological function of Atg8 proteins in PME?

We thank the reviewer for the excellent question. We now tested *atg8b* knockdown and *atg8a/atg8b* double knockdown for possible effect on PME. Similar to maternal *atg8a* knockdown, maternal knockdown of *atg8b* also showed no effect on PME. Interestingly, however, a significant PME attenuation was detected in the double knockdown (**new Fig. 8d-j**). Consistently, we show that maternal *atg8b* mRNA expression is elevated upon maternal knockdown of *atg8a*, revealing a maternal compensatory mechanism between the two paralogs (**new Extended Data Fig. 4l-n**).

Do Atg8a-positive structures contain paternal mitochondria fragments?

Do Atg8a-positive structures directly fuse with lysosomes?

Indeed, MD fragments are readily detected inside the Atg8a positive string of vesicles derived from the FVS (**Fig. 6A, IIa**). Furthermore, using the lysosomal transgenic marker, *GFP-LAMP1*, we now also show that these vesicles contain lysosomal contents, indicating that lysosomes fuse with the string of vesicles (**new Fig. 8a-c** and **new Supplementary Video 9**).

8) The authors showed that immunostaining of mouse and bull sperm cells with an anti-human RUBICON antibody labeled the entire flagellum and even head region (not specific for mitochondria). However, these results do not reflect the localization of RUBICON in fertilized eggs. Since sperm contain minimal cytosol, this data may just show that RUBICON exists in cytosol of sperm cell. The authors should conduct immunostaining of mouse and bull-fertilized eggs with this antibody. In addition, absorption control, which is the incubation of the primary antibody with the antigen used to generate the antibody, is necessary for this experiment. Since Rubicon KO mice is fertile (Yamamuro et al. 2021 PLOS Genetics), sperms derived these mice would be a good negative control for immunostaining of Rubicon in mouse sperm and fertilized eggs.

Per Reviewer #2's suggestion, we decided to remove the data concerning the mammalian sperm staining altogether (previous Extended Data Fig. 7). We agree that this is too preliminary to derive firm conclusions about PME in mammals.

Minor comments

1) Line 66, Line 344: P62 should be changed to p62.

Done.

2) line 99 and Fig. 1d: Vps15 should be included in a LAP-specific PI3 kinase complex.

We added Vps15 to the schematic model and in the relevant text.

3) Lines 143-145: "all transgenes and RNAi lines used in this study are under the control of the maternal promoters UASp or UASz." should be changed to "all transgenes and RNAi lines used in this study are under the control of the maternal promoters UASp or UASz except for transgenes labeling PM."

Done.

4) Fig 5B: It is difficult to see what the arrowheads indicate, so an enlarged figure should be produced and shown with the arrowheads.

The arrowheads were replaced with dashed line rectangles and the confined areas were enlarged (now Fig. 6b-i).

5) Fig 5c: “atg8ShR” should be changed to “atg8aShR”.

Done (now Fig. 7e).

6) Line 343: “Although the involvement of classical autophagy was proposed to be dispensable for PME in mice” would be changed to “Although the involvement of classical autophagy was controversial for PME in mice”.

This section was removed altogether.

Reviewer #2 (Remarks to the Author):

Mitochondria are generally considered to be maternally-inherited, but how paternal mitochondria (PM) are eliminated in the embryo has not yet been resolved. This paper extends a prior study from this group that showed that extracellular multivesicular bodies (MVB) in the egg target PM for degradation and recycle the remains. Here the authors used imaging and proteomics to show that that some of the egg’s MVB contain components of the LAP phagocytosis pathway, and that mutation or knockdown of some LAP components decreased elimination of PMs. The extent of interference in PM elimination differed for different components. The authors also showed that MVBs that contain LAP components fuse with the sperm, and that some of the ROS that is involved in PM elimination comes from the egg’s Duox. They use imaging in wildtype and in knockdowns to order some of the steps in MVB’s association with sperm and PM elimination. Finally, they show that mouse and bovine sperm immunostain for Rubicon, and suggest that the model they generate for *Drosophila* may also apply to mammals.

This was a difficult paper to assess, because it had great strengths, but also significant weaknesses.

Strengths were that the question is important and pressing, the experiments were designed well, the data were clear and convincing, and the authors’ comprehensive cell biological and genetic interrogation of the LAP pathway showed that many of its members have roles in degrading paternally-derived sperm mitochondria. The encapsulation of distinct sections of the sperm tail in sheath-like groups of MVBs is also intriguing. The authors’ case for doing this study with *Drosophila* was compelling, and they used marked and functional proteins and genetic tools excellently.

Weaknesses were that multiple previous studies (cited by the authors) already indicate that certain autophagy/phagocytosis pathways are implicated in PME, and the authors’ previous paper (ref 53) already showed a role for MVBs in degradation of PM, association of Atg8a, and suggested involvement of the endocytic/autophagy pathways. Also, the results in the present manuscript are largely correlative, being consistent with the authors’ model but not conclusively proving it. In multiple cases the effects were partial, making the interpretations suggestive but not firm. The authors were careful and honest about this in their writing, but one is left not fully convinced that their conclusions are unambiguous.

Specific concerns:

1. The *Drosophila*, sperm with membrane and acrosome intact, enters the egg. The sperm membrane must be dissolved before the PM is accessible to MVBs. Are the immunofluorescence signal locations precise enough to be sure that what the authors see is fusion of MVBs with PMs, followed by PM degradation? Or could MVBs be fusing with the sperm membrane and breaking it, then exposing the PMs to maternal cytoplasm, which degrades them? Is there something special about PM mitochondria that makes them susceptible to this degradation, or could some maternal mitochondria also be degraded by this system? There are many Rubicon-positive vesicles in the egg cytoplasm. If mitochondrial degradation is a general paradigm in eggs, the interpretations related to immune defense are less supported.

Please refer to our replies to points 1 and 3, raised by Reviewer #1. We believe that our new data about the breakdown of the sperm plasma membrane after fertilization addresses these comments and concerns.

2. Can the authors confirm degradation or loss of the PMs by methods other immunofluorescence methods, such as loss of PM DNA?

The mtDNA cannot be used as a marker for degradation of the mitochondrial derivative because the mtDNA gets degraded already during late spermatogenesis stages, as previously reported¹⁸. Nevertheless, we now also used an expansion microscopy protocol which facilitated super-resolution imaging of the anatomy of this process (**new Fig. 3e,f, new Fig. 9a,f, and new Supplementary Video 3**). Together with our live-imaging analyses throughout the paper and electron micrographs in the current study and in the previous paper (**new Extended Data Fig. 1a-c**)⁹, we establish that specific degradation of the MD by egg-derived MVBs occurs soon after fertilization.

3. Line 183 notes that rubicon mutants are viable and fertile, though no data were provided. Is this true for females as well as males? If so, it indicates that either Rubicon-mediated processes are not essential for PM degradation, or PM degradation is not necessary for normal embryogenesis. Either of these is important to establish and affects interpretation of the authors' model.

To address these comments, we analyzed both female and male fertility in *rubicon* mutant and knockdown flies. In **new Extended Data Fig. 3a-d**, we show that flies homozygous for the *rubicon* mutant alleles are viable but display varying levels of female and male fertility. Whereas homozygous females for the four *rubicon* mutant alleles show a significant reduction in fertility, with *rubicon*^{A13} and *rubicon*^{A196} causing the most dramatic effect, a milder but still significant reduction in male fertility is only detected in flies homozygous for these two alleles. Accordingly, maternal but not male germline knockdown of *rubicon*, leads to a significant reduction in female fertility (albeit the effect is less pronounced than in the mutants) with no change in male fertility, respectively. However, whereas these findings indicate that maternal *rubicon* is required for normal development, this effect likely occurs after the cellularization stage, as both maternal *rubicon* mutants and knockdown progressed to the embryonic cellularization stage in a rate similar to their respective control embryos (**Extended Data Fig. 3e,f**).

Related to this concern, in lines 197-197 the authors note that early embryo development is normal in such embryos. But then they note that they did not examine “the embryos that failed to develop”, leading one to wonder whether they missed an important embryo population.

In our live imaging analyses, we omitted embryos that failed to develop in order to avoid possible indirect effects on PME due to developmental failure of the early embryos. These developmental failures may occur also due to the procedure that we use to prepare the embryos for analysis. Indeed, embryos that failed to develop were also detected in our control fly lines, and were similarly omitted from our analyses. Nevertheless, our reply to the previous point of this Reviewer also clarifies that maternal *rubicon* deficiency has no effect on early development of the fertilized eggs during the examined period.

Related, interpretation of the significance the *rubicon*'s mutants' partial effects on PME depends on whether they are null alleles or retain partial functionality. Can the authors clarify?

In addition to early frameshifting lesions found in all four *rubicon* alleles, we now also analyzed mRNA expression levels of these alleles in early fertilized eggs by reverse transcriptase PCR (RT-PCR). We show that two *rubicon* mutant alleles, *rubicon*^{A10} and *rubicon*^{A13}, display a significant reduction in the expression levels of maternal *rubicon* mRNA (with *rubicon*^{A13} essentially being an RNA null allele), whereas a third mutant allele, *rubicon*^{A196}, containing a deletion encompassing the first splice-acceptor site, generates a truncated mRNA lacking part of the translated 5' region (**new Extended Data Fig. 2b**). Nevertheless, all four *rubicon* mutant alleles, including the RNA null allele, *rubicon*^{A13}, displayed similar PME attenuation levels, suggesting that at least for the function of Rubicon in PME, these alleles are all functionally null.

4. While consistent with the authors' data, the model needs stronger support regarding its timeline. For example, while the section about Rubicon→Atg8a shows this order convincingly, the involvement and timing of ROS and PtdIns(3)P is less clear.
For the PtdIns(3)P, please refer to our reply to point 2.1 of Reviewer #1.
As for the comment about ROS, please refer to point 6 of this reviewer.
5. Atg8a doesn't affect PM degradation, but is still used in the process. Perhaps Atg8b provides redundant function. The authors note that Atg8b exists, but didn't test it.
We thank the reviewer for the excellent point. Indeed, we revealed a maternal compensation mechanism between the two paralogs. Please refer to our reply to point 7 of Reviewer #1.
6. It would be useful to stain sperm in males with for H2DCFDA to rule out that ROS was not already present on sperm before fertilization.
We thank the reviewer for this suggestion. We now stained mature sperm in the male sperm storage organs, the seminal vesicles, with H2DCFDA. Indeed, sperm flagellar ROS are readily detected already prior to fertilization in the seminal vesicles (**new Fig. 5i**).
7. Fig. 3, and the text around line 204, is convincing that Rubicon-containing vesicles associate with sperm. Do the larger vesicles described in line 208 also contain Rubicon? Do all Rubicon vesicles in the vicinity of the sperm associate with the sperm?
Using expansion microscopy, additional staining, and live imaging analyses, we were able to obtain high resolution images of the anatomy of the vesicles associated with PME. Importantly, our model now reliably discriminates between the MVBs that engage, densely coat, and form the FVS, and the strings of large (Atg8 positive) vesicles derived from the FVS, containing degrading MD fragments and lysosomal contents. Rubicon localizes on both vesicles, as the

strings of large vesicles derived from the FVS are derived from the Rubicon positive MVBs (new Fig. 3b,c,e,f, Fig. 6a, new Fig. 8b,c, new Fig. 9a,f, new Supplementary Video 3). We do not know if all Rubicon MVBs associate with the sperm flagellum.

8. While interesting, the mammalian data belong in a different paper. In the Extended Data of this paper, the authors show Rubicon staining on mammalian sperm only. This is different from their model in which maternally-supplied Rubicon plays a role in PM elimination. And, staining does not on its own indicate function. More work is needed to convince readers that the model that the authors posit for *Drosophila* applies to mammals.

As suggested by the Reviewer, we decided to remove the data concerning the mammalian sperm staining altogether (previous Extended Data Fig. 7). We agree that this is too preliminary to derive firm conclusions about PME in mammals.

Minor:

Lines 55-58, it was confusing to read details about degradation of mtDNA, since that process is not addressed in this paper.

To avoid confusion, we now moved most details about the sperm mtDNA from the *Introduction* to the *Discussion* section.

Line 231 should read: dually labeled vesicles revealed that the two tagged proteins occupy different compartments

Done.

How many embryos were needed for MVB purification?

We added this information to the *Methods* section. Concisely, for the small-scale isolation of the MVBs, we used 60 mg of early embryos, which amounts to nearly 1000 embryos. For large scale analysis, we used 2 gm of early embryos, amounting to nearly 20,000 embryos.

Please note which images were subject to deconvolution. Can raw images be provided as Extended Data?

This information now appears in the *Methods* section. The raw images themselves are found herein as **Figure R2** for the Reviewers. If required, we can also deposit this Figure in the 'figshare' repository.

Reviewer #3 (Remarks to the Author):

OVERVIEW

Although mitochondria are maternally inherited, the mechanisms that destroy paternal mitochondria are poorly understood. The authors previously showed (Politi et al., 2014) that in *Drosophila* embryos paternal mitochondria are rapidly eliminated following fertilization in a process that involves association of multivesicular bodies (MVBs) with the sperm tail and ultimate breakdown of the mitochondrial derivative. In addition, they demonstrated that mitochondrial destruction (PMD) involves ubiquitination and relies on functional autophagic and endocytic pathways, as well as the autophagy receptor p62.

Here, the authors provide evidence that the pathway for paternal mitochondrial degradation (which they have renamed paternal mitochondrial elimination, PME) resembles a more recently described pathway, LC3-associated phagocytosis (LAP), which is typically employed to destroy invading microbes. They show that MVBs form extended flagellum vesicular sheaths (FVSs)

that coat the sperm tail and promote PME. They find that both Rubicon, a subunit of the LAP-specific class III PI3K complex, and the LC3/Atg8 conjugation machinery are required for recruitment of Atg8 to FVSs. They show that PtdIns(3)P and ROS production also occur during this process and provide evidence that they argue suggests a similar pathway might occur in mammals.

Overall, the *Drosophila* experiments are nicely done, the videos are convincing, and the topic will be of interest to readers who care about fertilization, organelle inheritance and autophagy-related processes. Nonetheless, the experiments using mammalian sperm are insufficient to support their claim that the phenomenon they are studying is likely conserved from *Drosophila* to mammals. Moreover, there are a number of issues with the writing that will need to be corrected prior to publication.

MAJOR COMMENTS

1. By far the weakest part of the manuscript is the section that the authors claim supports the idea of a LAP-like pathway mediating PME in mammals. The authors “immunostained spermatozoa from mouse cauda epididymis and from bull semen with two anti-Rubicon antibodies, raised against different human RUBICON epitopes” and pre-incubated samples with mitotracker to detect paternal mitochondria. Although the antibodies clearly label the sperm tails of both mouse and bull sperm, and the secondary antibodies have no signal on their own, there are several strong caveats to the experiment as performed.

First, the authors did not use any methods to deplete Rubicon from the sperm to show that the staining was specific. It could be that the primary antibodies are just sticky and will bind to sperm tails without specifically binding to Rubicon. Ways to demonstrate specificity would preferably include using a mouse Rubicon null mutant (if feasible) to demonstrate that the staining goes away or, alternatively, pre-incubating the antibodies with the Rubicon peptides that were used as immunogens to confirm that each specific immunogen (but not the other immunogen) blocks binding of the corresponding antibodies to the sperm tails. This would provide support for the idea that Rubicon is present on the sperm tails prior to fertilization.

Second, and more important, these experiments, which were performed on mammalian sperm in the absence of oocytes/fertilization, are in no way comparable to the experiments performed in *Drosophila* embryos, where maternally provided Rubicon (Rubicon-eGFP) was observed to coat flagellar segments and participate in PME. In the absence of significantly more compelling data, for example, a genetic experiment showing a requirement for maternal (or potentially paternal) Rubicon in PME in mice, I strongly recommend dialing back on the authors’ claim, as stated in the Abstract, that their data “provide evidence that a similarly pathway might also mediate PME in mammals”.

Per Reviewer #2’s suggestion, we decided to remove the data concerning the mammalian sperm staining altogether (previous Extended Data Fig. 7). We agree that this is too preliminary to derive firm conclusions about PME in mammals.

2. It would help the reader if the authors would spell out more clearly what precisely they showed in their previous manuscript (Politi et al., *Dev Cell* 2014). For example, they previously uncovered a role for *Drosophila* Uvrag (part of the same LAP-associated PI3K complex as Rubicon) in PME, but this was not evident from the description of their previous results. It would better set up the context to more clearly indicate what was known/ not known prior to the experiments described in the current manuscript.

We thank the Reviewer for this comment. The *Introduction* section of the revised paper, as well as different paragraphs in the *Results* section, now contain all relevant findings from our previous paper⁹.

3. The authors use a large number of abbreviations, including some that are non-intuitive. For example, in a manuscript describing membrane trafficking through different cellular compartments, it seems a bit odd to use PM to refer to paternal mitochondria rather than the plasma membrane. It is unclear why the authors didn't just use MD (mitochondrial derivative), as they did in their previous paper. Another abbreviation that I recommend avoiding is FVS (flagellum vesicular sheath). Like PM for paternal mitochondria, this is not a standard abbreviation and made it difficult, at least for this reader, to follow the meaning of the text. Other abbreviations (PME for paternal mitochondrial elimination, MVBs for multivesicular bodies, LAP for LC3-associated phagocytosis) were much easier to remember, in part because they are in wider use.

We thank the Reviewer for turning our attention to this point. To avoid confusion, we now refer to the *Drosophila* paternal mitochondria as mitochondrial derivative (MD), and we do not abbreviate plasma membrane. We decided to leave the abbreviation of the flagellum vesicular sheaths (FVS), as this term is rather long, appearing multiple times in the text. The abbreviation is further justified by the fact that this is the first study to describe such structure.

4. The authors state (lines 182-183) that “Flies homozygous for the rubicon mutant alleles are viable and fertile.” If so, does this mean that PME is not important for embryogenesis? Or is there an alternative pathway that can destroy the mitochondrial derivative in the absence of rubicon? This is an important point that should be made clear in the text.

Please refer to our reply to point 3 of Reviewer #2.

MINOR CORRECTIONS TO THE TEXT

We very much thank the Reviewer for carefully reading the text and for the edits.

Line 63: rather than “mice cross”, this should read “mouse crosses”

Done.

Line 64: ref. 37 does not appear to refer to an interspecific mouse cross

Reference 37 (now reference 25) refers to the paper by Kaneda et al., 1995. In this paper, the authors examined both inter- and intra-specific mouse crosses.

Line 114: for clarity, add a comma after “rescue transgene”

Done

Line 131: the marker shown in magenta in Supplementary Video 1 should be described here or labeled in an informative way in the video (the label “PM” is confusing and suggests that the sperm tail plasma membrane is labeled)

The marker in this video is now described in the legend. Also, we changed the ‘PM’ to ‘MD’ as suggested by this reviewer.

Line 137: this could be the start of a new paragraph, as it provides the rationale for the first set of experiments

Done.

Line 149: delete comma after “sperm”

Done.

Lines 159-160: it would help to provide (either in the Results or in the Extended Data Figure) examples of the proteins associated with innate immunity and phagocytosis that were identified in the proteomics experiments

Examples of these proteins are added in the legend for Extended Data Fig. 1f.

Lines 175-176: it would help to define and remind the reader what is in “the autophagy pre-initiation complex”

Done.

Line 177: suggest replacing “In order to” with “To”

Done.

Lines 194-195: suggest deleting “It is important to note that” and starting the sentence with “To”

Done.

Line 194: delete comma after “cellularization”

Done.

Line 219: delete comma after parenthesis

Done.

Lines 228-229: the statement, “essentially all the Rubicon-eGFP positive vesicles were also positive for hCD63-tdTomato, including large vesicles associated with the flagellum” requires quantitation of the degree of correlation and overlap (Pearson’s correlation and Manders’ overlap coefficients)

The quantification is now added in **new Fig. 4c**, showing that 85% of the Rubicon-eGFP vesicles are also positive for hCD63-tdTomato. Note that although Rubicon and hCD63 are both localized to the MVBs, they reside in different compartments, with Rubicon localizing on the MVB limiting membrane and hCD63 residing in the intraluminal vesicles (ILVs). Since the signals do not overlap, we performed an object-based analysis (AI-based segmentation of the Rubicon vesicles) instead of pixel-based analysis. These analyses revealed high overlap (Manders’ overlap co-efficient) between Rubicon-eGFP and hCD63-tdTomato on the MVBs, as well as significant correlation between the intensities of the two proteins (Pearson’s correlation; this correlation was less significant than the Manders’ overlap).

Lines 245-247: similarly, the claim that “a subset of [PtdIns(3)P positive] vesicles were also positive for Rubicon-eGFP” also requires quantitation; this would clarify the degree of correlation and overlap for the reader

As in the previous point, this analysis requires segmentation of the vesicles. However, since the Rubicon vesicles aggregate upon expression of the PtdIns(3)P reporter, the AI-based segmentation failed to define individual vesicles, hindering overlap and correlation analyses of signals emanating from individual MVBs.

Lines 250-252: ref. 75 describes PtdIns(3)P positive endosomes that accumulate yolk proteins during oogenesis; do these persist into embryonic stages?

The mentioned reference describes yolk proteins that reside within endosomes during oogenesis. Evidence that the yolk granules in early embryos recruit late endosomal markers is reported in another paper¹⁹. We therefore revised the sentence as follows: “Note that at least some of the vesicles labeled by PtdIns(3)P alone may correspond to other endosomes, such as late endosomes that enwrap yolk granules during the cellularization stage¹⁹.”

Line 257: has the ROS indicator H2DCFDA been used previously in Drosophila? If so, please provide a reference; if not, the indicator needs to be validated in some way

The ROS indicator H2DCFDA has been used and reported in multiple papers using *Drosophila* tissues. We are now citing two of them^{20,21}.

Line 263: replace “Extended Data Fig. 3e,f” with “Extended Data Fig. 3f,g” (unless change order of panels)

Done.

Lines 261-264: does knockdown of *duox* lead to reduced levels of ROS?

We now show that mature sperm in the male sperm storage organs, the seminal vesicles, readily display flagellar ROS (new Fig. 5i), suggesting that at least part of the detected ROS after fertilization is present in the sperm flagellum prior to fertilization. This finding, as well as the fact that knockdown of *duox* only mildly affects PME, imply that knockdown of *duox* is only weakly affecting the overall H2DCFDA staining signal. Furthermore, we found that early fertilized eggs stained with H2DCFDA display variations in intensity levels of this indicator signal, likely due to penetration efficiency, further hindering reliable evaluation of possible effect of *duox* knockdown on sperm flagellum ROS levels.

Line 276: suggest replacing “Consistently” with “Consistent with this”

This paragraph was revised.

Line 277: replace “Extended Data Fig. 3g” with “Extended Data Fig. 3e” (unless change order of panels)

Done.

Line 303: delete comma after “MVBs and insert before “therefore”

This paragraph was revised.

Line 322: the effect of rubicon knockdown on Atg8a accumulation is not very significant; is the effect stronger in rubicon mutants?

We believe that the differences between *rubicon* knockdown and *atg7* and *atg8a* knockdowns stem from the facts that Rubicon promotes LAP but not conventional autophagy, whereas Atg7 and Atg8 promote both pathways. It is therefore plausible that our quantification of Atg8a accumulation in the vicinity of the sperm flagellum in the *rubicon* knockdown eggs also includes other, non-relevant, Atg8a positive vesicles, such as autophagosomes (that are not formed in the Atg7 and Atg8a knockdowns).

Line 324: replace “which” with “that”

This paragraph was revised.

Line 325: for clarity, replace “that” with “and” and change “flagellum, both require” to “flagellum require both”

This paragraph was revised.

Line 363: delete comma after “membrane”

Done.

Line 378: are the “extended FVSs that encapsulate large flagellar segments” single or double membrane structures? Is this evident from TEM images? it would help to clarify this for the reader, as the topology relative to the plasma membrane of the sperm tail is not obvious

Please refer to our reply to points 1 of Reviewer #1. Note that we now added super-resolution images that demonstrate that the FVS are derived from the MVBs that densely coat the flagellum, hence they are single membrane vesicles. Furthermore, these images clearly demonstrate that the FVS are formed around the sperm flagellum plasma membrane, mediating its breakdown.

Line 396: replace “deliver” with “delivers”

We considered changing this, but since Mitochondria is a plural word, we believe that “deliver”, which relates to this word, is the more correct verb.

Line 408: suggest deleting “in terms of unusual size, anatomy and structure,” as this is awkward and uninformative

Done

Line 1117: replace “transgenes” with “proteins”

This paragraph was revised.

Lines 1255 and 1263: spell out the paternal mitochondrial marker (red-PM) that was used in Videos 1 and 2

Done

Lines 1272, 1289, 1297-1298: similarly, spell out the paternal mitochondrial marker (green-PM) that was used in Videos 3, 5 and 6

Done

COMMENTS ON THE FIGURES

Fig. 4c: this experiment lacks a negative control; is there a genetic condition or chemical treatment that could prevent or deplete egg-generated ROS?

Although we do not have a negative control for this experiment, we provide herein a positive control experiment (**Figure R3** for the reviewers). The male seminal vesicles containing mature sperm cells were incubated with H₂O₂ before staining with H₂DCFDA. We demonstrate a significant increase in the fluorescent signal of this ROS indicator in mature sperm treated with H₂O₂, as compared with untreated sperm (which exhibited much lower levels of ROS).

Fig. 4e: does knockdown of *duox* or *nox* affect ROS levels as measured by the indicator used in panel c?

We now show that mature sperm in the male sperm storage organs, the seminal vesicles, readily display flagellar ROS (**new Fig. 5i**), suggesting that at least part of the detected ROS after fertilization is present in the sperm flagellum prior to fertilization. This finding, as well as the fact that knockdown of *duox* only mildly affects PME, imply that knockdown of *duox* is only weakly affecting the overall H₂DCFDA staining signal. Furthermore, we found that early fertilized eggs stained with H₂DCFDA display variations in intensity levels of this indicator signal, likely due to penetration efficiency, further hindering reliable evaluation of possible effect of *duox* knockdown on sperm flagellum ROS levels.

Fig. 2b: why does the magenta signal appear on the outside of the embryos starting at 1 hour AEL (this appears to happen to at least some extent in all samples)?

Indeed, we have noticed this phenomenon in embryos at the beginning of the cellularization stage, regardless of their genotypes. While we do not know what the exact source of this autofluorescence is, our analyses of PME kinetics takes this autofluorescence into account by eroding 15 pixels from the surface of each embryo, as described in the *Methods* section.

Fig. 3e-g: recommend changing the order of the panels to correspond to the order described in the text (move current Fig. 3f,g to the left of current Fig. 3e)

The figure was updated as suggested to correspond the order described in the text.

[REDACTED]

Figure R2: Original images without deconvolution corresponding to images appearing in the paper. a-d, The images correspond to the images in Figure 3a-d. **e,f,** The images correspond to the images in Figure 4a,b. **g,h,** The images correspond to the images in Figure 5a,b.

Figure R3: The H2DCFDA fluorescent signal dramatically increases upon treatment with H₂O₂. Shown are seminal vesicles of males producing red-MD sperm (magenta; expressing *dj-(MTS)tdTomato*), either untreated (**a**) or incubated with 10% of the strong oxidizing agent H₂O₂ for 5 minutes (**b**). Staining these seminal vesicles with the ROS indicator, H2DCFDA, reveals significant increase in the levels of flagellar ROS (green), indicating the specificity of this indicator. Scale bar, 20 μ m.

References

1. Loppin, B., Dubruille, R. & Horard, B. The intimate genetics of *Drosophila* fertilization. *Open Biol* **5**, (2015).
2. Wilson, K. L., Fitch, K. R., Bafus, B. T. & Wakimoto, B. T. Sperm plasma membrane breakdown during *Drosophila* fertilization requires Sneaky, an acrosomal membrane protein. *Development* **133**, 4871–4879 (2006).
3. PEROTTI, M. E. ULTRASTRUCTURAL ASPECTS OF FERTILIZATION IN DROSOPHILA. *The Functional Anatomy of the Spermatozoon* 57–68 (1975) doi:10.1016/B978-0-08-018006-9.50011-4.
4. Fujita, N., Itoh, T., Omori, H., Fukuda, M., Noda, T. & Yoshimori, T. The Atg16L complex specifies the site of LC3 lipidation for membrane biogenesis in autophagy. *Mol Biol Cell* **19**, 2092–2100 (2008).
5. Fletcher, K. *et al.* The WD40 domain of ATG16L1 is required for its non-canonical role in lipidation of LC3 at single membranes. *EMBO J* **37**, e97840 (2018).
6. Lystad, A. H., Carlsson, S. R., de la Ballina, L. R., Kauffman, K. J., Nag, S., Yoshimori, T., Melia, T. J. & Simonsen, A. Distinct functions of ATG16L1 isoforms in membrane binding and LC3B lipidation in autophagy-related processes. *Nature Cell Biology* **21**, 372–383 (2019).
7. Sanjuan, M. A. *et al.* Toll-like receptor signalling in macrophages links the autophagy pathway to phagocytosis. *Nature* **450**, 1253–1257 (2007).
8. Magné, J. & Green, D. R. LC3-associated endocytosis and the functions of Rubicon and ATG16L1. *Sci Adv* **8**, eabo5600 (2022).
9. Politi, Y., Gal, L., Kalifa, Y., Ravid, L., Elazar, Z. & Arama, E. Paternal mitochondrial destruction after fertilization is mediated by a common endocytic and autophagic pathway in *drosophila*. *Dev Cell* **29**, 305–320 (2014).
10. Jipa, A., Vedelek, V., Merényi, Z., Ürmösi, A., Takáts, S., Kovács, A. L., Horváth, G. V., Sinka, R. & Juhász, G. Analysis of *Drosophila* Atg8 proteins reveals multiple lipidation-independent roles. *Autophagy* **17**, 2565–2575 (2021).
11. Al-Younes, H. M., Al-Zeer, M. A., Khalil, H., Gussmann, J., Karlas, A., Machuy, N., Brinkmann, V., Braun, P. R. & Meyer, T. F. Autophagy-independent function of MAP-LC3 during intracellular propagation of *Chlamydia trachomatis*. *Autophagy* **7**, 814–828 (2011).
12. Leboutet, R. *et al.* LGG-1/GABARAP lipidation is not required for autophagy and development in *Caenorhabditis elegans*. *Elife* **12**, e85748 (2023).
13. Liu, X. M., Yamasaki, A., Du, X. M., Coffman, V. C., Ohsumi, Y., Nakatogawa, H., Wu, J. Q., Noda, N. N. & Du, L. L. Lipidation-independent vacuolar functions of Atg8 rely on its noncanonical interaction with a vacuole membrane protein. *Elife* **7**, (2018).
14. Demir, E. & Kacew, S. *Drosophila* as a Robust Model System for Assessing Autophagy: A Review. *Toxics* **11**, (2023).
15. Lauwers, E., Erpapazoglou, Z., Haguenaer-Tsapis, R. & André, B. The ubiquitin code of yeast permease trafficking. *Trends Cell Biol* **20**, 196–204 (2010).
16. Piper, R. C., Dikic, I. & Lukacs, G. L. Ubiquitin-Dependent Sorting in Endocytosis. *Cold Spring Harb Perspect Biol* **6**, a016808 (2014).
17. Sardana, R. & Emr, S. D. Membrane Protein Quality Control Mechanisms in the Endo-Lysosome System. *Trends Cell Biol* **31**, 269–283 (2021).
18. DeLuca, S. Z. & O’Farrell, P. H. Barriers to Male Transmission of Mitochondrial DNA in Sperm Development. *Dev Cell* **22**, 660–668 (2012).

19. Reed, S., Chen, W., Bergstein, V. & He, B. Toll-Dorsal signaling regulates the spatiotemporal dynamics of yolk granule tubulation during *Drosophila* cleavage. *Dev Biol* **481**, 64–74 (2022).
20. Owusu-Ansah, E., Yavari, A. & Banerjee, U. A protocol for in vivo detection of reactive oxygen species. *Protoc Exch* (2008) doi:10.1038/NPROT.2008.23.
21. Hunter, M. V., Willoughby, P. M., Bruce, A. E. E. & Fernandez-Gonzalez, R. Oxidative Stress Orchestrates Cell Polarity to Promote Embryonic Wound Healing. *Dev Cell* **47**, 377-387.e4 (2018).

REVIEWER COMMENTS

Reviewer #1 (Remarks to the Author):

In the revised manuscript, the authors addressed most my concerns as well as comments raised by other reviewers. The manuscript is nicely improved and seems more reader-friendly. I endorse the current version for publication.

Reviewer #2 (Remarks to the Author):

The authors present a model for degradation of the *Drosophila* paternal mitochondrial derivative (MD) by a LAP-like pathway. In their revision, they responded well to most of the comments, including by removing the mammalian work that had concerned several referees, and adding new data of excellent quality to give higher resolution views of the proteins' and structures' relative localizations. The added data, in combination with the original paper's, favor an interesting model with a fairly linear progression of events. The authors resolved the temporal sequence of many events. They determined that the sperm plasma membrane is first destroyed, exposing the MD, which is degraded minutes later. They showed that PI3KC3 complex members first associate with MD (Rubicon), followed by Atg8a (though, it is not clear when lysosomes arrive at the MD). They identified numerous genes important for this process, including LAP, autophagy, and ROS genes. They showed that certain biomarkers for LAP activity or function are present (e.g. PtdIns3P, ROS). They also deconvolved the spatial relationship between the sperm MD and axoneme, revealing that LAP is specifically targeting the MD/plasma membrane, but not the axoneme. Altogether, this study advances our understanding of the mechanisms and players that facilitate PME.

While all of the above are great strengths of the paper, this revision has three serious problems that need to be addressed.

First, the authors propose a paradigm in which sperm are viewed by the egg as 'invaders'. This interesting idea is entirely speculative and untested. Yet the paper (abstract and elsewhere) is written to imply that the model is the clear outcome of their study. Other possibilities remain, such as that the microbe-directed LAP degradation pathway was repurposed in the fly egg to recognize substrates unrelated to immunity (more details below). The authors are welcome to speculate about their 'invader' model in the Discussion, but they must tone down the presentation.

Second, many problems with the writing and organization make the paper very hard follow, and its results and logic hard to grasp.

Writing: This revision includes many very long and complex sentences, such as in lines 39-42, 66-69, 182-190, 277-281. Breaking them into smaller sentences would improve the paper's clarity .

Organization: The paper's organization is confusing. It is possible that it follows the organization of the original submission, but if so it has been confused by insertion of additional data. A major reorganization, by the temporal order of events, would make the paper easier to follow and the significance of its results clearer. Here is one possible organization:

1. Sperm plasma membrane and then MD, but not axoneme, are degraded in a Rubicon-dependent manner. Then, the rest of the paper can focus on MD destruction.
2. Rubicon localizes to CD63-positive MVBs, which associate with the MD in segments. Pieces of MD then populate Rubicon-positive structures. Purified MVBs from early embryos are enriched for expected classes of proteins, including mitochondria and immunity. (Although immunity proteins are not the most significantly-enriched category, including them here can be used to motivate the focus on LAP).
3. Rubicon and other PI3KC3 members are required for MD degradation.
4. Signatures of LAP activity are present (PtdIns3P, ROS) and some are Rubicon-regulated (PtdIns3P).
5. Autophagy machinery (Atg8a) arrives next at the MD, and Atg8a or b are required for efficient MD degradation. Rubicon is required for proper timing of Atg8a-MD association.
6. Lysosomes are the terminal step of the pathway, so makes sense to end with LAMP1 staining (although the timing of lysosome association with the MD is not tested here, nor is whether lysosome association requires upstream LAP-factors).

Third, the addition of new data and text have made the paper enormous and thus hard to follow. This can be helped in part by moving the following figures into the Extended Data:

Figure 4 – it is good to show some detail about Rubicon localizing specifically to the membrane of CD63 vesicles. But the speculation about whether it is biogenesis or recruitment is presented without a conclusion and belongs in a supplement.

Figure 5 – Because the new data show that sperm is already rich in ROS before fertilization, and because the authors have not shown that Duox has any effect on ROS staining intensity, their results about Duox are now hard to interpret. They should be in a supplement.

Figure 7 – could be supplement, as it simply shows that their Atg8a staining is specific.

Figure 8 – Could put the Atg8a-b knockdown part in the supplement (could show the double knockdown only in figure 6, and move the rest to supplement).

Specific comments

- Lines 34-37 in the abstract feel more grandiose than the data can support. The authors should just say that their model might be conserved. Similarly, if they want to comment on the notion of immune defense, they need to tone down their language. It is convincing that LAP is used. While this might suggest a tie to immunity, the authors never show that the machinery specifically targets sperm components (or, sperm + other microbes, for example).

- Lines 42-43, most species, not just mammals, have highly morphologically distinct sperm/egg cells. Make the text more general.

- Lines 63-64, the authors mention data from interspecific mouse crosses without explaining why they believe their interspecific nature invalidates those studies.

- Line 158 notes that all transgenes are UASp or UASz, but some generated transgenes are driven by dj or ubiquitously expressed. It would be more accurate to just list the promoter in the figure legends. (e.g. UASp-Rubicon; Ubi-LAMP1; Rubicon-GFP). It would also make it clear to the reader whether a construct is overexpressed, endogenously tagged, or driven by a different promoter.

- In figure 3E (and video), why are there so many green puncta throughout the cytoplasm? Is this an artifact of ExM + superresolution imaging? Or is it staining all mitochondria rather than just fertilized with fluorescent sperm mitochondria? Or is there some other explanation?

- All bar graphs are shown as stacked bar charts, but would be easier to interpret if, instead, a single condition is plotted in any graph – for example, % Rubicon-positive puncta only rather than plotting both % positive and negative. With non-stacked graphs, it is also easier to display error bars, individual data points, etc.

- They show that Duox KD weakly affects PME, and claim that Duox is responsible for some ROS production (line 324), but they don't show that H2DCFDA signal is modified in Duox KD conditions.

- Figure 5 data presentation jumps from ROS staining in sperm inside the egg, to Duox KD effects on PME, and then jump back to ROS staining in sperm from the seminal vesicle (which brings the

interpretation of this experiment into question). Recommend that the SV sperm staining should be included above, or should be put in the supplement.

- Figure 6 legend, please note what I, IIa,b, and III mean. It is in the main text, but it will be much easier to locate this information in the legend.

- Figure 8, It was surprising that they did not test lysosome association in any of their knockdown/knockouts (unless this is so labor-intensive that it belongs in a future paper).

- Remove lines 415-417, and simply say similar results were found in a mouse study.

- They still propose this notion throughout the paper that the sperm might be treated as a foreign invading microbe, thus triggering LAP. But they don't show that LAP isn't already acting to degrade components of the maternal cytoplasm (there are Rubicon-positive structures everywhere). They have no 'positive control' of an invading microbe here to show that LAP actually could act in the fly egg milieu. If LAP is acting already to degrade maternal materials, then the sperm might just be entering an already 'hostile' environment. The fact that the axoneme isn't degraded shows that the system must be able to specify what to degrade; but, it doesn't prove that this is a sperm-specific phenomenon or that the sperm triggers the activity of LAP machinery.

- Line 536, why do negative effects from assisted repro technologies "inevitably" involve paternal mitochondrial leakage?

- Line 542, three parents??

- Line 579, "the" is underlined?

- Line 637, is "undiluted 6% bleach" a mistake? Did they mean bleach diluted to 6%?

- Line 747, should be 'replaced'

- Lines 772-774 reference the mouse sperm Rubicon staining, so should be removed.

- Lines 896-898 are underlined?
- Line 913, please state which experiments only had one biological replicate.
- Line 924, “Reagents” is misspelled
- Line 1441, “flagellum” is misspelled

Reviewer #3 (Remarks to the Author):

This manuscript dramatically advances our understanding of paternal mitochondrial elimination (PME) in *Drosophila melanogaster* embryos. Using a powerful combination of multivesicular body (MVB) proteomics, genetic analysis, and high resolution imaging of MVBs and flagellar vesicular sheaths, the authors convincingly describe a LC3-associated phagocytosis (LAP)-related pathway for destruction of paternal mitochondrial derivatives (MD) but not axonemes following fertilization. The authors draw interesting parallels between MDs and intracellular pathogens, potentially explaining the similarities in their mechanisms of elimination by the invaded "host" cell. In addition, they describe evidence from the literature suggesting that a similar pathway may carry out PME in mammalian embryos.

I feel the authors have satisfactorily addressed all the reviewers' comments and have dramatically improved the imaging and other aspects of the manuscript (clarity of the writing, etc.) in this revised version. The new Supplementary Video 3 is particularly spectacular.

I found only a couple of small things that should be addressed prior to publication:

In line 63, the word "an" should be deleted if multiple crosses are being referred to (as indicated in line 64).

The dashed rectangles in Fig. 3b appear to correspond to a much larger area of the image than that shown in the magnified images in Fig. 3c. Indeed, it was hard to figure out precisely which Rubicon/

MD fragment-positive organelles in Fig. 3b correspond to those shown in Fig. 3c. This could be addressed by the use of smaller dashed rectangles, potentially with the addition of arrows that point directly to the vesicles of interest, in Fig. 3b.

Reviewer #4 (Remarks to the Author):

We thank the Reviewers for their additional comments and input which we addressed in the text of the last version of the manuscripts. Reviewers' comments are in black text, while our replies are in blue text.

REVIEWER COMMENTS

Reviewer #1 (Remarks to the Author):

In the revised manuscript, the authors addressed most my concerns as well as comments raised by other reviewers. The manuscript is nicely improved and seems more reader-friendly. I endorse the current version for publication.

We thank the Reviewer for all his comments which significantly helped improving our manuscript.

Reviewer #2 (Remarks to the Author):

The authors present a model for degradation of the *Drosophila* paternal mitochondrial derivative (MD) by a LAP-like pathway. In their revision, they responded well to most of the comments, including by removing the mammalian work that had concerned several referees, and adding new data of excellent quality to give higher resolution views of the proteins' and structures' relative localizations. The added data, in combination with the original paper's, favor an interesting model with a fairly linear progression of events. The authors resolved the temporal sequence of many events. They determined that the sperm plasma membrane is first destroyed, exposing the MD, which is degraded minutes later. They showed that PI3KC3 complex members first associate with MD (Rubicon), followed by Atg8a (though, it is not clear when lysosomes arrive at the MD). They identified numerous genes important for this process, including LAP, autophagy, and ROS genes. They showed that certain biomarkers for LAP activity or function are present (e.g. PtdIns3P, ROS). They also deconvolved the spatial relationship between the sperm MD and axoneme, revealing that LAP is specifically targeting the MD/plasma membrane, but not the axoneme. Altogether, this study advances our understanding of the mechanisms and players that facilitate PME.

We thank the Reviewer for the appreciation of the significance of our study.

While all of the above are great strengths of the paper, this revision has three serious problems that need to be addressed.

First, the authors propose a paradigm in which sperm are viewed by the egg as 'invaders'. This interesting idea is entirely speculative and untested. Yet the paper (abstract and elsewhere) is written to imply that the model is the clear outcome of their study. Other possibilities remain, such as that the microbe-directed LAP degradation pathway was repurposed in the fly egg to recognize substrates unrelated to immunity (more details below). The authors are welcome to speculate about their 'invader' model in the Discussion, but they must tone down the presentation.

This point is now toned-down in the text and only appears in the Discussion section.

Second, many problems with the writing and organization make the paper very hard follow, and its results and logic hard to grasp.

Writing: This revision includes many very long and complex sentences, such as in lines 39-42, 66-69, 182-190, 277-281. Breaking them into smaller sentences would improve the paper's clarity.

Done.

Organization: The paper's organization is confusing. It is possible that it follows the organization of the original submission, but if so it has been confused by insertion of additional data. A major reorganization, by the temporal order of events, would make the paper easier to follow and the significance of its results clearer. Here is one possible organization:

1. Sperm plasma membrane and then MD, but not axoneme, are degraded in a Rubicon-dependent manner. Then, the rest of the paper can focus on MD destruction.
2. Rubicon localizes to CD63-positive MVBs, which associate with the MD in segments. Pieces of MD then populate Rubicon-positive structures. Purified MVBs from early embryos are enriched for expected classes of proteins, including mitochondria and immunity. (Although immunity proteins are not the most significantly-enriched category, including them here can be used to motivate the focus on LAP).
3. Rubicon and other PI3KC3 members are required for MD degradation.
4. Signatures of LAP activity are present (PtdIns3P, ROS) and some are Rubicon-regulated (PtdIns3P).
5. Autophagy machinery (Atg8a) arrives next at the MD, and Atg8a or b are required for efficient MD degradation. Rubicon is required for proper timing of Atg8a-MD association.
6. Lysosomes are the terminal step of the pathway, so makes sense to end with LAMP1 staining (although the timing of lysosome association with the MD is not tested here, nor is whether lysosome association requires upstream LAP-factors).

While both Reviewers #1 and #3 feel that the paper is now more reader-friendly, we agree that the addition of ample amount of new data might be somewhat hard to follow for the non-professional reader. Therefore, we made some text rearrangements which also involved some figure shuffling in accordance with the Reviewer suggestions.

The Major changes are as follows:

- I. Old Fig. 9 (describing the breakdown of the sperm plasma membrane) is now presented as Fig. 4 (accordingly, Supplementary video 10 is now Supplementary video 4).
- II. The main findings in old Figs. 4 and 5, which presented the data about the hCD63 MVB marker and PtdIns(3)P and ROS, respectively, were merged to new Fig. 5, while the less significant data was merged and is now presented in new Extended Data Fig. 5.
- III. The second part of old Fig. 8, which presented single and double knockdowns of *atg8a* and *atg8b*, was split, such that the double knockdown data is now presented in new Fig. 6b, while the single knockdowns (which have no effect on PME kinetics) were moved to new Extended Data Fig. 6b-e. Consequently, the first part of old Fig. 8 was merged with the illustration of the model (old Fig. 10) and both are now presented as new Fig. 8.

Third, the addition of new data and text have made the paper enormous and thus hard to follow. This can be helped in part by moving the following figures into the Extended Data:

Figure 4 – it is good to show some detail about Rubicon localizing specifically to the membrane of CD63 vesicles. But the speculation about whether it is biogenesis or recruitment is presented without a conclusion and belongs in a supplement.

Figure 5 – Because the new data show that sperm is already rich in ROS before fertilization, and because the authors have not shown that Duox has any effect on ROS staining intensity, their results about Duox are now hard to interpret. They should be in a supplement.

Figure 7 – could be supplement, as it simply shows that their Atg8a staining is specific.

Figure 8 – Could put the Atg8a-b knockdown part in the supplement (could show the double knockdown only in figure 6, and move the rest to supplement).

We accepted and applied all the above suggestions, except for the suggestion to move Fig. 7 to the Extended Data, as this Figure is not merely a confirmation of antibody specificity, but rather indicates that recruitment of Atg8 to the FVS requires Rubicon.

Specific comments

- Lines 34-37 in the abstract feel more grandiose than the data can support. The authors should just say that their model might be conserved. Similarly, if they want to comment on the notion of immune defense, they need to tone down their language. It is convincing that LAP is used. While this might suggest a tie to immunity, the authors never show that the machinery specifically targets sperm components (or, sperm + other microbes, for example).

Done.

- Lines 42-43, most species, not just mammals, have highly morphologically distinct sperm/egg cells. Make the text more general.

Done.

- Lines 63-64, the authors mention data from interspecific mouse crosses without explaining why they believe their interspecific nature invalidates those studies.

Done.

- Line 158 notes that all transgenes are UASp or UASz, but some generated transgenes are driven by dj or ubiquitously expressed. It would be more accurate to just list the promoter in the figure legends. (e.g. UASp-Rubicon; Ubi-LAMP1; Rubicon-GFP). It would also make it clear to the reader whether a construct is overexpressed, endogenously tagged, or driven by a different promoter.

We change the text accordingly and also added this information in the specific Fig. legends.

- In figure 3E (and video), why are there so many green puncta throughout the cytoplasm? Is this an artifact of ExM + superresolution imaging? Or is it staining all mitochondria rather than just fertilized with fluorescent sperm mitochondria? Or is there some other explanation?

The following information was added to the legends of Supplementary video 3: “Multiple non-specific green puncta scattered throughout the non-computed 3D projection may correspond to endogenously biotinylated proteins (as staining of the MD involved an anti-GFP antibody conjugated to Biotin, followed by a fluorescently tagged streptavidin).”

- All bar graphs are shown as stacked bar charts, but would be easier to interpret if, instead, a single condition is plotted in any graph – for example, % Rubicon-positive puncta only rather than plotting both % positive and negative. With non-stacked graphs, it is also easier to display error bars, individual data points, etc.

Where possible, we added individual data points to the bar charts.

The specific graphs in new Fig. 5g and Extended Data Fig. 5b represent contingency tables analyzed by Fisher's exact test. These graphs represent numerical results of experiments in which the outcome is a categorical variable (yes or no flagellar labeling), hence there are no individual data points or error bars to present.

- They show that Duox KD weakly affects PME, and claim that Duox is responsible for some ROS production (line 324), but they don't show that H2DCFDA signal is modified in Duox KD conditions.

Although ROS is readily detected on the sperm flagellum, the role of Duox in flagellar ROS accumulation is still unclear. Since the effect of *duox* knockdown on PME is only mild-moderate, we decided (as also suggested by the Reviewer) to move this result to new Extended data Fig. 5h-i.

- Figure 5 data presentation jumps from ROS staining in sperm inside the egg, to Duox KD effects on PME, and then jump back to ROS staining in sperm from the seminal vesicle (which brings the interpretation of this experiment into question). Recommend that the SV sperm staining should be included above, or should be put in the supplement.

This figure was moved to the Extended data Fig. 5j.

- Figure 6 legend, please note what I, IIa,b, and III mean. It is in the main text, but it will be much easier to locate this information in the legend.

Done.

- Figure 8, It was surprising that they did not test lysosome association in any of their knockdown/knockouts (unless this is so labor-intensive that it belongs in a future paper).

Although this point is interesting, the fact that we detect lysosomal contents inside the string of large Atg8 positive vesicles, is consistent with lysosomes participating in the degradation of the MD fragments. We agree that such questions definitely have a place in a future paper.

- Remove lines 415-417, and simply say similar results were found in a mouse study.

Done.

- They still propose this notion throughout the paper that the sperm might be treated as a foreign invading microbe, thus triggering LAP. But they don't show that LAP isn't already acting to degrade components of the maternal cytoplasm (there are Rubicon-positive structures everywhere). They have no 'positive control' of an invading microbe here to show that LAP actually could act in the fly egg milieu. If LAP is acting already to degrade maternal materials, then the sperm might just be entering an already 'hostile' environment. The fact that the axoneme isn't degraded shows that the system must be able to specify what to degrade; but, it doesn't prove that this is a sperm-specific phenomenon or that the sperm triggers the activity of LAP machinery.

As abovementioned, we toned-down this theory.

•Line 536, why do negative effects from assisted repro technologies “inevitably” involve paternal mitochondrial leakage?

Engagement of microbes with the cell plasma membrane is known to be essential for labeling of the invader microbes by ubiquitination and subsequent targeting by autophagy (xenophagy). Since most of the MAR technologies involve Intracytoplasmic sperm injection (ICSI), which bypasses sperm-egg engagement, targeting of the paternal mitochondria following ICSI might be impaired. To be less conclusive, we now changed “inevitably” to “may”.

•Line 542, three parents??

The mitochondrial replacement therapy (MRT) indeed involves three parents, as the procedure involves the transfer of a zygote nucleus from a fertilized egg with unhealthy mitochondria into a healthy fertilized egg from which the nucleus was removed. Therefore, there is a third “mother” that donates the mitochondria (with their genetic material).

•Line 579, “the” is underlined?

Erased.

•Line 637, is “undiluted 6% bleach” a mistake? Did they mean bleach diluted to 6%?

This was corrected.

•Line 747, should be ‘replaced’

Done.

•Lines 772-774 reference the mouse sperm Rubicon staining, so should be removed.

Done.

•Lines 896-898 are underlined?

Erased.

•Line 913, please state which experiments only had one biological replicate.

By now, all experiments were repeated at least twice. The text was corrected accordingly.

•Line 924, “Reagents” is misspelled

Corrected.

•Line 1441, “flagellum” is misspelled

Corrected.

Reviewer #3 (Remarks to the Author):

This manuscript dramatically advances our understanding of paternal mitochondrial elimination (PME) in *Drosophila melanogaster* embryos. Using a powerful combination of multivesicular body (MVB) proteomics, genetic analysis, and high resolution imaging of MVBs and flagellar vesicular sheaths, the authors convincingly describe a LC3-associated phagocytosis (LAP)-related pathway for destruction of paternal mitochondrial derivatives (MD) but not axonemes

following fertilization. The authors draw interesting parallels between MDs and intracellular pathogens, potentially explaining the similarities in their mechanisms of elimination by the invaded "host" cell. In addition, they describe evidence from the literature suggesting that a similar pathway may carry out PME in mammalian embryos.

I feel the authors have satisfactorily addressed all the reviewers' comments and have dramatically improved the imaging and other aspects of the manuscript (clarity of the writing, etc.) in this revised version. The new Supplementary Video 3 is particularly spectacular.

We thank Prof. Julie Brill for all her input and appreciation of our study.

I found only a couple of small things that should be addressed prior to publication:

In line 63, the word "an" should be deleted if multiple crosses are being referred to (as indicated in line 64).

Done.

The dashed rectangles in Fig. 3b appear to correspond to a much larger area of the image than that shown in the magnified images in Fig. 3c. Indeed, it was hard to figure out precisely which Rubicon/ MD fragment-positive organelles in Fig. 3b correspond to those shown in Fig. 3c. This could be addressed by the use of smaller dashed rectangles, potentially with the addition of arrows that point directly to the vesicles of interest, in Fig. 3b.

This was corrected.

Julie Brill

Reviewer #4 (Remarks to the Author):

We think that this is a great initiative and thank the Reviewer for his comments.

REVIEWERS' COMMENTS

Reviewer #2 (Remarks to the Author):

Original Reviewer 2: The authors have now fully addressed my concerns. The two rounds of revision greatly improved this report's accuracy and accessibility. It is now an excellent and interesting paper, and I recommend publication.